# SIMULATOR-BASED SYNTHETIC ECGS FOR SELF-SUPERVISED PRETRAINING

## ABSTRACT

Medical data remain scarce and sensitive despite rich domain knowledge. We ask whether knowledge-driven parametric ECG simulators can supply scalable self-supervised pretraining signals without using patient records during pretraining. We use two established simulator-based ECG generators to synthesize 10-s, 500-Hz lead II signals for pretraining a Transformer encoder with masked autoencoding, and compare it to pretraining on real PTB-XL ECGs and on VAE/GAN-generated ECGs. We then fine-tune on 26 abnormal-ECG classification tasks across PTB-XL, G12EC, and CPSC2018 and benchmark against five strong supervised baselines. The Transformer pretrained on simulator-based synthetic ECG (SimECG) performs comparably to real-data pretraining and outperforms supervised baselines on 24 of 26 tasks, yielding a mean $5.49\%$ relative $F_1$ improvement over the strongest baseline. Under reduced labeled-data budgets and across patient demographics, it largely preserves the advantages of real-data pretraining and maintains competitive performance across all 12 single-lead configurations. Crucially, simulator pretraining still avoids any exposure to patient data during pretraining. These results indicate that knowledge-driven synthetic ECG corpora can provide practical, privacy-enhancing initialization for downstream ECG models in data-limited regimes.

## 1 INTRODUCTION

Self-supervised pretraining on large-scale datasets underpins modern foundation models in language and vision, enabling a single encoder to support many downstream tasks (Brown et al., 2020; Ramesh et al., 2021; Rombach et al., 2022). These models are typically trained on vast, heterogeneous collections of real data, whose diversity makes the resulting representations broadly transferable.

In clinical settings, assembling comparable corpora is difficult: privacy regulations limit data sharing, expert labels are expensive, and distributions vary across devices and populations. Existing approaches either pretrain directly on large institutional repositories of ECG and other time-series data, or train high-capacity generative models to synthesize additional examples (Biswal et al., 2021; Chintagunta et al., 2021; Hannun et al., 2019; Naseer et al., 2023). Both strategies ultimately rely on substantial amounts of real patient data, and the generators themselves can memorize sensitive records (Hayes et al., 2017; Webster et al., 2019; Carlini et al., 2023). In this work, we focus on an intermediate data regime: clinical time-series domains where unlabeled recordings exist in the thousands to tens of thousands—large enough that self-supervised pretraining is meaningful, but far below the scales routinely used for foundation models.

A complementary route is to exploit decades of domain knowledge. Knowledge-driven simulators provide parameterized, physiologically plausible generators for a range of signals, including glucose dynamics (Man et al., 2014), electrodermal activity (Bach et al., 2010), and electrocardiograms (ECGs) (McSharry et al., 2003; Sayadi et al., 2010; Nonaka & Seita, 2024). Such simulators can produce essentially unbounded synthetic corpora without accessing patient records and therefore offer an appealing, privacy-preserving pretraining source.

ECG is also unusual among clinical modalities in having several large public datasets (e.g., MIMIC-IV-ECG, HEEDB, and PTB-XL), comprising on the order of $10^5$–$10^7$ 12-lead recordings (Gow et al., 2023; Wagner et al., 2020; Goldberger et al., 2000; Sun et al., 2025; Ribeiro et al., 2020). This

makes ECG a natural testbed for studying whether simulator-based synthetic pretraining can replace real-data pretraining for representation learning in such intermediate regimes. In our main experiment, we deliberately fix the encoder family and objective—a Transformer encoder (Vaswani et al., 2017) trained with masked autoencoding (MAE; He et al., 2022)—and vary only the pretraining corpus. Specifically, we pretrain on (i) knowledge-driven simulator ECGs, (ii) real PTB-XL ECGs, and (iii) VAE- and GAN-generated ECGs trained on PTB-XL, then fine-tune on 26 abnormal-ECG classification tasks from PTB-XL, G12EC, and CPSC2018, alongside five strong supervised baselines.

Across these tasks, simulator-based pretraining yields downstream performance close to real-data pretraining while consistently outperforming strong supervised baselines and improving robustness to limited labels, patient demographics, and ECG leads. During pretraining, we focus on single-lead ECGs (lead II) to match the available knowledge-based simulators and common bedside monitor/wearable configurations, and then verify that the learned representations generalize to other leads and to 12-lead ECGs. Because simulator pretraining never consumes identifiable patient records, it scales without new privacy or governance burdens. Rather than competing with ECG foundation models and multimodal encoders trained on millions of real ECGs and reports (Han et al., 2024; Li et al., 2025; McKeen et al., 2025; Liu et al., 2024; Song et al., 2024), our contribution is orthogonal. We show that knowledge-driven simulators can supply much of the benefit of self-supervised pretraining in intermediate regimes where large clinical repositories cannot be centralized or shared. We further analyze how these benefits evolve as we scale synthetic data and compute.

## 2  RELATED WORK

In the medical domain, synthetic data are widely used to alleviate data scarcity and class imbalance. Deep generative models such as GANs synthesize chest X-ray images to improve generalization and domain adaptation (Salehinejad et al., 2018; Madani et al., 2018; Koga et al., 2018), and VAEs or diffusion models generate electronic health records (Biswal et al., 2021; Chintagunta et al., 2021; Naseer et al., 2023). In most of these studies, synthetic data augment supervised training, and the generators themselves are trained on large collections of real patient records, so they inherit the access and privacy constraints of real-world datasets.

For ECG analysis, deep neural networks have achieved cardiologist-level arrhythmia classification (Hannun et al., 2019) and mitigate label scarcity via augmentation (Zhu et al., 2022; Nonaka & Seita, 2021b; Raghu et al., 2022) or weak labels (Chen et al., 2021). ECG-specific self-supervised learning has progressed through contrastive and reconstruction-based methods such as CLOCS (Kiyasseh et al., 2021) and ST-MEM (Na et al., 2024). Building on these ideas, ECG foundation models pretrain on millions of real recordings, e.g., (Li et al., 2025) and ECG-FM (McKeen et al., 2025), while BiTimelyGPT (Song et al., 2024) and MERL (Liu et al., 2024) extend to multimodal and time-series settings. These works show the benefit of self-supervised and foundation-style pretraining when very large clinical repositories can be centralized, but they all assume access to millions of real ECGs (and often clinical reports) and do not directly address intermediate regimes where only thousands to tens of thousands of unlabeled recordings are available.

ECG synthesis has followed two main paths. Knowledge-driven simulators generate physiologically plausible signals using dynamical models (McSharry et al., 2003; Sayadi et al., 2010) and have been used for augmentation and domain randomization (Kaisti et al., 2023; Landajuela et al., 2022; Nonaka & Seita, 2024). In parallel, data-driven GANs and diffusion models learn high-capacity ECG generators from real corpora for fidelity and controllability (Golany et al., 2020; 2021; Delaney et al., 2019; Zhu et al., 2019; Delaney et al., 2019; Lin et al., 2024; Lai et al., 2025; Alcaraz & Strodthoff, 2023; Zama & Schwenker, 2023; Neifar et al., 2024; Bedin et al., 2024), again mainly to supply auxiliary training data and, because they are trained on large clinical ECG repositories, inheriting the same privacy and governance constraints as real-data pretraining.

**Our work is orthogonal to these lines:** we do not propose a new self-supervised objective or foundation model. Instead, we conduct a controlled comparison where, in our main experiments, encoder architecture and self-supervised learning (SSL) algorithm are fixed while only the pretraining data source varies—simulator-generated, VAE/GAN-generated, or real PTB-XL data. This design isolates the impact of different synthetic data generation approaches on downstream task perfor-

mance, complementing prior work that focuses on algorithmic innovations for large-scale real-world datasets.

## 3 ECG Data

This section describes both real-world and simulator-based synthetic ECG datasets used for classification tasks and pretraining experiments.

### 3.1 Real-world ECG Data

In this work, we use normal and abnormal ECGs extracted from three real-world ECG datasets. The primary real-world dataset used in this work is the PTB-XL dataset (Wagner et al., 2020). This dataset consists of $21,799$ samples of 12-lead ECGs acquired from $18,869$ subjects in Germany between October 1989 and June 1996 using devices manufactured by Schiller AG. Each sample has a duration of ten seconds, with a sampling frequency of $500$ Hz. Each sample is associated with at least one of a total of 71 distinct statements, providing information relevant to disease classification.

**Lead II selection.** We focus on lead II throughout this study as it provides the most consistent P-QRS-T morphology for rhythm analysis and aligns with the output characteristics of both ECG simulators (McSharry et al., 2003; Nonaka & Seita, 2024). We later verify that these conclusions extend to all 12 standard leads and to full 12-lead models (Appendices O and P). We extracted lead II ECG from samples labeled "ABQRS", "AF", "ASMI", "CRBBB", "IAVB", "IMI", "IRBBB", "ISC", "LAFB", "LVH", "PAC", "PVC", "STD", and "NORM". For pretraining comparisons, we prepared two subsets: "PTB-XL_NORM" (normal samples only) and "PTB-XL_All" (all available samples)[1].

### 3.2 Knowledge-Driven Synthetic ECG Data

In this study, we employed two electrocardiogram (ECG) synthesis simulators: McSharry et al. (2003) and Nonaka & Seita (2024). The simulator of McSharry et al. (2003) uses three coupled ordinary differential equations to generate realistic ECG signals with configurable heart rate parameters, PQRST morphology, and spectral components that replicate physiological variations. The method of Nonaka & Seita (2024) synthesizes ECG signals through superposition of Gaussian curves controlled by 15 variables governing peak characteristics and baseline fluctuations. Both simulators were configured to match the PTB-XL dataset specifications (10-second duration, $500$ Hz sampling frequency), with parameters randomly sampled from predefined distributions. We generated $1,000,000$ training samples and $10,000$ validation samples from each simulator to support subsequent pretraining applications requiring large-scale ECG data.[2] Hereafter, we refer to data generated by these knowledge-driven simulators collectively as SimECG. Specifically, we write SimECG-M for data generated by the McSharry model (McSharry et al., 2003) and SimECG-N for data generated by the Nonaka model (Nonaka & Seita, 2024).

## 4 Method: Self-supervised Pretraining with Simulator-Synthesized ECG

We study whether *purely simulator-based* pretraining can match real-data pretraining for ECG representation learning. Our main experiments use a Transformer encoder with a masked autoencoder (MAE; He et al., 2022) objective; in supplementary analyses we also consider SimCLR (Chen et al., 2020) and DINO (Caron et al., 2021). Pretraining is performed under three data conditions: (1) simulator-synthesized ECG, (2) real-world ECG from PTB-XL, and (3) ECG generated by VAEs or GANs trained on PTB-XL. For each method–data combination, we conduct pretraining and then evaluate the models on abnormal-ECG classification tasks by fine-tuning on the respective training splits. Further details of architecture selection and ablations are provided in Appendix D.2, and generative-model pretraining settings are given in Appendix G.

---

[1]Further details of the real-world datasets including CPSC and G12EC are in Appendix B.

[2]Details of the synthesis are shown in Appendix C.

## 5 EXPERIMENTS

We evaluate along four axes: (i) overall classification performance across 26 abnormal-ECG tasks drawn from three public datasets (PTB-XL, G12EC, and CPSC2018), (ii) data efficiency under training-set subsampling on PTB-XL, (iii) demographic robustness (age and sex) on PTB-XL,[3] and (iv) input lead robustness on PTB-XL. We use $F_1$-score as the primary metric throughout, aggregating it differently for each axis: mean $F_1$ and $F_1$ rank for (i); mean $F_1$ for (ii); normalized $F_1$ disparity for (iii); mean $F_1$ rank for (iv). Preliminary studies for selecting the Transformer configuration and supervised baselines are consolidated in Appendix D.2 and Appendix D.3.

### 5.1 DATA SPLITS

We adopt a consistent procedure across tasks and datasets. For each target abnormal class, we extract all labeled samples and form a binary dataset by merging the abnormal subset (positive) with the NORM subset (negative). We first partition the data into a development set (train + validation) and a test set in an 8:2 ratio. Within the development set, we create six independent 8:2 splits: one split is used for hyperparameter optimization (HPO), and the remaining five are used to train five models with the selected hyperparameters. We report mean±SD on the common held-out test set. Details of pretraining and preliminary-experiment splits are provided in Appendix D.2.

### 5.2 TRAINING PROTOCOL FOR SUPERVISED FINE-TUNING

Unless otherwise noted, inputs are per-sample standardized (subtract mean, divide by standard deviation). Data augmentation includes random temporal shifting and random masking, with rates tuned via HPO. We use batch size 512 and train for up to 500 epochs with validation every five epochs and early stopping after five non-improving validations. Optimization uses Adam (Kingma & Ba, 2014) with the HPO-selected learning rate and binary cross-entropy loss, with positive class weighting by the negative-to-positive ratio to mitigate class imbalance.

### 5.3 MAIN EXPERIMENT: EVALUATING SYNTHETIC-DATA PRETRAINING

We compare Transformer (SimECG) against Transformer (PTB-XL) and Transformer (VAE/GAN), as well as five supervised baselines and Transformer (Scratch).[4] Hereafter, we use a uniform notation for pretrained models: Transformer (SimECG) for simulator-synthesized ECG pretraining, Transformer (PTB-XL) for real-world PTB-XL pretraining, and Transformer (Scratch) for training from random initialization. When applicable, Transformer (VAE/GAN) denotes pretraining on deep generative ECG.

**Pretraining corpora and objectives.** We consider three pretraining data families: (i) knowledge-driven simulators (SimECG-M/N), (ii) real-world PTB-XL (lead-II; PTB-XL_NORM and PTB-XL_All), and (iii) deep generative ECG (VAE/GAN) trained on PTB-XL. For each simulator, we generate 1,000,000 training and 10,000 validation samples at 500 Hz/10 s to match PTB-XL specifications; for MAE we use a Transformer encoder with batch size 512 and Adam (lr $10^{-3}$), and mask 75% of tokens unless noted, training up to $\sim 1$B sample exposures and selecting the best checkpoint on a pretraining validation split. Further details of architecture is in Appendix E, generative-model settings in Appendix G. *Pretraining never uses labels from downstream tasks.*

**Compute-normalized comparison.** To ensure that performance differences are not artifacts of unequal computational budgets, we track downstream PTB-XL $F_1$ as a function of the number of samples processed during pretraining, comparing models pretrained on PTB-XL, G12EC, and SimECG under identical computational budgets ranging from $10^6$ to $10^9$ processed samples.

**Baselines and HPO summary.** We compare the pretrained Transformer against five supervised baselines (GRU, LSTM, MEGA, ResNet-18, and Luna) selected from an initial pool of 16 architectures via a supervised AF vs. NORM screen; HPO is performed with Optuna for 6 hours per

---

[3]Compute resources are summarized in Appendix A.

[4]Details of baseline model selection is in Appendix D.3.

{architecture, task}, and the best settings are reused in the five retraining runs. Search spaces, early stopping, and optimization details follow Appendix E. *A Transformer trained from scratch* and *a real-data-pretrained Transformer* serve as additional references.

**Statistical testing.** For all experiments we run each model five times with different random seeds and report the mean and standard deviation of $F_1$. To assess statistical significance, we apply non-parametric Wilcoxon signed-rank tests with Bonferroni correction (Dunn, 1961) across the 26 classification tasks. For comparisons between pretraining data sources (Table 1), we use a non-inferiority test with margin $\delta = 0.025$ (allowing up to 2.5% relative decrease in $F_1$) relative to the Real-world (PTB-XL) condition. For comparisons between Transformer (SimECG) and supervised baselines (Table 2), we test for superiority. Details are in Appendix F.

### 5.3.1 COMPARING CLASSIFICATION PERFORMANCE

Across 26 tasks (13 on PTB-XL, 7 on G12EC, 6 on CPSC2018), each model undergoes HPO for six hours on the dedicated HPO split (Appendix E); we then retrain using the best setting on each of the five train/validation splits and evaluate on the common held-out test set. We report $F_1$ as the metric.

### 5.3.2 REDUCING TRAINING DATA SIZE

For 13 PTB-XL tasks, we assess data efficiency by subsampling the training set to $n$ positives and $n$ negatives with $n \in \{1, 2, 5, 10, 25, 50, 100, 250, 500, 1000\}$.[5] We train the supervised baselines and both pretrained Transformers (SimECG-N and PTB-XL) under the same hyperparameters selected in the full-data setting and evaluate on the full test set.

### 5.3.3 PERFORMANCE GAPS ACROSS DEMOGRAPHICS

Using PTB-XL metadata, we quantify performance gaps across sex and age on the test set for 13 tasks. We evaluate eight models: the Transformer (SimECG-N), the Transformer (PTB-XL), the five supervised baselines, and a Transformer (Scratch). For each of the five runs per setting, we compute the overall score $F_{1,\text{all}}$. For sex, we compute $F_{1,f}$ and $F_{1,m}$ on female and male subsets and report the normalized absolute difference $|F_{1,f} - F_{1,m}|/F_{1,\text{all}}$. For age, we partition patients into four groups[6] and compute $F_{1,\text{group1}} - F_{1,\text{group4}}$; the summary statistic is $\left(\max_g F_{1,g} - \min_g F_{1,g}\right)/F_{1,\text{all}}$.

### 5.3.4 LEAD-WISE EVALUATION PROTOCOL (PTB-XL)

Beyond the default lead II, we repeat training and evaluation on single-lead ECGs for I, II, III, aVF, aVL, aVR, and V1–V6 to assess robustness to electrode placement. Unless stated otherwise, we reuse the lead II fine-tuning hyperparameters for each lead.[7]

## 6 RESULTS

### 6.1 OVERALL CLASSIFICATION PERFORMANCE

We quantify how pretraining data source affects downstream abnormal-ECG classification for an MAE-based Transformer encoder, then compare against strong supervised baselines across all datasets.

**Transformer (SimECG-M/N) vs. Transformer (PTB-XL/VAE/GAN)** Table 1 summarizes the effect of the pretraining data source on downstream abnormal-ECG classification. On PTB-XL, the Transformer (SimECG-N) *nearly matches* Transformer (PTB-XL), while Transformer (SimECG-M) remains competitive. Both Tranformer (SimECG-M/N) conditions outperform Transformer (VAE/GAN) on every dataset.

---

[5]If the available training examples are fewer than $n$, we use all available.

[6]Under 39; 40–53; 54–64; $\geq$65 years (25th, 50th, 75th percentiles of PTB-XL NORM).

[7]Due to dataset issues, V5 excludes ASMI/ISC/STD (Appendix O).

Table 1: PTB-XL, G12EC, and CPSC (MAE pretraining). Performance of Transformers pretrained on different data sources. We report mean $F_1$ ($\bar{F}_1$; higher is better) and average rank $\bar{r}$ (lower is better) across tasks in each dataset. **Bold** indicates the best performance. Full per-task tables: PTB-XL (Table 12), G12EC (Table 13), CPSC (Table 14).

| | PTB-XL (13) | | G12EC (7) | | CPSC (6) | |
|---|---|---|---|---|---|---|
| Pretraining data | $\bar{F}_1 \uparrow$ | $\bar{r} \downarrow$ | $\bar{F}_1 \uparrow$ | $\bar{r} \downarrow$ | $\bar{F}_1 \uparrow$ | $\bar{r} \downarrow$ |
| Real-world (PTB-XL) | **0.8488** | **1.54** | **0.8715** | **1.14** | 0.9347 | 2.00 |
| SimECG-M (McSharry et al., 2003) | 0.8222 | 3.08 | 0.8368 | 2.71 | 0.9226 | 2.67 |
| SimECG-N (Nonaka & Seita, 2024) | 0.8451 | 1.85 | 0.8491 | 2.57 | **0.9365** | **1.50** |
| VAE (Kingma & Welling, 2013) | 0.7763 | 4.88 | 0.7176 | 4.86 | 0.8666 | 4.67 |
| DCGAN (Radford et al., 2015) | 0.8111 | 3.65 | 0.7795 | 3.71 | 0.8825 | 4.17 |

Table 2: Mean $F_1$ ($\bar{F}_1$; higher is better) and average rank $\bar{r}$ (lower is better) computed from per task $F_1$ within each dataset for supervised baselines and a Transformer (SimECG-N). PTB-XL: 13 tasks; G12EC: 7 tasks; CPSC: 6 tasks. **Bold** indicates the best performance. Full per-task tables: PTB-XL (Table 21), G12EC (Table 22), CPSC (Table 23).

| | | PTB-XL (13) | | G12EC (7) | | CPSC (6) | |
|---|---|---|---|---|---|---|---|
| Model | Pretraining | $\bar{F}_1 \uparrow$ | $\bar{r} \downarrow$ | $\bar{F}_1 \uparrow$ | $\bar{r} \downarrow$ | $\bar{F}_1 \uparrow$ | $\bar{r} \downarrow$ |
| GRU | - | 0.7854 | 2.92 | 0.8136 | 2.71 | 0.9246 | 3.33 |
| LSTM | - | 0.7726 | 4.08 | 0.7834 | 4.13 | 0.9253 | 3.00 |
| Luna | - | 0.7228 | 6.54 | 0.7273 | 6.43 | 0.8176 | 6.83 |
| MEGA | - | 0.7601 | 5.08 | 0.7616 | 4.57 | 0.8674 | 4.50 |
| ResNet-18 | - | 0.7617 | 3.46 | 0.7380 | 3.71 | 0.8857 | 4.17 |
| Transformer | - | 0.7378 | 4.92 | 0.7526 | 5.14 | 0.8825 | 4.83 |
| Transformer | SimECG-N | **0.8451** | **1.00** | **0.8491** | **1.29** | **0.9365** | **1.33** |

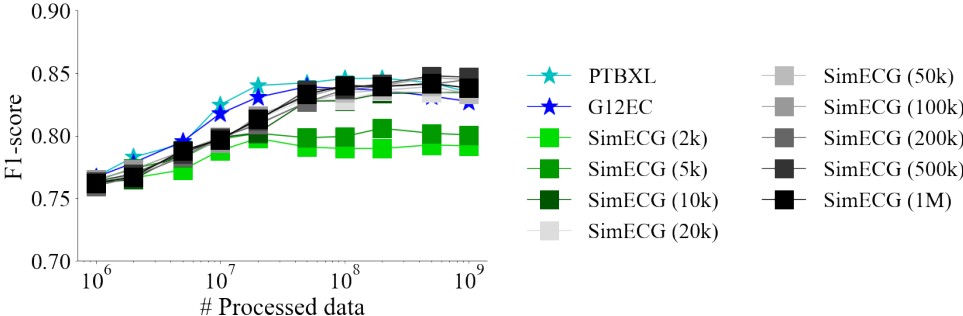

Figure 1: PTB-XL downstream performance (macro $F_1$) as a function of the number of processed pretraining samples. We compare pretraining on real PTB-XL, G12EC, and SimECG-N synthetic datasets with varying sizes under equal computational budgets.

To formalize this observation, we apply the non-inferiority test, treating Transformer (PTB-XL) as the reference and using a relative margin $\delta = 0.025$. Across all 26 tasks, Transformer (SimECG-N) is non-inferior to real-data pretraining (Bonferroni-corrected $p = 0.0008$), whereas the null hypothesis cannot be rejected for Transformer (SimECG-M/VAE/GAN) at this margin. Sensitivity analyses over $\delta \in \{0.01, 0.05, 0.10\}$ show that both Transformer (SimECG-M/N) become non-inferior for $\delta = 0.05$, Transformer (GAN) achieves non-inferiority only for $\delta = 0.10$, and Transformer (VAE) remains inferior even with this most permissive margin. Detailed $p$-values are provided in the Appendix F.2.2. These results indicate that SimECG-N can closely substitute for real ECG data during pretraining within a small performance tolerance, with SimECG-M also becoming non-inferior under a slightly looser margin ($\delta = 0.05$), whereas VAE/GAN-based pretraining lags behind.

**Transformer (SimECG-N) vs. supervised baselines.** Table 2 compares Transformer (SimECG-N) against supervised architectures. The Transformer (SimECG-N) attains the highest mean $F_1$ on all three datasets with the best average rank in each case. Using the superiority test, we compute per-task $F_1$ differences $d_i = F_{1,i}^{\text{Transformer(SimECG-N)}} - F_{1,i}^{\text{baseline}}$ across the 26 tasks and apply a one-sided Wilcoxon signed-rank test. After Bonferroni correction, all pairwise comparisons yield $p < 0.001$, indicating that Transformer (SimECG-N) is consistently and significantly better than every supervised baseline. Full test statistics are reported in Appendix F.3.2. Further evaluations with non-Transformer architectures are presented in Appendix M, and additional results on PTB-XL benchmark tasks are provided in Appendix J.

**Compute-normalized comparison.** To ensure that the small gaps between pretraining data sources in Tables 1,2 are not artifacts of unequal compute, we additionally track PTB-XL $F_1$ as a function of the number of pretraining samples processed under a shared architecture and hyperparameters (Figure 1; see Appendix K for details). Across budgets from $10^6$ to $10^9$ processed samples, Transformer (PTB-XL/G12EC), and Transformer (SimECG-N) converge to essentially the same downstream performance ($F_1 \approx 0.83$–$0.85$). Transformer (G12EC/PTB-XL) enjoys slightly faster gains at very small budgets ($< 10^7$ samples), but Transformer (SimECG-N) with $\geq 200k$ traces catches up around $10^8$ samples, after which all curves plateau. This compute-normalized view shows that simulator-based pretraining can match real-data pretraining under equal computational budgets, and that the modest average advantage of Transformer (PTB-XL) in Table 1 reflects data realism rather than privileged compute.

## 6.2 DATA EFFICIENCY UNDER SUBSAMPLING

Following the subsampling protocol in Section 5.3.2, we vary the number of labeled examples per class on PTB-XL and compare (i) Transformer (SimECG-N) vs. Transformer (PTB-XL/VAE/GAN) and (ii) Transformer (SimECG-N) with supervised baselines under the same hyperparameters selected in full-data runs.

**Transformer (SimECG-M/N) vs. Transformer (PTB-XL/VAE/GAN).** Figure 2 contrasts simulator and real-data pretraining directly. Overall trends are comparable, with small gaps for ABQRS, AF, and LVH when $n$ is extremely small. Concrete examples (from the same evaluation protocol) include ISC with only 50 positives/50 negatives (Transformer (SimECG-N) average $F_1 = 0.8633$) and PAC with 100/100 ($F_1 = 0.6488$), both on par with or exceeding baselines trained on full data, underscoring the practical value of pretraining when labels are expensive.

**Transformer (SimECG-N) vs. supervised baselines.** Figure 3 (Appendix H) shows that the Transformer (SimECG) maintains higher performance than baselines across most low-data regimes. For example, with $n = 100$ samples per class, it achieves mean $F_1 = 0.7777$ across 13 tasks compared to the best baseline's $F_1 = 0.6971$ ($+11.6\%$ improvement). A notable exception is ABQRS at very small $n$ ($\leq 5$), where supervised baselines outperform the Transformer (SimECG-N).

## 6.3 DEMOGRAPHIC ROBUSTNESS

We report normalized $F_1$ gaps across sex and age (lower is better) averaged over 13 tasks (Table 24; Appendix I). Both pretrained Transformers (PTB-XL and SimECG-N) substantially reduce gaps compared to supervised baselines (e.g., age gap drops from 0.3168–0.4567 for baselines to 0.1898–0.1940 for pretrained models; sex gap from 0.0767–0.1023 to 0.0422–0.0466). Notably, the Transformer (SimECG-N) yields the lowest *age* gap (0.1898), while the Transformer (PTB-XL) yields the lowest *sex* gap (0.0422), suggesting complementary fairness properties.

## 6.4 ROBUSTNESS ACROSS ECG LEADS.

Beyond aggregate performance, the near-uniform average rank of the Transformer (SimECG-N) (1.00–1.31 across I/II/III/aVF/aVL/aVR/V1–V6) indicates that the learned representation is largely invariant to the chosen lead. It is also robust to device or electrode placement differences likely to arise in practice. This property, together with the data-efficiency gains above, supports deployment in heterogeneous acquisition environments (e.g., single-lead wearables vs. 12-lead clinical systems).

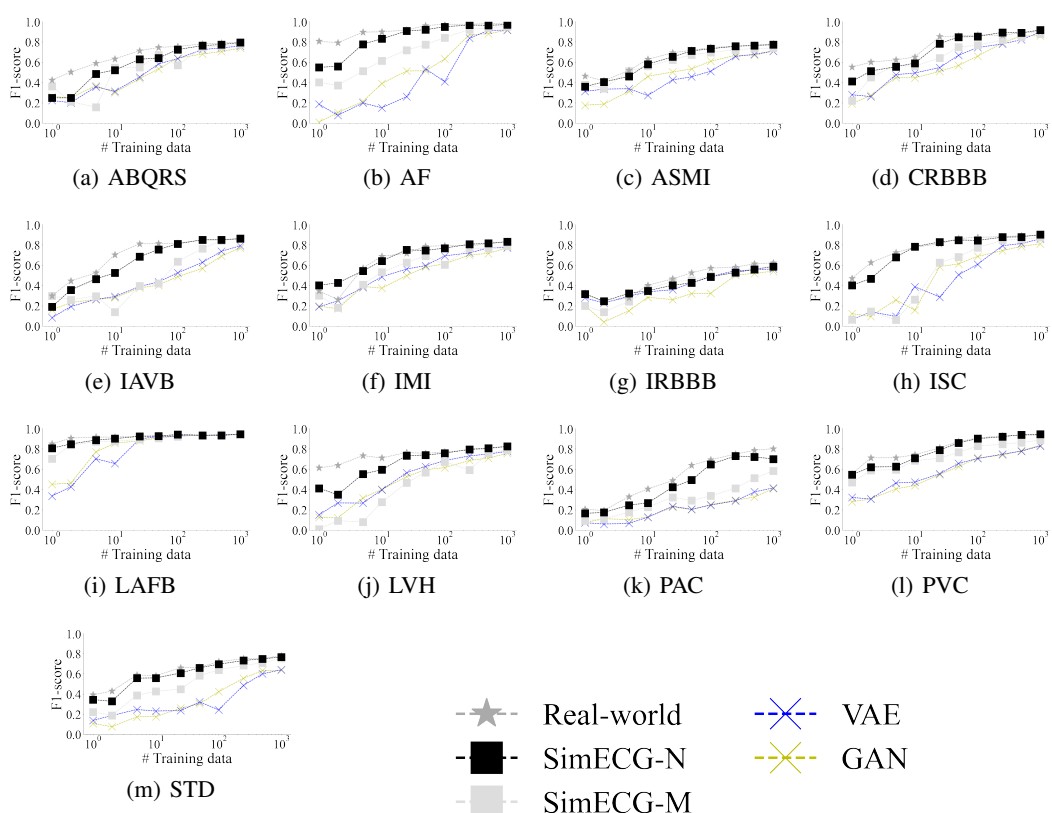

Figure 2: The abnormal ECG classification performance with a reduced number of training samples, under different pretraining data.

# 7 DISCUSSION

This study evaluated whether pretraining on knowledge-driven, simulator-synthesized ECG can substitute for large real-world corpora in abnormal-ECG classification. Across 26 tasks and three public datasets, Transformer (SimECG-N) achieved *near-parity* with Transformer (PTB-XL) while consistently outperforming supervised baselines (Tables 1 and 2). It generalized well across single-lead configurations (Table 3) and reduced demographic performance gaps to levels comparable to real-data pretraining (Table 24). Taken together, our findings suggest that simulator-based corpora capture most of the ECG structure that current downstream classifiers exploit, while avoiding many of the privacy and governance burdens of real-data pretraining. Moreover, additional experiments on phonocardiogram data indicate that this benefit extends beyond ECG-rich settings (Appendix N).

**Low-resource regimes.** Under training-set subsampling on PTB-XL, Transformer (SimECG-N) exhibited smaller performance degradation than supervised baselines across most reduced-data regimes (Figure 3). In realistic small-$n$ cases (e.g., ISC with 50 positives/50 negatives and PAC with 100/100), the Transformer (SimECG-N) matched or exceeded baselines trained on the full dataset (average $F_1 = 0.8633$ for ISC; $F_1 = 0.6488$ for PAC), underscoring practical value when labels are expensive. At the extreme few-shot end ($n < 10$), real-data pretraining retained a modest edge (Figure 2), suggesting that tiny downstream supervision may be insufficient to fully align synthetic pretraining to dataset-specific artifacts.

**Computational equivalence.** As shown in the compute-normalized analysis (Figure 1; Appendix K), all pretraining conditions with SimECG $\geq 200k$ converge to $F_1 \approx 0.83$–$0.85$, essentially the same PTB-XL performance when given the same number of processed samples. This indicates that the small average advantage of Transformer (PTB-XL) over Transformer (SimECG-N) in Ta-

Table 3: Average rankings (lower is better) for classification on the PTB-XL dataset for different leads. **Bold** indicates the best performance. Per-lead detailed $F_1$ tables are provided in Appendix O. †V5 excludes ASMI/ISC/STD due to dataset issues noted in Appendix O.

| Model | Pretraining | I | II | III | AVF | AVL | AVR |
|---|---|---|---|---|---|---|---|
| GRU | - | 3.15 | 2.93 | 3.46 | 3.23 | 3.54 | 2.69 |
| LSTM | - | 3.54 | 4.08 | 3.54 | 3.77 | 3.54 | 3.69 |
| LUNA | - | 6.31 | 6.54 | 6.38 | 6.15 | 6.38 | 6.46 |
| MEGA | - | 4.54 | 5.08 | 5.69 | 5.31 | 5.77 | 5.38 |
| ResNet-18 | - | 3.62 | 3.46 | 2.77 | 2.77 | 2.31 | 3.31 |
| Transformer | - | 5.54 | 4.92 | 5.08 | 5.08 | 5.23 | 5.31 |
| Transformer | SimECG-N | **1.31** | **1.00** | **1.08** | **1.23** | **1.23** | **1.23** |

| Model | Pretraining | V1 | V2 | V3 | V4 | V5† | V6 |
|---|---|---|---|---|---|---|---|
| GRU | - | 3.38 | 3.15 | 3.54 | 3.46 | 3.30 | 3.23 |
| LSTM | - | 3.08 | 4.00 | 3.62 | 3.31 | 2.80 | 3.23 |
| LUNA | - | 6.69 | 6.23 | 6.23 | 6.38 | 6.90 | 6.15 |
| MEGA | - | 4.85 | 5.69 | 5.69 | 5.46 | 4.90 | 5.23 |
| ResNet-18 | - | 3.15 | 3.23 | 3.23 | 2.92 | 3.90 | 3.85 |
| Transformer | - | 5.62 | 4.62 | 4.62 | 5.23 | 5.00 | 4.92 |
| Transformer | SimECG-N | **1.23** | **1.00** | **1.08** | **1.23** | **1.20** | **1.15** |

ble1 stems from data realism rather than privileged compute, and that sufficiently large synthetic corpora can be competitive under equal computational budgets.

**What the synthetic pretraining learns.** The frozen-encoder analysis (Appendix L) further clarifies the nature of learned representations: with all parameters updated, Transformer (PTB-XL) vs. Transformer (SimECG-N) are nearly tied on PTB-XL (mean $F_1$: 0.8488 vs. 0.8451; Table 1), whereas with a linear probe on frozen features the gap widens (0.6972 vs. 0.6380). This implies that Transformer (SimECG-N) captures broadly useful morphology and rhythm regularities, but requires moderate downstream adaptation to accommodate real-data nuisances such as device-specific noise, baseline wander, and labeling conventions. Once given that adaptation budget, most of the remaining performance benefits of real-data pretraining can be recovered by synthetic pretraining.

**Fairness and robustness.** Both real- and synthetic-pretrained models substantially reduced normalized $F_1$ gaps across sex and age on PTB-XL (Table 24). The synthetic-pretrained model achieved the lowest age gap, while Transformer (PTB-XL) achieved the lowest sex gap, indicating complementary fairness profiles. Lead-wise analysis (Table 3) shows synthetic pretraining promotes representations largely invariant to electrode placement, supporting deployment from single-lead wearables to 12-lead clinical systems.

**Implications.** Because synthetic pretraining does not consume patient-identifiable data, it can be scaled without institutional review, data-sharing agreements, or cross-site harmonization. Our results show that such scaling can deliver strong starting points for downstream ECG tasks, reducing labeled-data requirements, improving robustness, and narrowing demographic gaps—benefits that directly address chronic bottlenecks in clinical ML workflows.

**Limitations.** This study has several limitations. First, we only evaluate ECG and PCG, which we regard as intermediate-scale settings for SSL; our results should be interpreted as evidence that simulator-based synthetic pretraining is promising in limited-scale physiological time-series domains, rather than proof of seamless generalization to all data-scarce settings. Second, a domain gap remains between simulator outputs and real ECGs, and real-data pretraining still performs better in some settings. Third, our main experiments focus on single-lead models and one SSL objective; other architectures or objectives may change the relative value of synthetic versus real data, and the fidelity/coverage threshold for synthetic data to rival real-data pretraining is unknown.

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

## LLM Usage Disclosure

We used an LLM "GPT-5, OpenAI" only for linguistic editing (grammar, wording, and fluency) of text written by the authors. No ideas, analyses, or results were produced by the LLM. All suggestions were reviewed and accepted or rejected by the authors. No private or proprietary data were shared with the LLM. The authors bear full responsibility for the final content.

## A    Computational Resources

In this section, we provide a concise overview of the computational resources utilized in this study. All experiments were conducted on a system equipped with four NVIDIA A5000 GPUs, 64 AMD EPYC CPU cores, and 503 GiB of RAM. The synthesis of one million ECGs required approximately 1 hour. The MAE-Transformer model pretraining process spanned six days on a single NVIDIA A5000 GPU. Training of the classification models, performed on a single NVIDIA A5000 GPU, ranged from 5 minutes to 1.5 hours per model, contingent upon early stopping criteria and model complexity. Additional technical specifications, including versions of Python libraries employed, are available in our GitHub repository.

## B    Details of Real-world Data

Table 4: Number of samples in each subset used in this work.

| Subset | Description | PTB-XL | G12EC | CPSC |
|--------|-------------|--------|-------|------|
| ABQRS | Abnormal QRS | 3,327 | — | — |
| AF | Atrial fibrillation | 1,514 | 568 | 1,221 |
| ASMI | Anteroseptal myocardial infarction | 1,528 | — | — |
| CRBBB | Complete right bundle branch block | 537 | — | — |
| IAVB | First degree AV block | 783 | 766 | 772 |
| IMI | Inferior myocardial infarction | 1,025 | — | — |
| IRBBB | Incomplete right bundle branch block | 1,074 | 402 | — |
| ISC | Ischemic | 1,122 | — | — |
| LAFB | Left anterior fascicular block | 1,583 | — | — |
| LVH | Left ventricular hypertrophy | 1,089 | 1,225 | — |
| PAC | Atrial premature complex | 398 | 636 | 616 |
| PVC | Ventricular premature complex | 1,030 | 355 | 700 |
| RBBB | Right bundle branch block | — | 540 | 1857 |
| STD | ST depression | 1009 | — | 869 |
| NORM | Normal ECG | 7,185 | 1,735 | 918 |

This appendix details the three real-world ECG datasets used in our study, the label harmonization across them, and the exact preprocessing choices applied prior to model training. Throughout, we focus on single–lead II waveforms to enable a uniform pipeline across sources.

### B.1    PTB-XL

We use PTB-XL as a primary source of real-world ECGs for both pretraining (`PTB-XL_All`) and downstream classification. Unless otherwise noted, we operate on lead II signals at 500 Hz and 10 s duration (consistent with the pretraining protocol described elsewhere). From PTB-XL we construct one-vs-normal binary classification tasks for 13 abnormal labels together with a `NORM` class (*Normal ECG*). The abnormal labels are: `ABQRS` (Abnormal QRS), `AF` (Atrial fibrillation), `ASMI` (Anteroseptal myocardial infarction), `CRBBB` (Complete right bundle branch block), `IAVB` (First-degree AV block), `IMI` (Inferior myocardial infarction), `IRBBB` (Incomplete right bundle branch block), `ISC` (Ischemic), `LAFB` (Left anterior fascicular block), `LVH` (Left ventricular hypertrophy), `PAC` (Atrial premature complex), `PVC` (Ventricular premature complex), and `STD` (ST depression). Per-label sample counts are summarized in Table 4.

## B.2 G12EC

In addition to PTB-XL, we use the Georgia 12-Lead ECG Challenge Database (G12EC)[8] (Goldberger et al., 2000). The dataset contains 12-lead ECGs collected in the US (10,344 samples). Each record is sampled at 500 Hz with a length of 10 s and is annotated with one or more of 64 labels. In this work, we extract lead II and use samples annotated with AF, IAVB, IRBBB, LVH, PAC, RBBB, and VPB (treated as PVC in our unified label set) as abnormal classes, and NSR (*Normal Sinus Rhythm*) as the normal class. For pretraining condition examinations, we also prepare two convenience subsets: G12EC-NORM (lead II from normal sinus records; note that G12EC-NORM is equivalent to NSR used in our binary tasks) and G12EC-All (lead II from all available samples).[9] Per-task counts for G12EC are listed in Table 4.

## B.3 CPSC

As a third real-world source, we use the CPSC2018 dataset (Liu et al., 2018), which consists of 12-lead ECGs collected in intensive care units of multiple hospitals in China. Signals are sampled at 500 Hz, and the maximum record length is 60 s. Following our single-lead protocol, we extract lead II and use the following labels in our abnormal-vs-normal tasks: AF, IAVB, PAC, PVC, RBBB, and STD, together with Normal as the negative class. Corresponding sample counts appear in Table 4.

## B.4 LABEL SELECTION AND SUBSET DEFINITIONS

Across datasets, we unify diagnosis labels into the following set of 14 abnormal classes plus a normal class: ABQRS, AF, ASMI, CRBBB, IAVB, IMI, IRBBB, ISC, LAFB, LVH, PAC, PVC (including VPB from G12EC), RBBB, and STD, with NORM/NSR/Normal used as the negative class depending on the source dataset. For each abnormal label $\ell$, we construct a one-vs-normal binary classification task using all recordings tagged with $\ell$ as positives and normal-sinus recordings as negatives. We further define source-specific pretraining pools such as PTB-XL-All, G12EC-All, and G12EC-NORM as described above. Because the source datasets are multi-label, a single ECG may contribute to multiple abnormal subsets; counts across rows are therefore non-exclusive.

## B.5 LEAD EXTRACTION, SAMPLING, AND PREPROCESSING

**Lead selection and rationale.** Unless otherwise noted, all signals are processed as single-lead II time series. Lead II was selected because: (1) it provides the most reliable P-QRS-T morphology for abnormality detection, (2) both simulators (McSharry et al., 2003; Nonaka & Seita, 2024) are calibrated to replicate lead II characteristics, (3) it is the standard lead for rhythm analysis in clinical ECG interpretation, and (4) limiting to a single lead reduces computational overhead while maintaining clinical utility. To verify that our findings generalize across electrode placements, we additionally evaluate all 12 standard leads in Appendix O and 12-lead classification using the same pretrained encoders in Appendix P.

**Signal specifications and preprocessing.** G12EC and PTB-XL recordings are 10 s at 500 Hz; CPSC recordings are at 500 Hz and up to 60 s. Unless otherwise stated, we apply per-sample standardization (subtract mean and divide by standard deviation) before model input. For supervised baselines, we use random temporal shifting and random masking as data augmentations (augmentation rates tuned by hyperparameter search; details in Appendix E). Pretraining-specific settings (e.g., masking ratios for MAE) follow Appendix D.

## B.6 SPLITS AND SAMPLE COUNTS (TABLE 4)

Table 4 reports the number of available recordings per abnormal label and per dataset after lead II extraction; the NORM/NSR/Normal row gives the size of the corresponding normal pool used as negatives. A dash in the table indicates that the label is not used or not present in that dataset. We

---

[8]https://www.kaggle.com/datasets/bjoernjostein/georgia-12lead-ecg-challenge-database

[9]G12EC-NORM for pretraining setting examination and NSR used for abnormal ECG classification are equivalent.

use fixed train/validation splits within each task and keep these splits identical across all models to ensure fair comparisons; the table summarizes the total pool sizes underlying those splits.

## C  DETAILS OF SIMULATOR PARAMETERIZATION

This section provides detailed specifications of the two ECG simulators used in our study: the McSharry (McSharry et al., 2003) model and the Nonaka model (Nonaka & Seita, 2024). For each simulator, we describe the parameterization scheme, including which physiological parameters are varied, their sampling distributions, and the parameter values used in our experiments.

### C.1  MCSHARRY MODEL PARAMETERIZATION

The McSharry model generates synthetic ECG waveforms through a dynamical systems approach with parameters controlling heart rate variability and morphological characteristics. We parameterize the simulator following the implementation provided in the PhysioNet ECGSYN toolbox.[10]

**Heart Rate Parameters:**  The mean heart rate h is sampled from a normal distribution $\mathcal{N}(80, 10^2)$ bpm, covering the typical resting heart rate range for healthy adults (60-100 bpm). The heart rate standard deviation H is sampled from $\mathcal{N}(0, 1^2)$ bpm with a lower bound of 0, then shifted by $+1$ to ensure $H \geq 1$. These parameters control the baseline cardiac rhythm and its beat-to-beat variability, reflecting normal sinus rhythm variations.

**Morphological Parameters:**  We introduce amplitude noise a sampled from $\mathcal{N}(1, 1^2)$ mV with a lower bound of 0, which modulates the overall ECG amplitude. This variation captures physiological factors such as electrode positioning, body habitus, and inter-individual differences in cardiac electrical axis. The number of heart beats $n$ is determined dynamically as $n = \lfloor h/60 \times \text{duration} \rfloor + 5$, where duration is set to 12 seconds, with a buffer of 5 beats to ensure sufficient signal length.

**Frequency Domain Parameters:**  To simulate physiological heart rate variability, we incorporate spectral components in the low-frequency (LF) and high-frequency (HF) bands. The LF center frequency f and its standard deviation v are sampled uniformly as $f \sim \mathcal{U}(0,1)/5$ Hz and $v \sim \mathcal{U}(0,1)/25$ Hz. The typical LF band (0.04-0.15 Hz) reflects baroreceptor-mediated blood pressure regulation and thermoregulation. Similarly, the HF center frequency $F$ and its standard deviation $V$ are sampled as $F \sim \mathcal{U}(0,1)/2$ Hz and $V \sim \mathcal{U}(0,1)/25$ Hz. The HF band (0.15-0.4 Hz) corresponds to respiratory sinus arrhythmia, driven by parasympathetic vagal activity. The LF/HF ratio $q$ is sampled as $q \sim \mathcal{U}(0,1)/1$, representing the balance between sympathetic and parasympathetic nervous system activity, which varies with physiological states such as rest, stress, and exercise.

**Sampling Parameters:**  Both the ECG sampling frequency $s$ and internal sampling frequency $S$ are fixed at 500 Hz, which exceeds the Nyquist criterion for capturing high-frequency ECG components and meets clinical ECG recording standards.

This parameterization scheme allows the simulator to generate diverse synthetic ECG waveforms spanning a physiologically plausible range of heart rates, morphologies, and autonomic modulation patterns.

### C.2  NONAKA MODEL PARAMETERIZATION

The Nonaka model generates synthetic ECG waveforms by composing Gaussian-shaped peaks representing the P wave, QRS complex (Q, R, S waves), and T wave. Each waveform is characterized by peak amplitude, temporal position (shift), and width parameters, with noise components added to simulate realistic signal characteristics. We followed the parametarization used in Nonaka & Seita (2024).

**Waveform Generation Framework:**  Each ECG beat is constructed by summing five Gaussian peaks defined as peak $\times \exp(-0.5 \times ((t - \text{shift})/\text{width})^2)$, where Q and S waves are inverted

---

[10]https://www.physionet.org/content/ecgsyn/1.0.0/

(multiplied by -1). The model employs a two-level perturbation scheme: base-level perturbations capture inter-individual variability, while beat-level perturbations simulate beat-to-beat variations within a single recording.

**P Wave Parameters:** The P wave amplitude has a base value of 0.3 with inter-individual perturbation $\mathcal{N}(0, 0.2^2)$ and intra-individual perturbation $\mathcal{N}(0, 0.1^2)$. The P wave width (0.03) serves as the standard deviation of the Gaussian function with inter-individual perturbation $\mathcal{N}(0, 0.02^2)$ and intra-individual perturbation $\mathcal{N}(0, 0.005^2)$, covering typical atrial depolarization durations of 60-120 ms.

**QRS Complex Parameters:** The Q, R, and S wave amplitudes have base values of 0.1, 1.5, and 0.5 respectively, with inter-individual perturbations $\mathcal{N}(0, 0.15^2)$, $\mathcal{N}(0, 0.75^2)$, and $\mathcal{N}(0, 0.7^2)$, and intra-individual perturbations $\mathcal{N}(0, 0.016^2)$, $\mathcal{N}(0, 0.25^2)$, and $\mathcal{N}(0, 0.05^2)$, covering typical QRS complex length. The corresponding widths and temporal positions are defined with similar hierarchical perturbations to cover physiologically plausible QRS morphology.

**T Wave Parameters:** The T wave amplitude (t_peak = 0.45) has inter-individual perturbation $\mathcal{N}(0, 0.5^2)$ and intra-individual perturbation $\mathcal{N}(0, 0.07^2)$. The T wave width (t_width = 0.03) has inter-individual perturbation $\mathcal{N}(0, 0.03^2)$ and intra-individual perturbation $\mathcal{N}(0, 0.0075^2)$, representing ventricular repolarization.

Inter-individual perturbations were intentionally set with large standard deviations to maximize diversity across synthetic samples, accepting that some generated waveforms may be physiologically atypical. Intra-individual perturbations were constrained to smaller ranges to maintain beat-to-beat consistency within individual recordings. This design enables SimECG to generate diverse synthetic data while grounded in physiological knowledge.

# D    PRELIMINARY EXPERIMENTS: ARCHITECTURE SEARCH AND BASELINE SELECTION

This appendix consolidates the settings and outcomes of our preliminary studies conducted prior to the main evaluations. We ran (i) a systematic architecture search for self-supervised pretraining with a Transformer encoder (MAE), and (ii) a supervised baseline model sweep. Unless otherwise noted, the model-selection metric is the validation $F_1$ on a binary atrial fibrillation (AF) vs. normal sinus rhythm (NSR) task.

## D.1    COMMON TRAINING AND EVALUATION PROTOCOL

**Evaluation task.** To compare representation quality under different pretraining configurations, we fix a downstream task: AF vs. NSR classification on **G12EC**, using lead II ECG. During evaluation, the Transformer backbone is *frozen* and only a terminal linear classifier is trained (*linear probing*); model selection is based on validation performance.

**Pretraining budget and data.** For fair comparison across conditions in the architecture search, each MAE configuration is pretrained for up to $\sim 100$ million sample exposures on **PTB-XL-All** (lead II, 10 s at 500 Hz), with a single fixed train/validation split. We monitor the pretraining validation signal and carry forward the best checkpoint to downstream evaluation.

**Preprocessing and optimization (unless specified).** We apply per-sample standardization (subtract mean, divide by standard deviation). Optimization uses Adam with learning rate $10^{-3}$ and batch size 512. For MAE, the token masking ratio is 0.75. For the linear probe, only the final layer is trained on G12EC; early stopping is triggered by the validation metric.

## D.2    SELF-SUPERVISED TRANSFORMER ARCHITECTURE SEARCH (MAE)

We determine the MAE-Transformer configuration in three stages: (1) token **mask size**, (2) encoder **depth**, and (3) **embedding** dimension and **#heads**. For every candidate, we pretrain on PTB-XL-All up to the shared exposure budget and then run a linear probe on G12EC AF vs. NSR.

Table 5: Comparison of the mask size.

| Mask size | $F_1$-score | AUROC | AUPRC |
|---|---|---|---|
| 25 | 0.921 | 0.990 | 0.966 |
| 50 | **0.958** | 0.998 | 0.985 |
| 100 | 0.943 | 0.998 | 0.990 |
| 125 | 0.956 | 0.998 | 0.988 |

**Mask size.** We sweep mask sizes $\{25, 50, 100, 125\}$ tokens. At 500 Hz sampling, 50 tokens correspond to $\sim 0.1$ s. With depth = 4, embedding = 256, and heads = 8 fixed, Table 5 shows that a mask size of 50 yields the highest $F_1$ (0.958), and we adopt it in subsequent steps.

Table 6: Comparison of depth.

| Depth | $F_1$-score | AUROC | AUPRC |
|---|---|---|---|
| 2 | 0.851 | 0.970 | 0.921 |
| 4 | 0.958 | 0.998 | 0.985 |
| 6 | 0.963 | 0.998 | 0.989 |
| 8 | 0.964 | 0.998 | 0.983 |
| 10 | **0.973** | 0.998 | 0.988 |
| 12 | 0.969 | 0.997 | 0.986 |

**Depth.** With mask size fixed to 50, we evaluate depths $\{2, 4, 6, 8, 10, 12\}$. As summarized in Table 6, depth 10 achieves the best $F_1$ (0.973). Deeper models do not improve linear-probe performance under the fixed budget.

**Embedding dimension and heads.** We grid-search embedding $\{32, 64, 128, 256, 512\}$ and heads $\{2, 4, 8, 16, 32\}$ (25 combinations). Table 7 indicates the best $F_1$ (0.977) at embedding = 128 with 8 heads. Larger embeddings or head counts did not translate into better linear-probe scores under the same pretraining budget.

**Chosen configuration.** The MAE-Transformer used in the main experiments therefore uses: mask size = 50 tokens, depth = 10, embedding = 128, heads = 8, masking ratio = 0.75, Adam ($10^{-3}$) and batch size 512. Unless noted, all other training details follow the common protocol above.

D.3 BASELINE CLASSIFICATION MODEL SELECTION

To contextualize our self-supervised results, we benchmarked 16 supervised architectures spanning CNNs, RNNs, and Transformers on a PTB-XL (lead II) AF-vs-NSR task. For EfficientNet and ResNets, we replaced 2D convolutions/pooling with 1D counterparts following Nonaka & Seita (2021a). We use per-sample standardization, random temporal shifting and random masking (rates tuned), batch size 512, Adam (learning rate tuned), validation every 5 epochs, and early stopping on validation loss. Class imbalance is addressed by weighting the positive class by the ratio of negatives to positives in the training split. For non-CNNs, a linear embedding projects the raw 1D input before the backbone.

Hyperparameters are selected with Optuna (Akiba et al., 2019): for each architecture we run a six-hour search (per-trial max 250 epochs) on a fixed train/validation split and then retrain with the best configuration; details of search ranges are provided in Appendix E. Validation results are summarized in Table 8. The top five by $F_1$ are **LSTM**, **GRU**, **MEGA**, **ResNet18**, and **Luna**, which serve as supervised baselines throughout the main experiments.

Table 7: Comparison of embedding dimension and number of heads.

| Emb | #heads | $F_1$-score | AUROC | AUPRC |
|---|---|---|---|---|
| | 2 | 0.880 | 0.979 | 0.948 |
| | 4 | 0.854 | 0.972 | 0.917 |
| 32 | 8 | 0.859 | 0.975 | 0.932 |
| | 16 | 0.817 | 0.963 | 0.907 |
| | 32 | 0.836 | 0.964 | 0.911 |
| | 2 | 0.969 | 0.998 | 0.993 |
| | 4 | 0.957 | 0.997 | 0.986 |
| 64 | 8 | 0.927 | 0.985 | 0.961 |
| | 16 | 0.918 | 0.987 | 0.963 |
| | 32 | 0.840 | 0.966 | 0.918 |
| | 2 | 0.975 | 0.998 | 0.995 |
| | 4 | 0.965 | 0.998 | 0.991 |
| 128 | 8 | **0.977** | 0.999 | 0.993 |
| | 16 | 0.947 | 0.996 | 0.977 |
| | 32 | 0.954 | 0.998 | 0.989 |
| | 2 | 0.971 | 0.999 | 0.993 |
| | 4 | 0.965 | 0.997 | 0.985 |
| 256 | 8 | 0.973 | 0.998 | 0.988 |
| | 16 | 0.971 | 0.997 | 0.991 |
| | 32 | 0.971 | 0.998 | 0.986 |
| | 2 | 0.964 | 0.998 | 0.989 |
| | 4 | 0.972 | 0.998 | 0.992 |
| 512 | 8 | 0.959 | 0.999 | 0.992 |
| | 16 | 0.957 | 0.997 | 0.978 |
| | 32 | 0.949 | 0.997 | 0.983 |

Table 8: Performance of AF classification on PTB-XL dataset.

| | $F_1$-score | Rank | AUROC | AUPRC |
|---|---|---|---|---|
| ResNet18 (He et al., 2016) | **0.868** | **4** | 0.970 | 0.928 |
| ResNet34 (He et al., 2016) | 0.857 | 8 | 0.974 | 0.942 |
| ResNet50 (He et al., 2016) | 0.822 | 11 | 0.959 | 0.907 |
| EfficientNet-B0 (Tan & Le, 2019) | 0.843 | 9 | 0.965 | 0.919 |
| Transformer (Vaswani et al., 2017) | 0.860 | 7 | 0.971 | 0.944 |
| FEDformer (Zhou et al., 2022) | 0.834 | 10 | 0.966 | 0.903 |
| Autoformer (Wu et al., 2021) | 0.806 | 14 | 0.956 | 0.864 |
| Informer (Zhou et al., 2021) | 0.813 | 13 | 0.955 | 0.920 |
| Luna (Ma et al., 2021) | **0.868** | **5** | 0.968 | 0.908 |
| Nystromformer (Xiong et al., 2021) | 0.767 | 15 | 0.935 | 0.827 |
| Performer (Choromanski et al., 2020) | 0.820 | 12 | 0.969 | 0.925 |
| Linear-Transformer (Katharopoulos et al., 2020) | 0.761 | 16 | 0.953 | 0.874 |
| LSTM (Hochreiter & Schmidhuber, 1997) | **0.900** | **1** | 0.980 | 0.961 |
| GRU (Chung et al., 2014) | **0.872** | **2** | 0.977 | 0.946 |
| S4 (Gu et al., 2021) | 0.863 | 6 | 0.976 | 0.938 |
| MEGA (Ma et al., 2022) | **0.871** | **3** | 0.969 | 0.932 |

## E  HYPERPARAMETER SEARCH DETAILS

### E.1  GLOBAL SEARCH SPACE AND OPTIMIZATION

We conduct hyperparameter search with Optuna (Akiba et al., 2019). For each *architecture* and each *abnormal-ECG task*, we allocate a fixed **6-hour** wall-clock budget and select the trial with the

lowest validation loss. Each trial is trained for up to **250 epochs** with validation every 5 epochs; early stopping is triggered if the validation loss does not improve for 5 validations. After selecting the best configuration, we *retrain* on the same train/validation split using the common training setup and report the best validation $F_1$ (max epochs 500; early stopping as above).

**Objective, data split, and preprocessing.** The search objective is the validation cross-entropy loss. A single fixed train/validation split is used per task (identical across trials). Per-sample standardization (zero mean, unit variance) is applied. To mitigate class imbalance, the positive class is weighted by the ratio of negatives to positives in the training set.

**Optimization and batch settings.** Unless otherwise specified, we use Adam with a *tuned* learning rate, default $\beta{=}(0.9, 0.999)$, batch size 512, and no learning-rate schedule during search.

**Globally tuned hyperparameters (shared across architectures).**

- Learning rate: log-uniform in $[5{\times}10^{-6}, 5{\times}10^{-3}]$.
- Augmentations (search space):
  - Random masking ratio (of time steps): uniform in $[0.0, 0.50]$.
  - Random temporal shift ratio (relative to input length): uniform in $[0.0, 0.75]$.
- Final classifier width (MLP head): $\{16, 32, 64, 128, 256, 512, 1024\}$.

**Notation.** "Step length" denotes the temporal stride / sub-sampling step for sequence models (candidates given below). At 500 Hz, a step of 50 corresponds to 0.1 s.

E.2 ARCHITECTURE-SPECIFIC RANGES

Below we list the architecture-specific search spaces (in addition to the global hyperparameters above). CNN-based baselines (e.g., ResNets, EfficientNet) use their canonical macro-architectures; only the global hyperparameters are tuned for them.

**GRU / LSTM.**

- Recurrent depth: $\{1, 2, 3, 4\}$
- Hidden size: $\{16, 32, 64, 128, 256\}$
- Step length: $\{50, 100, 250\}$

**S4 (Gu et al., 2021).**

- Depth (number of S4 blocks): $\{3, 4, 5, 6\}$
- Number of heads: $\{16, 32, 64\}$
- Feed-forward width: $\{32, 64, 128, 256\}$
- Step length: $\{50, 100, 250\}$

**MEGA (Ma et al., 2022).**

- Depth: $\{3, 4, 5, 6\}$
- Number of heads: $\{16, 32, 64\}$
- Feed-forward width: $\{32, 64, 128, 256\}$
- Q/K/V size: $\{16, 32, 64\}$
- Step length: $\{50, 100, 250\}$

**Luna (Ma et al., 2021).**

- Depth: $\{3, 4, 5, 6\}$
- Number of heads: $\{16, 32, 64\}$
- Feed-forward width: $\{32, 64, 128, 256\}$
- Q/K/V size: $\{16, 32, 64\}$
- Context length: $\{16, 32, 64\}$
- Step length: $\{50, 100, 250\}$

**Transformer (Vaswani et al., 2017).**

- Depth: $\{3, 4, 5, 6\}$
- Number of heads: $\{16, 32, 64\}$
- Feed-forward width: $\{32, 64, 128, 256\}$
- Q/K/V size: $\{16, 32, 64\}$
- Step length: $\{50, 100, 250\}$

# F  STATISTICAL TESTING

This appendix provides comprehensive details on the statistical testing procedures used to evaluate (1) non-inferiority of simulator-based pretraining compared to real-world pretraining, and (2) superiority of pretrained models over supervised baselines.

## F.1  METHODOLOGY

We conducted rigorous statistical testing using the Wilcoxon signed-rank test, a non-parametric method appropriate for paired observations that does not assume normality.

**Rationale for test selection** : We treat each of the 26 downstream classification tasks as a paired observation, comparing performance across different pretraining conditions or model architectures on the same task. The Wilcoxon signed-rank test is robust to outliers and appropriate for our setting where we cannot assume that the differences in $F_1$ scores follow a normal distribution.

**Multiple comparison correction** : Because we perform multiple pairwise comparisons against the same reference condition, we apply Bonferroni correction to control the family-wise error rate. For $k$ comparisons, we use an adjusted significance threshold of $\alpha/k$, where $\alpha = 0.025$ for one-sided tests.

**Experimental setup** : All models are trained five times with different random seeds on each of the 26 tasks. We report mean $F_1$ scores and use these means as the basis for statistical comparisons.

## F.2  NON-INFERIORITY TESTING: TABLE 1

### F.2.1  TEST FORMULATION

For comparisons between pretraining data sources in Table 1, we use a non-inferiority formulation with the Real-world (PTB-XL) condition as the reference. Let $F_{1,i}^{\text{test}}$ and $F_{1,i}^{\text{real}}$ denote the mean $F_1$ for task $i$ under a test condition and the Real-world condition, respectively. Given a relative non-inferiority margin $\delta$, we form:

$$d_i = F_{1,i}^{\text{test}} - (1 - \delta)F_{1,i}^{\text{real}} \tag{1}$$

We apply a one-sided Wilcoxon signed-rank test across the 26 tasks with:

- $H_0 : \text{median}(d_i) \leq 0$ (the test condition is worse than Real-world by more than $\delta$)
- $H_1 : \text{median}(d_i) > 0$ (the test condition is non-inferior)

Table 9: Non-inferiority test results comparing synthetic data approaches against real-world baseline using Wilcoxon signed-rank test with Bonferroni correction across PTB-XL tasks.

|  | $p$-value | $p$-value (corrected) |
|---|---|---|
| SimECG-M | 0.1767 | 0.7066 |
| SimECG-N | 0.0002 | 0.0008 |
| VAE | 1.0000 | 1.0000 |
| DCGAN | 0.9387 | 1.0000 |

Table 10: Sensitivity test for non-inferiority test of results shown in Table 9 (corrected $p$-value).

|  | $\delta$ | | | |
|---|---|---|---|---|
|  | 0.01 | 0.025 | 0.05 | 0.10 |
| SimECG-M | 1.0000 | 0.7066 | 0.0015 | 0.0000 |
| SimECG-N | 1.0000 | 0.0008 | 0.0000 | 0.0000 |
| VAE | 1.0000 | 1.0000 | 1.0000 | 0.4345 |
| DCGAN | 1.0000 | 1.0000 | 1.0000 | 0.0380 |

**Choice of non-inferiority margin:** We set $\delta = 0.025$ (2.5% relative decrease) based on the observed measurement variability. The median standard deviation across conditions is 1.25%, which corresponds to an expected 95% confidence interval width of approximately 3% for pairwise comparisons. The chosen margin of 2.5% represents a practically negligible performance difference while accounting for experimental noise.

**Multiple comparison correction:** We test four pretraining conditions (SimECG-M/N, VAE, DCGAN) against the same Real-world reference, so we apply Bonferroni correction with adjusted significance level $\alpha = 0.025/4 \approx 0.00625$.

### F.2.2 RESULT

Using a non-inferiority margin of $\delta = 0.025$ (2.5% performance degradation) justified by measurement variability across experimental conditions, we conducted Wilcoxon signed-rank tests with Bonferroni correction to assess whether synthetic data approaches perform comparably to real-world pretraining (Table 9). The results demonstrate that SimECG-N achieves statistical non-inferiority to the real-world baseline, while SimECG-M, VAE, and DCGAN all fail to reject the inferiority hypothesis at the conventional significance level. To evaluate the robustness of these conclusions, we performed a sensitivity analysis across different margin values ranging from 1% to 10% (Table 10). The sensitivity test reveals that the statistical conclusions are sensitive to the choice of margin, with stricter margins ($\delta = 0.01$) failing to establish non-inferiority for any method, while more permissive margins ($\delta = 0.10$) demonstrate non-inferiority for the simulator-based approaches but not for the deep generative models.

**Interpretation:** At the strictest margin ($\delta = 0.01$), no method achieves non-inferiority, suggesting that small but detectable differences exist between synthetic and real-world pretraining. Both simulator-based methods achieve non-inferiority at $\delta = 0.05$ (5% relative decrease), demonstrating robust performance within a practically acceptable tolerance. VAE remains inferior even at $\delta = 0.10$, while DCGAN achieves non-inferiority only at this most permissive margin.

### F.3 SUPERIORITY TESTING: TABLE 2

### F.3.1 TEST FORMULATION

For comparisons between Transformer (SimECG-N) and supervised baselines in Table 2, we test for superiority. For each task $i$ and baseline model, we compute:

$$d_i = F_{1,i}^{\text{Transformer(SimECG)}} - F_{1,i}^{\text{baseline}} \tag{2}$$

Table 11: Non-inferiority test results comparing SimECG-N pre-trained models against baseline models trained from scratch using Wilcoxon signed-rank test with Bonferroni correction across PTB-XL tasks.

| Baseline model | $p$-value | $p$-value (corrected) |
|---|---|---|
| GRU | 0.0000 | 0.0000 |
| LSTM | 0.0000 | 0.0000 |
| LUNA | 0.0000 | 0.0000 |
| MEGA | 0.0000 | 0.0000 |
| ResNet18 | 0.0000 | 0.0000 |
| Transformer (scratch) | 0.0000 | 0.0000 |

We apply a one-sided Wilcoxon signed-rank test with:

- $H_0 : \text{median}(d_i) \leq 0$ (no superiority)
- $H_1 : \text{median}(d_i) > 0$ (the pretrained model is superior)

**Multiple comparison correction:** We test six pairwise comparisons (GRU, LSTM, Luna, MEGA, ResNet-18, Transformer (Scratch)), so we apply Bonferroni correction with adjusted significance level $\alpha = 0.025/6 \approx 0.0042$.

### F.3.2 RESULT

Here, we present detailed $p$-values obtained from paired superiority tests on the per-task $F_1$ scores. We applied a one-sided Wilcoxon signed-rank test to these paired differences across the 26 tasks, as this test is appropriate for paired data and does not require the assumption of normality. To verify that SimECG pretraining provides meaningful performance improvements over training from scratch, we conducted superiority tests comparing SimECG-N pre-trained models against various baseline architectures trained without pretraining (Table 11). The results demonstrate that SimECG pretraining achieves statistically significant superior performance compared to all baseline models trained from scratch, with all corrected $p$-values below the significance threshold, confirming that the pretraining approach provides substantial benefits across different model architectures.

**Interpretation:** The uniformly significant results (all $p < 0.001$ after Bonferroni correction) across all baselines indicate that the performance gains from SimECG pretraining are not architecture-specific but represent a fundamental advantage of self-supervised representation learning on synthetic data over purely supervised training from scratch. The consistency across diverse architectures (RNNs, Transformers, CNNs) demonstrates the broad applicability of simulator-based pretraining.

### F.4 DISCUSSION

Our statistical testing framework provides strong evidence for two key claims:

1. **Non-inferiority of simulator-based pretraining:** SimECG-N achieves statistical non-inferiority to real-world PTB-XL pretraining within a $2.5\%$ relative performance margin. This threshold is justified by measurement variability and represents a practically negligible difference in clinical applications.

2. **Superiority over supervised baselines:** Transformer (SimECG) consistently and significantly outperforms all supervised baselines, with uniform statistical significance ($p < 0.001$) across architectures and tasks.

The sensitivity analysis reveals that conclusions are robust across reasonable margin choices ($\delta = 0.025$–$0.10$), while the strictest margin ($\delta = 0.01$) fails to establish non-inferiority for any method, suggesting that small but detectable differences exist between synthetic and real pretraining. However, these differences fall within the noise tolerance of typical clinical validation studies and do not meaningfully impact the practical utility of simulator-based pretraining.

Table 12: $F_1$-scores for classification on the PTB-XL dataset with Transformer pretrained by MAE approach. Transformer (SimECG) shows performance close to the Transformer (PTB-XL). Text colors indicate relative performance: Red denotes the best performance, while blue indicates the second-best performance for each task.

| | Real-world | Simulator | | Deep generative model | |
| | | SimECG-M[*] | SimECG-N[†] | VAE | GAN |
|---|---|---|---|---|---|
| ABQRS | 0.8063 ±0.0063 | 0.7958 ±0.0114 | 0.8023 ±0.0085 | 0.7732 ±0.0079 | 0.7949 ±0.0112 |
| AF | 0.9808 ±0.0030 | 0.9698 ±0.0053 | 0.9753 ±0.0023 | 0.9374 ±0.0023 | 0.9604 ±0.0069 |
| ASMI | 0.8083 ±0.0089 | 0.7803 ±0.0101 | 0.7939 ±0.0070 | 0.7630 ±0.0296 | 0.7758 ±0.0133 |
| CRBBB | 0.9189 ±0.0180 | 0.9171 ±0.0040 | 0.9184 ±0.0066 | 0.9048 ±0.0074 | 0.9051 ±0.0056 |
| IAVB | 0.8875 ±0.0067 | 0.8747 ±0.0151 | 0.8847 ±0.0050 | 0.8278 ±0.0117 | 0.8705 ±0.0123 |
| IMI | 0.8368 ±0.0069 | 0.8282 ±0.0192 | 0.8588 ±0.0119 | 0.8195 ±0.0137 | 0.8281 ±0.0118 |
| IRBBB | 0.5758 ±0.0333 | 0.5480 ±0.0319 | 0.5822 ±0.0200 | 0.5287 ±0.0131 | 0.5400 ±0.0159 |
| ISC | 0.8989 ±0.0120 | 0.8857 ±0.0082 | 0.8947 ±0.0157 | 0.8718 ±0.0205 | 0.8866 ±0.0071 |
| LAFB | 0.9477 ±0.0055 | 0.9534 ±0.0013 | 0.9495 ±0.0051 | 0.9489 ±0.0049 | 0.9489 ±0.0043 |
| LVH | 0.8316 ±0.0123 | 0.8180 ±0.0099 | 0.8312 ±0.0050 | 0.7999 ±0.0165 | 0.8260 ±0.0115 |
| PAC | 0.8070 ±0.0114 | 0.6365 ±0.0490 | 0.7621 ±0.0433 | 0.3172 ±0.0540 | 0.5229 ±0.1171 |
| PVC | 0.9438 ±0.0057 | 0.9091 ±0.0108 | 0.9454 ±0.0056 | 0.8902 ±0.0120 | 0.8963 ±0.0054 |
| STD | 0.7911 ±0.0129 | 0.7723 ±0.0164 | 0.7881 ±0.0156 | 0.7093 ±0.0216 | 0.7891 ±0.0149 |
| Mean | 0.8488 | 0.8222 | 0.8451 | 0.7763 | 0.8111 |
| Relative score | - | −3.1320 | −0.4341 | −8.5441 | −4.4397 |

[*] Synthesized using simulator from McSharry et al. (2003).
[†] Synthesized using simulator from Nonaka & Seita (2024).

**Limitations:** Our statistical tests assume independence across tasks, which may be violated if tasks share similar underlying patterns (e.g., multiple myocardial infarction subtypes). Additionally, the Wilcoxon test examines median differences rather than mean differences, which is conservative but may underestimate effect sizes when distributions are skewed.

# G PERFORMANCE WITH VARIOUS SELF-SUPERVISED TRAINING APPROACHES

We compare three self-supervised learning (SSL) objectives—MAE, SimCLR, and DINO—when pretraining a Transformer on (i) real-world ECGs, (ii) simulator–synthesized ECGs (SimECG-M (McSharry et al., 2003) and SimECG-N (Nonaka & Seita, 2024)), and (iii) deep generative ECGs (VAE/GAN). Unless otherwise noted, we report mean $F_1$ (with standard deviations in the tables) on the downstream abnormal-ECG classification tasks for **PTB-XL**, **G12EC**, and **CPSC**.

## G.1 MAE (PTB-XL / G12EC / CPSC)

Across datasets, MAE is the most stable objective and simulator pretraining is consistently competitive with Transformer (PTB-XL) (Tables 12–14). On **PTB-XL**, Transformer (SimECG-N) (0.8451)

Table 13: $F_1$-scores for classification on the G12EC dataset with Transformer pretrained by MAE approach. Transformer (SimECG) shows performance close to the Transformer (PTB-XL). Text colors indicate relative performance: Red denotes the best performance, while blue indicates the second-best performance for each task.

| | Real-world | Simulator | | Deep generative model | |
| | | SimECG-M* | SimECG-N† | VAE | GAN |
|---|---|---|---|---|---|
| AF | 0.9185 ±0.0101 | 0.8987 ±0.0114 | 0.9022 ±0.0239 | 0.8110 ±0.0225 | 0.8602 ±0.0309 |
| IAVB | 0.9390 ±0.0044 | 0.9219 ±0.0213 | 0.9125 ±0.0334 | 0.8912 ±0.0075 | 0.9425 ±0.0129 |
| IRBBB | 0.7833 ±0.0156 | 0.6965 ±0.0276 | 0.7680 ±0.0284 | 0.5665 ±0.0372 | 0.6717 ±0.0262 |
| LVH | 0.7620 ±0.0167 | 0.7440 ±0.0142 | 0.7304 ±0.0473 | 0.6440 ±0.0207 | 0.6977 ±0.0246 |
| PAC | 0.9190 ±0.0078 | 0.8917 ±0.0166 | 0.8945 ±0.0141 | 0.5578 ±0.0690 | 0.7200 ±0.0261 |
| PVC | 0.8768 ±0.0190 | 0.8428 ±0.0233 | 0.8385 ±0.0162 | 0.7699 ±0.0155 | 0.7523 ±0.0266 |
| RBBB | 0.9020 ±0.0168 | 0.8619 ±0.0230 | 0.8975 ±0.0155 | 0.7829 ±0.0383 | 0.8118 ±0.0231 |
| Mean | 0.8715 | 0.8368 | 0.8491 | 0.7176 | 0.7795 |
| Relative score | - | −3.9849 | −2.5735 | −17.6589 | −10.5629 |

\* Synthesized using simulator from McSharry et al. (2003).
† Synthesized using simulator from Nonaka & Seita (2024).

Table 14: $F_1$-scores for classification on the CPSC dataset with Transformer pretrained by MAE approach. Transformer (SimECG) shows performance close to the Transformer (PTB-XL). Text colors indicate relative performance: Red denotes the best performance, while blue indicates the second-best performance for each task.

| | Real-world | Simulator | | Deep generative model | |
| | | SimECG-M* | SimECG-N† | VAE | GAN |
|---|---|---|---|---|---|
| AF | 0.9889 ±0.0021 | 0.9761 ±0.0037 | 0.9782 ±0.0038 | 0.9572 ±0.0042 | 0.9679 ±0.0074 |
| IAVB | 0.9563 ±0.0057 | 0.9471 ±0.0123 | 0.9589 ±0.0053 | 0.9085 ±0.0299 | 0.9460 ±0.0051 |
| PAC | 0.9038 ±0.0503 | 0.8733 ±0.0239 | 0.9391 ±0.0124 | 0.6835 ±0.0337 | 0.7311 ±0.0332 |
| PVC | 0.9437 ±0.0067 | 0.9218 ±0.0126 | 0.9141 ±0.0245 | 0.8464 ±0.0128 | 0.8514 ±0.0393 |
| RBBB | 0.9198 ±0.0052 | 0.9159 ±0.0065 | 0.9243 ±0.0059 | 0.9042 ±0.0042 | 0.9063 ±0.0112 |
| STD | 0.8959 ±0.0030 | 0.9014 ±0.0037 | 0.9045 ±0.0073 | 0.8995 ±0.0098 | 0.8922 ±0.0068 |
| Mean | 0.9347 | 0.9226 | 0.9365 | 0.8666 | 0.8825 |
| Relative score | - | −1.2981 | 0.1908 | −7.2944 | −5.5898 |

\* Synthesized using simulator from McSharry et al. (2003).
† Synthesized using simulator from Nonaka & Seita (2024).

nearly matches Transformer (PTB-XL) (0.8488; −0.43% relative), and exceeds VAE/GAN baselines (0.7763/0.8111). Transformer (SimECG-N) attains the best per-class $F_1$ on several tasks (e.g., IMI, PVC, LAFB) while remaining close on AF and STD. On **G12EC**, Transformer (PTB-XL) is best on average (0.8715), with Transformer (SimECG-N) second (0.8491; −2.57% relative). The gaps are small for AF, IRBBB, and RBBB (e.g., RBBB: 0.9020 vs. 0.8975), and both Transformer

Table 15: $F_1$-scores for classification on the PTB-XL dataset with Transformer pretrained by Sim-CLR approach. Transformer (SimECG) shows performance close to the Transformer (PTB-XL). Text colors indicate relative performance: Red denotes the best performance, while blue indicates the second-best performance for each task.

| | Real-world | Simulator | Deep generative model | |
| --- | --- | --- | --- | --- |
| | | | VAE | GAN |
| ABQRS | 0.7599 ±0.0049 | 0.7580 ±0.0136 | 0.7572 ±0.0098 | 0.7546 ±0.0043 |
| AF | 0.9236 ±0.0095 | 0.9136 ±0.0133 | 0.9179 ±0.0108 | 0.9238 ±0.0156 |
| ASMI | 0.7282 ±0.0084 | 0.7091 ±0.0200 | 0.6542 ±0.0382 | 0.7123 ±0.0194 |
| CRBBB | 0.8693 ±0.0128 | 0.8606 ±0.0259 | 0.8635 ±0.0172 | 0.8642 ±0.0194 |
| IAVB | 0.7174 ±0.0241 | 0.7054 ±0.0434 | 0.7167 ±0.0169 | 0.7223 ±0.0199 |
| IMI | 0.7698 ±0.0236 | 0.7798 ±0.0224 | 0.7673 ±0.0086 | 0.7807 ±0.0211 |
| IRBBB | 0.4581 ±0.0393 | 0.4848 ±0.0171 | 0.4627 ±0.0339 | 0.4469 ±0.0392 |
| ISC | 0.8291 ±0.0269 | 0.8102 ±0.0243 | 0.8240 ±0.0173 | 0.8430 ±0.0240 |
| LAFB | 0.9455 ±0.0045 | 0.9377 ±0.0070 | 0.9456 ±0.0021 | 0.9402 ±0.0053 |
| LVH | 0.7598 ±0.0212 | 0.7832 ±0.0153 | 0.7598 ±0.0303 | 0.7721 ±0.0159 |
| PAC | 0.3022 ±0.0549 | 0.2665 ±0.0387 | 0.2994 ±0.0595 | 0.3438 ±0.0310 |
| PVC | 0.8377 ±0.0157 | 0.8192 ±0.0318 | 0.8370 ±0.0103 | 0.8598 ±0.0144 |
| STD | 0.7143 ±0.0203 | 0.6727 ±0.0343 | 0.6884 ±0.0217 | 0.6903 ±0.0196 |
| Mean | 0.7396 | 0.7308 | 0.7303 | 0.7426 |
| Relative score | - | −1.1867 | −1.2605 | 0.4067 |

(SimECG-M) and Transformer (SimECG-N) clearly outperform VAE/GAN. On **CPSC**, Transformer (SimECG-N) yields the highest mean (0.9365), slightly surpassing Transformer (PTB-XL) (0.9347; +0.19% relative). Transformer (SimECG-N) is best on four of six tasks (IAVB, PAC, RBBB, STD), while real-world remains best on AF and PVC.

G.2 SimCLR (PTB-XL / G12EC / CPSC)

SimCLR exhibits higher sensitivity to the pretraining data source (Tables 15–17). Across **PTB-XL**, **G12EC**, and **CPSC**, the **GAN**-pretrained model attains the highest mean (0.7426 / 0.6966 / 0.8561), slightly outperforming real-world (0.7396 / 0.6898 / 0.8462). The Transformer (SimECG-N) remains competitive on **PTB-XL** (0.7308; −1.19% vs. real) and **CPSC** (0.8247; −2.54%), and even achieves the best AF on CPSC (0.9463 vs. 0.9462). However, it underperforms on **G12EC** (0.6275; −9.04% vs. real). Despite this variability, Transformer (SimECG-N) still secures strong second-place finishes in several tasks (e.g., PTB-XL IRBBB/LVH; G12EC AF), and typically tracks or exceeds VAE.

G.3 DINO (PTB-XL / G12EC / CPSC)

DINO sits between MAE and SimCLR in robustness (Tables 18–20). On **PTB-XL**, Transformer (PTB-XL) is best (0.6210); Transformer (SimECG-N) is close (0.6073; −2.20%) and outperforms

Table 16: $F_1$-scores for classification on the G12EC dataset with Transformer pretrained by Sim-CLR approach. Simulator generated data pretrained model shows performance close to the Transformer (PTB-XL). Text colors indicate relative performance: Red denotes the best performance, while blue indicates the second-best performance for each task.

| | Real-world | SimECG-N | Deep generative model | |
| | | | VAE | GAN |
|---|---|---|---|---|
| AF | 0.8041 ±0.0437 | 0.7469 ±0.0354 | 0.7802 ±0.0310 | 0.7902 ±0.0251 |
| IAVB | 0.7815 ±0.0390 | 0.7852 ±0.0313 | 0.7896 ±0.0210 | 0.7892 ±0.0643 |
| IRBBB | 0.6463 ±0.0556 | 0.5049 ±0.0951 | 0.6512 ±0.0413 | 0.6074 ±0.0176 |
| LVH | 0.6441 ±0.0230 | 0.5718 ±0.0618 | 0.6451 ±0.0231 | 0.6268 ±0.0303 |
| PAC | 0.4892 ±0.0634 | 0.4912 ±0.0639 | 0.5120 ±0.0228 | 0.5869 ±0.0472 |
| PVC | 0.7249 ±0.0564 | 0.5609 ±0.1088 | 0.7184 ±0.0246 | 0.7065 ±0.0527 |
| RBBB | 0.7387 ±0.0284 | 0.7314 ±0.0396 | 0.7111 ±0.0404 | 0.7690 ±0.0276 |
| Mean | 0.6898 | 0.6275 | 0.6868 | 0.6966 |
| Relative score | - | −9.0395 | −0.4390 | 0.9775 |

Table 17: $F_1$-scores for classification on the CPSC dataset with Transformer pretrained by SimCLR approach. Transformer (SimECG) shows performance close to the Transformer (PTB-XL). Text colors indicate relative performance: Red denotes the best performance, while blue indicates the second-best performance for each task.

| | Real-world | SimECG-N | Deep generative model | |
| | | | VAE | GAN |
|---|---|---|---|---|
| AF | 0.9462 ±0.0075 | 0.9463 ±0.0142 | 0.9443 ±0.0079 | 0.9387 ±0.0074 |
| IAVB | 0.8894 ±0.0195 | 0.8855 ±0.0216 | 0.8670 ±0.0268 | 0.8688 ±0.0491 |
| PAC | 0.6382 ±0.0361 | 0.5238 ±0.2624 | 0.5858 ±0.0896 | 0.6763 ±0.0162 |
| PVC | 0.8536 ±0.0296 | 0.8479 ±0.0263 | 0.8284 ±0.0205 | 0.8725 ±0.0234 |
| RBBB | 0.8855 ±0.0145 | 0.8812 ±0.0111 | 0.8782 ±0.0184 | 0.8994 ±0.0097 |
| STD | 0.8641 ±0.0183 | 0.8636 ±0.0098 | 0.8663 ±0.0254 | 0.8809 ±0.0138 |
| Mean | 0.8462 | 0.8247 | 0.8283 | 0.8561 |
| Relative score | - | −2.5350 | −2.1075 | 1.1739 |

VAE/GAN on average, while also winning on specific tasks (AF, IAVB, IRBBB, ISC). On **G12EC**, Transformer (SimECG-N) achieves the highest mean (0.5580), outperforming real-world (0.5402; +3.29%) and both VAE/GAN; it is best on IAVB, IRBBB, LVH, and PAC, and second on AF and PVC. On **CPSC**, Transformer (PTB-XL) again leads (0.7769), with GAN (0.7700) and Transformer (SimECG-N) (0.7664) close behind; notably, Transformer (SimECG-N) is best on PAC and STD and second on AF/PVC.

**Overall.** MAE offers the strongest and most consistent benefits from simulator pretraining; DINO is competitive with pockets of simulator superiority; SimCLR is more sensitive to the generator,

Table 18: $F_1$-scores for classification on the PTB-XL dataset with Transformer pretrained by DINO approach. Transformerv(SimECG) shows performance slightly decreased performance compare to the Transformer (PTB-XL). Text colors indicate relative performance: Red denotes the best performance, while blue indicates the second-best performance for each task.

| | Real-world | SimECG-N | Deep generative model | |
| | | | VAE | GAN |
|---|---|---|---|---|
| ABQRS | 0.7036 ±0.0103 | 0.6933 ±0.0085 | 0.6993 ±0.0190 | 0.7005 ±0.0103 |
| AF | 0.7001 ±0.0085 | 0.7158 ±0.0210 | 0.6869 ±0.0361 | 0.7109 ±0.0239 |
| ASMI | 0.6430 ±0.0107 | 0.6108 ±0.0110 | 0.5944 ±0.0214 | 0.6313 ±0.0245 |
| CRBBB | 0.7332 ±0.0130 | 0.7116 ±0.0254 | 0.7191 ±0.0089 | 0.7623 ±0.0077 |
| IAVB | 0.5569 ±0.0210 | 0.5721 ±0.0066 | 0.5261 ±0.0215 | 0.4946 ±0.0229 |
| IMI | 0.6429 ±0.0149 | 0.6234 ±0.0093 | 0.6367 ±0.0239 | 0.6148 ±0.0390 |
| IRBBB | 0.3877 ±0.0324 | 0.3982 ±0.0268 | 0.3503 ±0.0453 | 0.3697 ±0.0180 |
| ISC | 0.6712 ±0.0107 | 0.6747 ±0.0231 | 0.6710 ±0.0157 | 0.6556 ±0.0146 |
| LAFB | 0.9258 ±0.0040 | 0.9215 ±0.0050 | 0.9214 ±0.0028 | 0.9247 ±0.0063 |
| LVH | 0.6744 ±0.0192 | 0.6424 ±0.0359 | 0.6574 ±0.0214 | 0.6452 ±0.0315 |
| PAC | 0.2131 ±0.0080 | 0.1887 ±0.0144 | 0.1665 ±0.0224 | 0.1940 ±0.0265 |
| PVC | 0.7387 ±0.0101 | 0.7309 ±0.0100 | 0.7300 ±0.0155 | 0.7325 ±0.0083 |
| STD | 0.4827 ±0.0334 | 0.4121 ±0.0488 | 0.4818 ±0.0351 | 0.4572 ±0.0396 |
| Mean | 0.6210 | 0.6073 | 0.6031 | 0.6072 |
| Relative score | - | −2.2023 | −2.8786 | −2.2296 |

with GAN occasionally strongest. Across objectives and datasets, simulator pretraining (especially SimECG-N) is a reliable privacy-enhancing alternative to Transformer (PTB-XL), typically outperforming VAE and matching or surpassing GAN outside of SimCLR.

## G.4 ADDITIONAL ANALYSIS OF PRETRAINING OBJECTIVES AND SYNTHETIC ECG SOURCES

Here, we summarize higher-level observations from our comparison of simulator-based synthetic ECGs (SimECG), deep generative ECGs (VAE/DCGAN), and real PTB-XL ECGs as pretraining corpora for three self-supervised objectives (MAE, SimCLR, DINO). Our goal is to provide a conceptual synthesis beyond the per-task tables.

### G.4.1 DATA-GENERATION MECHANISMS.

SimECG is knowledge driven: the simulators of McSharry et al. (2003) and Nonaka & Seita (2024) explicitly parameterize cardiac morphology and rhythm, and we use them to generate clean, physiologically plausible 10 s, 500 Hz signals whose P–QRS–T structure closely matches real ECG but with relatively simple acquisition noise. By contrast, our VAE and DCGAN baselines are data driven and trained on PTB-XL lead II. A basic VAE with an isotropic Gaussian prior (Kingma & Welling, 2013) tends to produce smooth, "averaged" morphologies and underestimates high-frequency details and rare patterns, even though it inherits typical PTB-XL noise characteristics. A DCGAN-style

Table 19: $F_1$-scores for classification on the G12EC dataset with Transformer pretrained by DINO approach. Transformer (SimECG) shows performance slightly decreased performance compare to the Transformer (PTB-XL). Text colors indicate relative performance: Red denotes the best performance, while blue indicates the second-best performance for each task.

| | Real-world | SimECG-N | Deep generative model | |
| | | | VAE | GAN |
|---|---|---|---|---|
| AF | 0.4977 ±0.0480 | 0.5341 ±0.0237 | 0.5320 ±0.0146 | 0.5418 ±0.0594 |
| IAVB | 0.5400 ±0.0109 | 0.5974 ±0.0195 | 0.5788 ±0.0225 | 0.5850 ±0.0127 |
| IRBBB | 0.5124 ±0.0339 | 0.5383 ±0.0317 | 0.4795 ±0.0230 | 0.4731 ±0.0289 |
| LVH | 0.5571 ±0.0224 | 0.6035 ±0.0253 | 0.5670 ±0.0407 | 0.6000 ±0.0228 |
| PAC | 0.3821 ±0.0308 | 0.3848 ±0.0304 | 0.3830 ±0.0163 | 0.3034 ±0.0229 |
| PVC | 0.6007 ±0.0236 | 0.5783 ±0.0325 | 0.5290 ±0.0200 | 0.5497 ±0.0193 |
| RBBB | 0.6917 ±0.0122 | 0.6699 ±0.0245 | 0.6696 ±0.0143 | 0.6642 ±0.0194 |
| Mean | 0.5402 | 0.5580 | 0.5341 | 0.5310 |
| Relative score | - | 3.2948 | −1.1318 | −1.7056 |

Table 20: $F_1$-scores for classification on the CPSC dataset with Transformer pretrained by DINO approach. Transformer (SimECG) shows performance slightly decreased performance compare to the Transformer (PTB-XL). Text colors indicate relative performance: Red denotes the best performance, while blue indicates the second-best performance for each task.

| | Real-world | SimECG-N | Deep generative model | |
| | | | VAE | GAN |
|---|---|---|---|---|
| AF | 0.8656 ±0.0227 | 0.8400 ±0.0117 | 0.8242 ±0.0238 | 0.8217 ±0.0238 |
| IAVB | 0.7436 ±0.0198 | 0.7074 ±0.0389 | 0.5895 ±0.0694 | 0.7472 ±0.0119 |
| PAC | 0.5681 ±0.0374 | 0.5891 ±0.0216 | 0.5850 ±0.0265 | 0.5874 ±0.0242 |
| PVC | 0.7994 ±0.0125 | 0.7912 ±0.0261 | 0.7523 ±0.0297 | 0.7881 ±0.0192 |
| RBBB | 0.8634 ±0.0095 | 0.8464 ±0.0102 | 0.8524 ±0.0141 | 0.8562 ±0.0120 |
| STD | 0.8214 ±0.0113 | 0.8246 ±0.0045 | 0.8114 ±0.0172 | 0.8196 ±0.0090 |
| Mean | 0.7769 | 0.7664 | 0.7358 | 0.7700 |
| Relative score | - | −1.3472 | −5.2923 | −0.8860 |

generator (Radford et al., 2015) often better reproduces local sharpness and realistic noise and artifacts, but its diversity is limited by the training corpus and it can be prone to mode dropping or memorization. In short, SimECG emphasizes coverage of physiologic variation under idealized noise; VAE emphasizes smooth interpolation with reduced extremes; GAN emphasizes realistic measurement noise and texture on top of the PTB-XL distribution.

Overall, these three families span a spectrum:

- SimECG: high morphological control, low acquisition noise (idealized but flexible physiology).

- VAE: good coverage, but smoothed morphology and attenuated fine detail.

- GAN: sharp, noisy, realistic-looking samples, but with possible mode bias.

### G.4.2 SELF-SUPERVISED OBJECTIVES

The three SSL objectives stress different aspects of the signal. SimCLR (Chen et al., 2020) is contrastive and learns invariances to augmentations; with strong augmentations it particularly rewards robustness to local perturbations and nuisance factors that resemble real measurement noise. DINO (Caron et al., 2021) matches student–teacher predictions under augmentations and is known to produce clustered, semantically structured features; in time series this encourages representations that capture recurring morphology patterns, while still depending on the distribution of augmented "views." MAE (He et al., 2022) is reconstruction based: masking a large fraction of tokens forces the encoder to model fine-grained local structure and global rhythm/morphology in order to reconstruct missing segments, and is comparatively tolerant to moderate mismatches in noise statistics because the decoder can absorb simple noise components.

### G.4.3 INTERACTIONS AND EMPIRICAL TRENDS

Across PTB-XL, G12EC, and CPSC, simulator-based pretraining consistently outperforms VAE/GAN pretraining for MAE and is competitive with or close to real-data pretraining, while VAE is weakest overall and GAN lies in between. With MAE, SimECG-2 nearly matches PTB-XL pretraining on PTB-XL and G12EC and slightly surpasses it on CPSC; VAE lags substantially, and GAN shows intermediate performance (Tables 12-14). For SimCLR, the ranking shifts: GAN-based pretraining often attains the best mean $F_1$, with SimECG close on PTB-XL and CPSC but less competitive on G12EC (Table 15-17). For DINO, SimECG-2 is slightly below PTB-XL on PTB-XL/CPSC but outperforms real-data pretraining on G12EC (Table 18-20). A qualitative interpretation consistent with these trends is:

- MAE benefits most from diverse, coherent morphology and rhythm, where SimECG excels; the lack of complex acquisition artifacts can be corrected during fine-tuning on real data.

- SimCLR benefits more from realistic nuisance variation; GAN captures PTB-XL-like noise and artifacts that align well with SimCLR augmentations, whereas SimECG traces are relatively clean and yield invariances that transfer slightly less well in some settings.

- DINO behaves in between MAE and SimCLR: it is somewhat sensitive to view distributions but also leverages cluster structure over morphology, so simulator-based pretraining remains competitive and can even surpass real-data pretraining on some datasets (e.g., G12EC).

### G.4.4 SCOPE OF THE COMPARISON

Our VAE and DCGAN are deliberately basic baselines rather than state-of-the-art ECG generators, so these results should not be read as a general statement that simulators always outperform modern generative models. Instead, they show that (i) knowledge-driven simulators can provide stronger MAE pretraining signals than simple VAE/GAN baselines trained on PTB-XL, and (ii) the relative advantages of simulators vs. learned generators are objective dependent, reflecting both what each generator reproduces best (physiology vs. noise vs. extremes) and which aspects each SSL objective encourages the encoder to retain or discard.

## H DETAILED RESULTS VS. SUPERVISED BASELINES

This section contrasts the MAE-pretrained **Transformer** (SimECG-N; see Appendix G) with five strong supervised baselines (GRU, LSTM, Luna, MEGA, ResNet18; search details in Appendix E). Unless noted otherwise, we report validation $F_1$ (mean $\pm$ std) on one-vs-normal binary tasks, using identical preprocessing, augmentations, and early stopping across models (Appendix D.3).

## H.1 Classification with Full Data

### H.1.1 PTB-XL

Table 21: $F_1$-scores for classification on the PTB-XL dataset (P.T. = pretraining / Rel. score = Relative score). Text colors indicate relative performance: Red denotes the best performance, while blue indicates the second-best performance for each task.

| | GRU | LSTM | Luna | MEGA | ResNet18 | Transformer | |
|---|---|---|---|---|---|---|---|
| P.T. | - | - | - | - | - | - | SimECG-N |
| ABQRS | 0.7958 ±0.0046 | 0.7931 ±0.0096 | 0.7741 ±0.0055 | 0.7821 ±0.0081 | 0.7923 ±0.0101 | 0.7955 ±0.0080 | 0.8023 ±0.0085 |
| AF | 0.9492 ±0.0136 | 0.9571 ±0.0084 | 0.9119 ±0.0149 | 0.9384 ±0.0050 | 0.9430 ±0.0193 | 0.9279 ±0.0042 | 0.9753 ±0.0023 |
| ASMI | 0.7512 ±0.0244 | 0.7305 ±0.0197 | 0.7221 ±0.0120 | 0.7147 ±0.0206 | 0.7544 ±0.0251 | 0.7338 ±0.0115 | 0.7939 ±0.0070 |
| CRBBB | 0.8019 ±0.0733 | 0.7783 ±0.0493 | 0.7595 ±0.0320 | 0.7158 ±0.0839 | 0.8067 ±0.0922 | 0.7273 ±0.0728 | 0.9184 ±0.0066 |
| IAVB | 0.7977 ±0.0348 | 0.7827 ±0.0411 | 0.7552 ±0.0409 | 0.7867 ±0.0275 | 0.7910 ±0.0155 | 0.7728 ±0.0195 | 0.8847 ±0.0050 |
| IMI | 0.7540 ±0.0129 | 0.7262 ±0.0322 | 0.7417 ±0.0237 | 0.7417 ±0.0392 | 0.7701 ±0.0284 | 0.7314 ±0.0169 | 0.8588 ±0.0119 |
| IRBBB | 0.5471 ±0.0194 | 0.5374 ±0.0161 | 0.5176 ±0.0232 | 0.5218 ±0.0204 | 0.5669 ±0.0240 | 0.5415 ±0.0078 | 0.5822 ±0.0200 |
| ISC | 0.8395 ±0.0359 | 0.8635 ±0.0133 | 0.8124 ±0.0096 | 0.8334 ±0.0112 | 0.8545 ±0.0324 | 0.8530 ±0.0073 | 0.8947 ±0.0157 |
| LAFB | 0.9264 ±0.0100 | 0.9245 ±0.0066 | 0.9223 ±0.0068 | 0.9271 ±0.0053 | 0.9224 ±0.0120 | 0.9288 ±0.0067 | 0.9495 ±0.0051 |
| LVH | 0.7762 ±0.0301 | 0.7994 ±0.0162 | 0.7353 ±0.0312 | 0.7471 ±0.0234 | 0.7845 ±0.0411 | 0.7554 ±0.0139 | 0.8312 ±0.0050 |
| PAC | 0.6547 ±0.0564 | 0.5792 ±0.0972 | 0.2951 ±0.0374 | 0.6257 ±0.0675 | 0.3606 ±0.0213 | 0.3385 ±0.0259 | 0.7621 ±0.0433 |
| PVC | 0.9212 ±0.0129 | 0.9101 ±0.0146 | 0.8035 ±0.0194 | 0.8699 ±0.0267 | 0.8838 ±0.0209 | 0.8275 ±0.0414 | 0.9454 ±0.0056 |
| STD | 0.6957 ±0.0167 | 0.6618 ±0.0285 | 0.6456 ±0.0275 | 0.6764 ±0.0257 | 0.6719 ±0.0229 | 0.6585 ±0.0227 | 0.7881 ±0.0156 |
| Mean | 0.7854 | 0.7726 | 0.7228 | 0.7601 | 0.7617 | 0.7378 | 0.8451 |
| Rel. score | - | −1.6336 | −7.9750 | −3.2300 | −3.0214 | −6.0594 | 7.5999 |

Table 21 lists 13 one-vs-normal tasks on PTB-XL (lead II). The **Transformer (SimECG-N)** attains the highest mean $F_1$ (**0.8451**), outperforming the best supervised baseline (GRU, 0.7854) by **+0.0597** absolute (**+7.6%** relative). It achieves the best score on *all 13* tasks, with especially large margins on: *CRBBB* (+0.1117 over the best baseline), *PAC* (+0.1074), *IMI* (+0.0887), *IAVB* (+0.0870), and *STD* (+0.0924). Gains are also consistent on easier tasks such as *PVC* (+0.0242) and *LAFB* (+0.0224). Relative to an Transformer (Scratch) (0.7378), Transformer (SimECG-N) yields a **+0.1073** absolute improvement (**+14.6%** relative), indicating that most of the lift comes from representation learning rather than architecture alone.

### H.1.2 G12EC

On G12EC (lead II; Table 22), the **Transformer (SimECG-N)** reaches the highest mean $F_1$ (**0.8491**), exceeding the best supervised average (GRU, 0.8136) by **+0.0355** absolute (**+4.37%** relative). It is the top performer in *6/7* tasks, with notable gains on *AF* (+0.0547 vs. ResNet18), *IRBBB* (+0.0550 vs. ResNet18), *PVC* (+0.0377 vs. GRU), and *RBBB* (+0.0306 vs. ResNet18); *LVH* shows a smaller but positive gain (+0.0059 vs. GRU). The only exception is *IAVB*, where ResNet18 slightly leads (0.9375 vs. 0.9125). Compared to Transformer (Scratch) (0.7526), Transformer (SimECG-N) improves mean $F_1$ by **+0.0965** (**+12.8%** relative).

Table 22: $F_1$-scores for classification on the G12EC dataset (P.T. = pretraining / Rel. score = Relative score). Text colors indicate relative performance: Red denotes the best performance, while blue indicates the second-best performance for each task.

| | GRU | LSTM | Luna | MEGA | ResNet18 | Transformer | |
| --- | --- | --- | --- | --- | --- | --- | --- |
| P.T. | - | - | - | - | - | - | SimECG-N |
| AF | 0.8395 ±0.0187 | 0.7818 ±0.0414 | 0.7918 ±0.0204 | 0.8433 ±0.0226 | 0.8475 ±0.0048 | 0.8248 ±0.0113 | 0.9022 ±0.0239 |
| IAVB | 0.9317 ±0.0096 | 0.9113 ±0.0142 | 0.8926 ±0.0173 | 0.9059 ±0.0169 | 0.9375 ±0.0060 | 0.9054 ±0.0054 | 0.9125 ±0.0334 |
| IRBBB | 0.6996 ±0.0185 | 0.6712 ±0.0165 | 0.6425 ±0.0192 | 0.5811 ±0.0252 | 0.7130 ±0.0201 | 0.6435 ±0.0246 | 0.7680 ±0.0284 |
| LVH | 0.7245 ±0.0126 | 0.7208 ±0.0164 | 0.6778 ±0.0500 | 0.7176 ±0.0097 | 0.4234 ±0.3470 | 0.7180 ±0.0130 | 0.7304 ±0.0473 |
| PAC | 0.8814 ±0.0111 | 0.8870 ±0.0060 | 0.6226 ±0.1317 | 0.7623 ±0.0572 | 0.6884 ±0.0505 | 0.6626 ±0.0278 | 0.8945 ±0.0141 |
| PVC | 0.8008 ±0.0416 | 0.7260 ±0.0337 | 0.6946 ±0.0219 | 0.7096 ±0.0223 | 0.6896 ±0.0534 | 0.7056 ±0.0280 | 0.8385 ±0.0162 |
| RBBB | 0.8175 ±0.0278 | 0.7856 ±0.0081 | 0.7693 ±0.0775 | 0.8117 ±0.0234 | 0.8669 ±0.0123 | 0.8084 ±0.0121 | 0.8975 ±0.0155 |
| Mean | 0.8136 | 0.7834 | 0.7273 | 0.7616 | 0.7380 | 0.7526 | 0.8491 |
| Rel. score | - | −3.7103 | −10.6023 | −6.3828 | −9.2836 | −7.4925 | 4.3652 |

### H.1.3 CPSC

Table 23: $F_1$-scores for classification on the CPSC dataset (P.T. = pretraining / Rel. score = Relative score). Text colors indicate relative performance: Red denotes the best performance, while blue indicates the second-best performance for each task.

| | GRU | LSTM | Luna | MEGA | ResNet18 | Transformer | |
| --- | --- | --- | --- | --- | --- | --- | --- |
| P.T. | - | - | - | - | - | - | SimECG-N |
| AF | 0.9750 ±0.0032 | 0.9747 ±0.0027 | 0.9624 ±0.0088 | 0.9725 ±0.0097 | 0.9676 ±0.0041 | 0.9648 ±0.0066 | 0.9782 ±0.0038 |
| IAVB | 0.9536 ±0.0084 | 0.9559 ±0.0080 | 0.6887 ±0.3469 | 0.8916 ±0.1164 | 0.9370 ±0.0172 | 0.9420 ±0.0088 | 0.9589 ±0.0053 |
| PAC | 0.9135 ±0.0155 | 0.8986 ±0.0128 | 0.6770 ±0.0407 | 0.7229 ±0.0111 | 0.7217 ±0.0154 | 0.7508 ±0.0214 | 0.9391 ±0.0124 |
| PVC | 0.9255 ±0.0287 | 0.9358 ±0.0150 | 0.8430 ±0.0145 | 0.8171 ±0.0420 | 0.8801 ±0.0127 | 0.8543 ±0.0156 | 0.9141 ±0.0245 |
| RBBB | 0.8839 ±0.0097 | 0.8900 ±0.0084 | 0.8467 ±0.0518 | 0.8979 ±0.0089 | 0.9101 ±0.0126 | 0.8928 ±0.0136 | 0.9243 ±0.0059 |
| STD | 0.8964 ±0.0040 | 0.8969 ±0.0047 | 0.8875 ±0.0103 | 0.9026 ±0.0061 | 0.8975 ±0.0065 | 0.8903 ±0.0107 | 0.9045 ±0.0073 |
| Mean | 0.9246 | 0.9253 | 0.8176 | 0.8674 | 0.8857 | 0.8825 | 0.9365 |
| Rel. score | - | 0.0721 | −11.5828 | −6.1879 | −4.2160 | −4.5585 | 1.2834 |

On CPSC (lead II; Table 23), the **Transformer (SimECG)** achieves the best mean $F_1$ (**0.9365**), surpassing the strongest supervised baseline average (LSTM, 0.9253) by **+0.0112** (**+1.28%** relative). It attains the top score in *5/6* tasks: *AF* (+0.0032 vs. GRU), *IAVB* (+0.0030 vs. LSTM), *PAC* (+0.0256 vs. GRU), *RBBB* (+0.0142 vs. ResNet18), and *STD* (+0.0019 vs. MEGA). The sole task where a supervised baseline remains best is *PVC*, where LSTM slightly outperforms the Transformer (SimECG) (0.9358 vs. 0.9141). Overall, Trasformer (SimECG-N) delivers consistent gains across datasets, with larger improvements on the more challenging PTB-XL tasks and smaller, but still positive, margins on G12EC and CPSC.

## H.2 Classification under Reduced Data

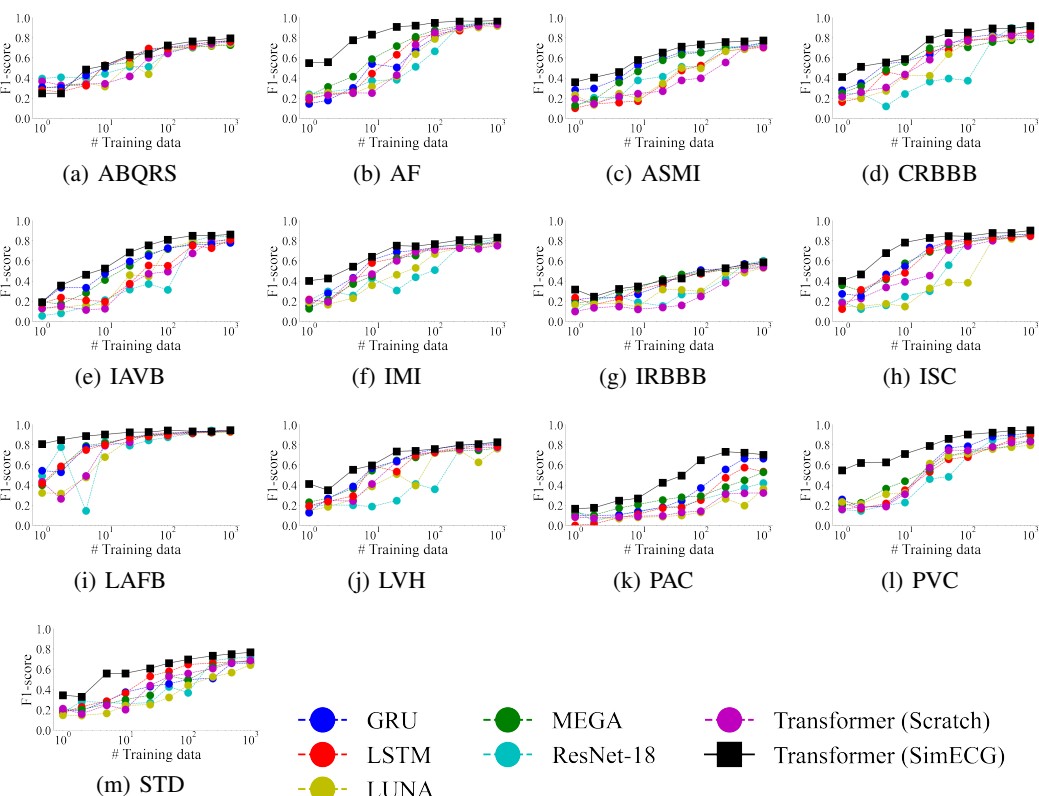

Figure 3: The abnormal ECG classification performance with a reduced number of training samples. Transformer (SimECG-N) vs. supervised baselines. The performance with Transformer (SimECG-N) shows equal or higher classification performance.

We evaluated performance as the number of training instances was reduced, as shown in Figure 3. Across all disease-specific classification tasks, performance decreased as the number of training examples decreased. The degradation was generally smaller for the Transformer (SimECG-N), especially for AF, ISC, LVH, PAC, and PVC, where it achieved notably better performance than the baseline models under severe data reduction. For ABQRS, ASMI, CRBBB, IRBBB, LAFB, STD, and IAVB, the Transformer (SimECG-N) performed comparably to or better than the best baseline. For IMI, GRU and LSTM achieved the best performance when 100 or fewer training instances were available. These results indicate that the Transformer (SimECG-N) has strong potential to reduce the amount of labeled data required for abnormal ECG classification.

## I Details: Demographic Robustness

### I.1 Overall Results

Table 24 summarizes demographic performance gaps across 13 PTB-XL classification tasks, quantified by normalized $F_1$-score differences across age groups and sex. Lower values indicate more equitable performance across demographic subgroups. Both pretrained Transformers substantially reduce demographic disparities compared to supervised baselines. For age-based gaps, the Transformer (SimECG-N) achieves the lowest mean gap (0.1898), narrowly outperforming the Transformer (PTB-XL) (0.1940), while supervised baselines exhibit gaps ranging from 0.3168 to 0.4567. For sex-based gaps, the Transformer (PTB-XL) attains the smallest mean gap (0.0422), with the Transformer (SimECG-N) close behind (0.0466); both are substantially lower than supervised base-

Table 24: PTB-XL (13 tasks): mean normalized gaps (lower is better) and average rank ($\bar{r}$; lower is better) for demographics. The Transformer (SimECG-N) exibits comparable fairness compared to Transformer (PTB-XL) and reduced gaps compared to supervised baseline models. **Bold** indicates the best performance. Full per-task tables in Appendix I.

| Model | Pretraining | Age gap Mean ↓ | Age gap $\bar{r}$ ↓ | Sex gap Mean ↓ | Sex gap $\bar{r}$ ↓ |
|---|---|---|---|---|---|
| GRU | - | 0.3168 | 4.08 | 0.0767 | 4.46 |
| LSTM | - | 0.3496 | 5.39 | 0.0888 | 5.08 |
| LUNA | - | 0.4567 | 7.08 | 0.1021 | 5.08 |
| MEGA | - | 0.3669 | 6.00 | 0.1023 | 6.85 |
| ResNet-18 | - | 0.3759 | 4.54 | 0.0767 | 4.31 |
| Transformer | - | 0.3936 | 4.39 | 0.0985 | 5.00 |
| Transformer | PTB-XL | 0.1940 | **2.23** | **0.0422** | **2.39** |
| Transformer | SimECG-N | **0.1898** | 2.31 | 0.0466 | 2.85 |

lines (0.0767–0.1023), representing approximately $45\%$ and $39\%$ reductions, respectively, compared to the best baseline.

The average rank metric corroborates these findings: both pretrained Transformers consistently rank in the top two positions across tasks for both demographic dimensions, whereas supervised baselines occupy ranks 4–7. Notably, the Transformer (SimECG-N) demonstrates the best age-gap performance while remaining highly competitive on sex-gap metrics, suggesting complementary fairness profiles between real-data and simulator-based pretraining that could be leveraged through ensemble methods or multi-corpus pretraining strategies.

These results indicate that self-supervised pretraining—whether on real or synthetic ECG data—yields representations that generalize more equitably across patient demographics than supervised learning from scratch, with simulator-based pretraining delivering fairness improvements comparable to real-data pretraining without requiring access to patient records during the pretraining phase.

### I.2 AGE-WISE ANALYSIS

Table 25: Age-based $F_1$-score differences demonstrating reduced bias in both Transformer (SimECG) and Transformer (PTB-XL) (lower score means less performance gap among age groups). Text colors indicate relative performance: Red denotes the best performance, while blue indicates the second-best performance for each task.

| | GRU | LSTM | Luna | MEGA | ResNet18 | | Transformer PTB-XL | SimECG-N |
|---|---|---|---|---|---|---|---|---|
| P.T. | - | - | - | - | - | - | PTB-XL | SimECG-N |
| ABQRS | 0.5558 | 0.5900 | 0.6517 | 0.5893 | 0.5893 | 0.5162 | 0.5299 | 0.5375 |
| AF | 0.1919 | 0.1733 | 0.2875 | 0.2520 | 0.1592 | 0.2321 | 0.1014 | 0.1140 |
| ASMI | 0.4837 | 0.5815 | 0.5941 | 0.6526 | 0.4879 | 0.5610 | 0.2322 | 0.2803 |
| CRBBB | 0.0988 | 0.1385 | 0.1557 | 0.1681 | 0.0742 | 0.1612 | 0.0166 | 0.0211 |
| IAVB | 0.5007 | 0.5610 | 0.5348 | 0.5400 | 0.5106 | 0.5397 | 0.1287 | 0.0948 |
| IMI | 0.2197 | 0.2114 | 0.2121 | 0.2138 | 0.2075 | 0.2028 | 0.0743 | 0.0782 |
| IRBBB | 0.2633 | 0.3418 | 0.3580 | 0.3667 | 0.3430 | 0.3168 | 0.5216 | 0.3841 |
| ISC | 0.3087 | 0.2650 | 0.4773 | 0.3350 | 0.2787 | 0.2629 | 0.2324 | 0.2431 |
| LAFB | 0.1602 | 0.1924 | 0.1861 | 0.1509 | 0.1572 | 0.1377 | 0.1748 | 0.1561 |
| LVH | 0.3902 | 0.3959 | 0.5160 | 0.4620 | 0.4082 | 0.4866 | 0.2326 | 0.2350 |
| PAC | 0.2953 | 0.3978 | 1.0548 | 0.2278 | 0.8930 | 0.9346 | 0.0582 | 0.0272 |
| PVC | 0.1495 | 0.1764 | 0.3250 | 0.2973 | 0.2613 | 0.2540 | 0.1143 | 0.1139 |
| STD | 0.4999 | 0.5203 | 0.5836 | 0.5136 | 0.5165 | 0.5109 | 0.1046 | 0.1817 |
| Mean | 0.3168 | 0.3496 | 0.4567 | 0.3669 | 0.3759 | 0.3936 | 0.1940 | 0.1898 |

**Metric.** We quantify age-related robustness as the absolute $F_1$-score gap across age groups within each task (smaller is better). Table 25 summarizes results over the 13 PTB-XL tasks.

**Aggregate effect.** Both Transformers (PTB-XL) and Transformer (SimECG-N) substantially reduce the average age gap compared with strong supervised baselines. The Transformer (SimECG-N) attains the lowest mean gap (0.1898), narrowly outperforming the **Real-world** pretrained counterpart (0.1940).

**Per-task patterns.** Across tasks, at least one of Transformer (PTB-XL/SimECG-N) is among the top two (lowest gap) on **11 of 13** tasks. The **Transformer (PTB-XL)** model yields the smallest gap on seven tasks (e.g., AF: 0.1014, ASMI: 0.2322, CRBBB: 0.0166, IMI: 0.0743, ISC: 0.2324, LVH: 0.2326, STD: 0.1046), while the **Synth** model is best on three tasks (IAVB: 0.0948, PAC: 0.0272, PVC: 0.1139). Two tasks are notable exceptions where a non-pretrained model attains the best gap (ABQRS and LAFB). IRBBB remains comparatively challenging for all models (best: GRU at 0.2633).

I.3 SEX/GENDER-WISE ANALYSIS

Table 26: Gender-based $F_1$-score differences demonstrating reduced bias in both Transformer (SimECG-N) and Transformer (PTB-XL) (lower score means less performance gap between gender). Text colors indicate relative performance: Red denotes the best performance, while blue indicates the second-best performance for each task.

| P.T. | GRU - | LSTM - | Luna - | MEGA - | ResNet18 - | - | Transformer PTB-XL | SimECG-N |
|------|-------|--------|--------|--------|-----------|---|--------|----------|
| ABQRS | 0.0915 | 0.0936 | 0.0897 | 0.1018 | 0.1126 | 0.0958 | 0.0845 | 0.0894 |
| AF | 0.0101 | 0.0072 | 0.0224 | 0.0293 | 0.0075 | 0.0191 | 0.0020 | 0.0134 |
| ASMI | 0.0360 | 0.0669 | 0.0512 | 0.0772 | 0.0269 | 0.0557 | 0.0453 | 0.0407 |
| CRBBB | 0.1839 | 0.2223 | 0.1959 | 0.2820 | 0.1698 | 0.2523 | 0.0819 | 0.1114 |
| IAVB | 0.0807 | 0.0862 | 0.0894 | 0.1045 | 0.0421 | 0.0746 | 0.0189 | 0.0031 |
| IMI | 0.1095 | 0.1515 | 0.1175 | 0.1261 | 0.1085 | 0.1000 | 0.0549 | 0.0520 |
| IRBBB | 0.1835 | 0.1736 | 0.2409 | 0.2547 | 0.0856 | 0.1884 | 0.0460 | 0.0283 |
| ISC | 0.0970 | 0.0840 | 0.0820 | 0.1035 | 0.0867 | 0.0529 | 0.0372 | 0.0643 |
| LAFB | 0.0142 | 0.0136 | 0.0064 | 0.0172 | 0.0220 | 0.0204 | 0.0065 | 0.0112 |
| LVH | 0.0439 | 0.0546 | 0.0271 | 0.0783 | 0.0039 | 0.0053 | 0.0614 | 0.0862 |
| PAC | 0.0978 | 0.1175 | 0.3121 | 0.1068 | 0.2757 | 0.3622 | 0.0578 | 0.0793 |
| PVC | 0.0253 | 0.0129 | 0.0745 | 0.0278 | 0.0162 | 0.0454 | 0.0133 | 0.0182 |
| STD | 0.0233 | 0.0704 | 0.0188 | 0.0201 | 0.0401 | 0.0086 | 0.0384 | 0.0089 |
| Mean | 0.0767 | 0.0888 | 0.1021 | 0.1023 | 0.0767 | 0.0985 | 0.0422 | 0.0466 |

**Metric.** Following the same protocol, we report the absolute $F_1$-score gap between the two sex/gender groups provided in the metadata (smaller is better). Table 26 reports results over the same 13 tasks.

**Aggregate effect.** Pretraining with SimECG-N again reduces disparities. The Transformer (PTB-XL) achieves the lowest mean gap (0.0422), with the Transformer (SimECG-N) close behind (0.0466). Compared with the best baseline mean (0.0767; GRU/ResNet18), these correspond to $\approx 45\%$ (**Transformer(PTB-XL)**) and $\approx 39\%$ (**Trasformer(SimECG)**) reductions (Table 26, "Mean"). Notably, the Transformer (PTB-XL)'s per-task gaps are $\leq 0.10$ on all tasks; the Transformer (SimECG) is $\leq 0.10$ on 12/13 tasks (CRBBB: 0.1114).

**Per-task patterns.** A pretrained Transformer ranks in the top two on **11 of 13** tasks. The Transformer (PTB-XL-N) attains the best (lowest) gap on five tasks (e.g., ABQRS: 0.0845, AF: 0.0020, CRBBB: 0.0819, ISC: 0.0372, PAC: 0.0578), while the Transformer (SimECG-N) leads on three (IAVB: 0.0031, IMI: 0.0520, IRBBB: 0.0283). For ASMI, LAFB, LVH, PVC, and STD, the smallest

gaps are achieved by baselines (e.g., ResNet18 on ASMI/LVH, LSTM on PVC), though Transformer (PTB-XL/SimECG-N) are typically close (often second-best; e.g., LAFB: Real 0.0065).

**Takeaway. Pretraining—on either real or synthesized ECGs—consistently improves sex/gender robustness.** The Transformer (PTB-XL) attains the lowest mean gap overall, while the Transformer (SimECG-N) is frequently competitive or best at the task level, indicating that privacy-enhancing simulator pretraining can deliver *comparable fairness gains*.

## J  PERFORMANCE ON PTB-XL BENCHMARK TASKS

To validate our approach against established benchmarks with competitive supervised baselines, we conducted additional experiments on PTB-XL using only lead II data. All models were trained at 500 Hz to maintain consistency with SimECG's sampling rate, following the standard PTB-XL benchmark protocol for dataset splits and evaluation metrics.

### J.1  EXPERIMENTAL SETUP

We used only lead II data and trained all models at 500 Hz to maintain consistency with SimECG's sampling rate. We evaluated five additional supervised CNN baselines implemented with the fastai library: XResNet1D-101, XResNet1D-152, and XceptionTime. These architectures represent strong 1D CNN models for ECG and time-series classification and are closely related to the CNN families used in prior PTB-XL benchmark studies. All models were trained using the same input representation, data augmentations, and optimization settings as SimECG to ensure fair comparison. The dataset split and evaluation metrics followed the standard PTB-XL benchmark protocol.

We conducted two sets of experiments: (1) training with the full PTB-XL dataset, and (2) training with reduced labeled data at $50\%, 25\%, 10\%, 5\%, 2.5\%, and 1\%$ of the original training set size for each task. For the data reduction experiments, we evaluated two sampling strategies: random sampling and stratified sampling. In random sampling, label distributions were not considered, which may result in certain labels being absent from the training data. In stratified sampling, we preserved all samples for labels with 5 or fewer instances in the training set, and applied stratified sampling based on label information for all other samples to maintain class distribution. These experiments assess the data efficiency of Transformer(SimECG-N) compared to supervised learning from scratch.

### J.2  RESULT

#### J.2.1  CLASSIFICATION WITH FULL DATA

Table 27: Performance comparison on PTB-XL benchmark (Macro-averaged AUC). Text colors indicate relative performance: Red denotes the best performance, while blue indicates the second-best performance for each task.

| Model | Pretraining | all | rhythm | form | diag. | sub-diag. | super-diag. |
|---|---|---|---|---|---|---|---|
| ResNet18 | - | 0.8069 | 0.9232 | 0.7454 | 0.8248 | 0.8327 | 0.8481 |
| GRU | - | 0.8167 | 0.9063 | 0.6331 | 0.7939 | 0.8073 | 0.8392 |
| LSTM | - | 0.8234 | 0.9512 | 0.7678 | 0.7940 | 0.8132 | 0.8474 |
| MEGA | - | 0.8249 | 0.9496 | 0.7776 | 0.8201 | 0.8421 | 0.8512 |
| Luna | - | 0.8000 | 0.8787 | 0.7239 | 0.8053 | 0.7948 | 0.8367 |
| Transformer | - | 0.8062 | 0.8838 | 0.7128 | 0.8095 | 0.8381 | 0.8408 |
| XceptionTime | - | 0.8034 | 0.9044 | 0.6877 | 0.8048 | 0.8029 | 0.8526 |
| XResNet1D-101 | - | 0.8294 | 0.9291 | 0.7577 | 0.8067 | 0.8487 | 0.8495 |
| XResNet1D-152 | - | 0.7901 | 0.9296 | 0.7437 | 0.8125 | 0.8513 | 0.8531 |
| Transformer | PTB-XL | 0.8377 | 0.9612 | 0.7979 | 0.8457 | 0.8428 | 0.8533 |
| Transformer | SimECG-N | 0.8280 | 0.9568 | 0.7838 | 0.8262 | 0.8426 | 0.8535 |

Table 27 presents the performance comparison on the PTB-XL benchmark using the full training dataset. Among supervised baselines, MEGA achieved the highest average performance across

all six tasks (0.8443), followed by XResNet1D-101 (0.8368) and LSTM (0.8328). Transformer (PTB-XL) achieved the best overall performance with an average AUC of 0.8564, demonstrating the effectiveness of self-supervised pretraining on real-world ECG data. Importantly, Transformer (SimECG) achieved an average AUC of 0.8485, outperforming all baselines including the strongest supervised baseline (MEGA) by 0.0042. This result demonstrates that synthetic data pretraining can effectively transfer to real-world ECG classification tasks and provides consistent performance gains over diverse supervised learning approaches.

### J.2.2 CLASSIFICATION WITH REDUCED TRAINING DATA

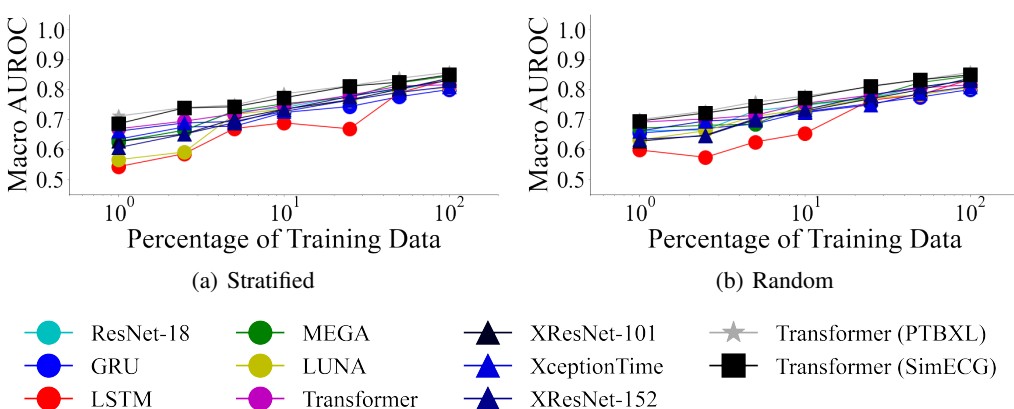

(a) Stratified

(b) Random

Figure 4: Performance comparison on PTB-XL with reduced training data. The plots show the average AUC across all six tasks as a function of training set size, using (a) stratified sampling and (b) random sampling. Models are compared at $50\%, 25\%, 10\%, 5\%, 2.5\%$, and $1\%$ of the original training set size.

Figure 4 shows performance comparison on PTB-XL with reduced training data, averaged over all six benchmark tasks. Under both stratified sampling (Figure 4(a)) and random sampling (Figure 4(b)), the Transformer (PTB-XL) consistently achieves the highest macro-AUC across all label budgets, with the SimECG-pretrained Transformer forming a stable second tier at every data fraction. This ordering between real-data and SimECG pretraining is preserved from the full-data regime down to the $1\%$ setting, whereas the purely supervised baselines degrade much more sharply as the number of labeled examples decreases. In particular, although MEGA is the strongest supervised model when trained on the full dataset, its performance drops substantially when the training set is reduced, and it is overtaken by both pretrained Transformers even at moderate data reductions. These trends highlight that pretraining—especially on real PTB-XL ECGs, but also on synthetic SimECG signals—yields markedly more label-efficient representations than supervised learning from scratch, with the important distinction that PTB-XL pretraining requires access to real ECG recordings, whereas SimECG pretraining can be performed entirely without real patient data.

Figure 5 shows task-specific performance on PTB-XL with reduced training data under stratified sampling. Each subplot reports macro-AUC as a function of the available training fraction ($50\%, 25\%, 10\%, 5\%, 2.5\%$, and $1\%$) for one of the six benchmark label spaces (all statements, diagnostic, sub-diagnostic, super-diagnostic, form, and rhythm). Across all tasks, the PTB-XL-pretrained Transformer and the SimECG-pretrained Transformer occupy the top curves for most label budgets, with the real-data pretrained model typically achieving the best or near-best performance and the SimECG-pretrained model closely tracking it. The gaps between different pretraining strategies are generally smaller than in our binary classification experiments, but the advantage of pretraining becomes more pronounced in the low-data regime, where purely supervised baselines exhibit noticeable drops in AUC. Overall, the figure shows that the qualitative pattern seen in the averaged results persists at the task level: pretraining on real ECGs gives the strongest performance, while SimECG pretraining provides a robust second-best alternative across diverse diagnostic groupings.

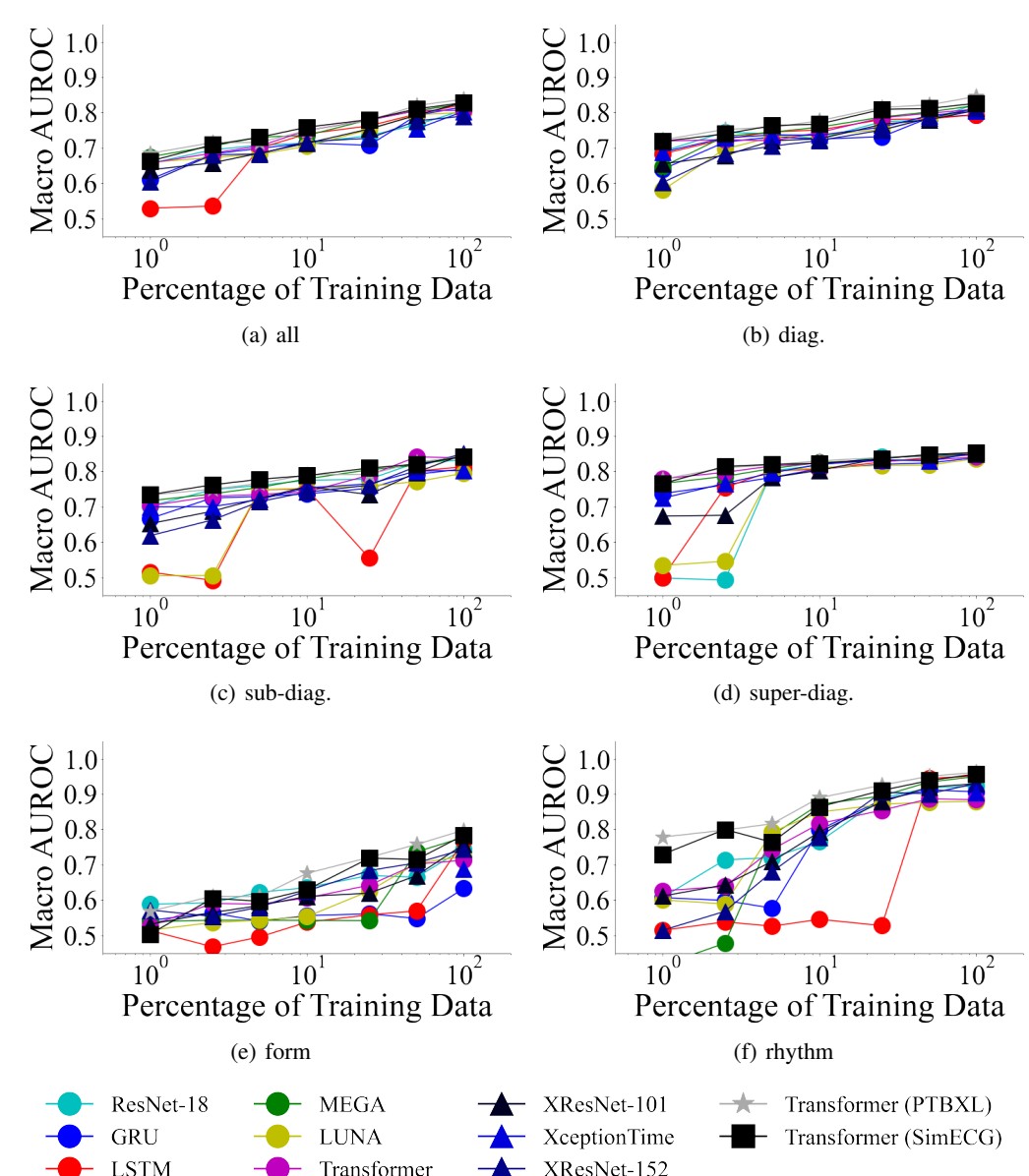

Figure 5: Task-specific performance comparison on PTB-XL with reduced training data using stratified sampling. Each subplot shows the performance on individual tasks (all, rhythm, form, diagnostic, subdiagnostic, and superdiagnostic) as a function of training set size at 50%, 25%, 10%, 5%, 2.5%, and 1% of the original training data.

Figure 6 shows task-specific performance on PTB-XL with reduced training data under random sampling. As in Figure 5, each subplot shows macro-AUC versus training set fraction for the six benchmark label spaces, but the reduced datasets are now obtained without enforcing label stratification. The overall trends are consistent with the stratified case: for each task, the PTB-XL-pretrained Transformer remains the strongest model over most label budgets, followed closely by the SimECG-pretrained Transformer, while the purely supervised baselines suffer larger performance degradation as the amount of labeled data decreases. Random sampling introduces additional variability at small data fractions—since some rare labels can be under-represented or absent—which further amplifies the instability of supervised training from scratch. In contrast, the two pretrained Transformers maintain relatively stable performance even when only 1–5% of the

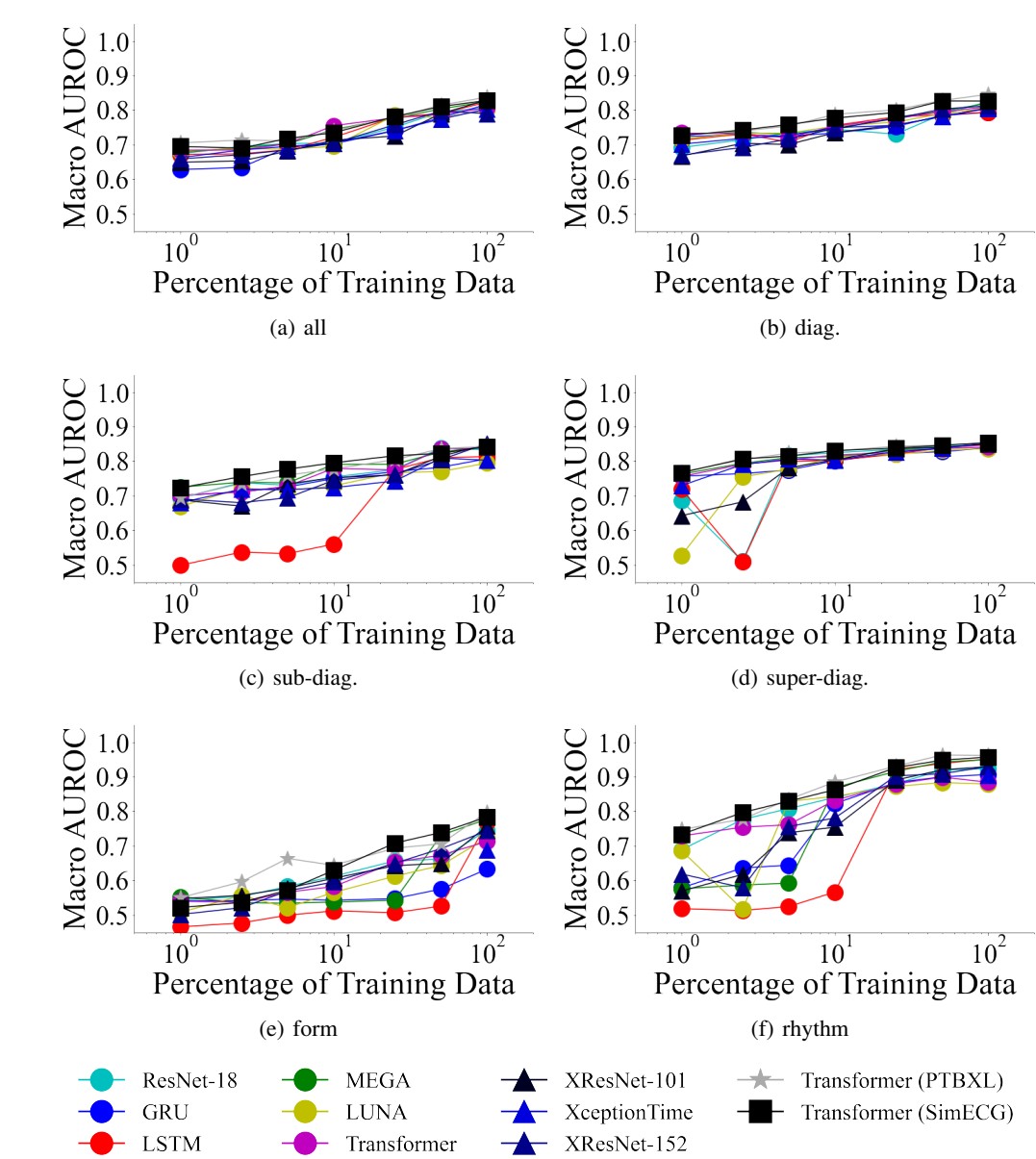

Figure 6: Task-specific performance comparison on PTB-XL with reduced training data using random sampling. Each subplot shows the performance on individual tasks (all, rhythm, form, diagnostic, subdiagnostic, and superdiagnostic) as a function of training set size at $50\%, 25\%, 10\%, 5\%, 2.5\%,$ and $1\%$ of the original training data.

original training set is available, reinforcing the conclusion that both real-data and simulator-based pretraining provide label-efficient representations for multi-label PTB-XL classification.

Overall, the experiments with PTB-XL benchmark task demonstrate that the benefits of pretraining extend beyond binary classification to multi-label benchmark tasks and persist even when the amount of labeled data is severely limited. Across all six PTB-XL label spaces and for both stratified and random subsampling at $50\%, 25\%, 10\%, 5\%, 2.5\%,$ and $1\%$ of the training set, Transformer (PTB-XL) consistently achieves the best or near-best macro-AUC, with Transformer (SimECG-N) forming a robust second tier, while strong supervised models such as MEGA degrade much more rapidly as the label budget shrinks. Although the gaps between pretraining strategies are smaller than in our binary classification setting, they become most apparent in the low-data regime and are qualitatively

consistent across tasks. These findings indicate that pretraining yields label-efficient representations for PTB-XL, with real-data pretraining providing the strongest performance when access to ECG records is possible, and SimECG pretraining offering a competitive, privacy-preserving alternative that does not require any real ECG data.

## K  COMPUTATIONAL BUDGET ANALYSIS AND SCALING BEHAVIOR

### K.1  EXPERIMENTAL SETUP

We conducted comprehensive experiments to compare pretraining performance under equivalent computational budgets across real-world and synthetic data. All models used identical architectures and were trained with the same hyperparameters to ensure fair comparison.

**Model Architecture:**  We employed a masked autoencoder framework with a Transformer based encoder backbone consistent with other sections. The input dimension was fixed across all experiments to maintain computational equivalence at each data point.

**Computational Budget:**  We tracked model performance by the number of processed data points during pretraining, ranging from $10^6$ to $10^9$. This metric ensures direct comparability regardless of the size of the pretraining dataset, as each condition processes the same total amount of data.

**Pretraining Datasets:**  We evaluated three types of pretraining data:

- Real-world ECG data: PTB-XL ($13,975$ training samples) and G12EC ($6,585$ samples)
- Synthetic ECG data: SimECG with varying dataset sizes ($2k, 5k, 10k, 20k, 50k, 100k, 200k, 500k$, and $1M$ samples)

**Evaluation Protocol:**  All pre-trained models were fine-tuned on the PTB-XL classification task, and performance was measured using macro $F_1$-score.

### K.2  PERFORMANCE UNDER EQUAL COMPUTATIONAL BUDGET

Here we explain details of Figure 1. In the early phase ($\sim 10^6$-$10^7$ processed samples), real-world pretraining achieves faster initial gains, reaching $F_1 \approx 0.82$-0.84. In the convergence phase ($\sim 10^8$ processed samples), synthetic pretraining with larger SimECG datasets ($\geq 200k$) converges to comparable performance, $F_1 \approx 0.84$-.85, essentially matching real-world pretraining. In the plateau phase ($> 10^8$ processed samples), all pretraining approaches plateau at $F_1 \approx 0.83$-0.85, and additional compute yields only marginal improvements.

Performance stabilizes around $200k$ samples, with $F_1$ scores in the range $0.8319 \sim 0.8481$ across different synthetic dataset sizes. These findings demonstrate that synthetic pretraining can match the performance of real-world pretraining under equivalent computational budgets, while offering the practical advantage of scalable data generation in data-scarce domains.

### K.3  PRETRAINING LOSS DYNAMICS

Figure 7 illustrates the pretraining loss trajectories for models trained on real-world data (PTB-XL, G12EC) and SimECG across the full computational budget range. This analysis reveals fundamental differences in learning stability between real-world and synthetic pretraining approaches.

Real-world pretraining models exhibit characteristic U-shaped loss curves, where the pretraining loss initially decreases but begins to increase after approximately $10^8$ steps. This pattern indicates overfitting, where continued training on limited real-world datasets leads to performance degradation. The limited size of real-world datasets (PTB-XL: $13,975$ samples, G12EC: $6,585$ samples) necessitates repeated cycling through the same data, eventually causing the model to overfit to specific patterns and lose previously learned representations.

For SimECG pretrainin, smaller synthetic datasets (e.g., $2k$–$50k$) still exhibit U-shaped curves similar to real-world data. As the synthetic dataset grows ($100k$–$1M$), overfitting is reduced and the

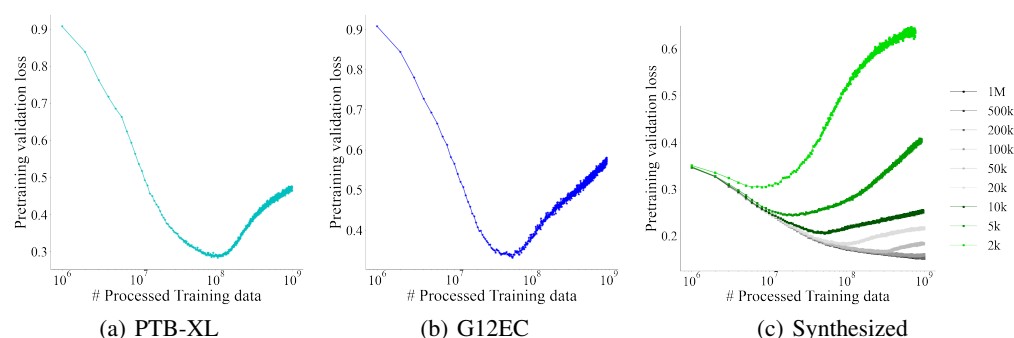

(a) PTB-XL      (b) G12EC      (c) Synthesized

Figure 7: Validation loss during pretraining. The loss for $500k$ and $1M$ synthesized data shows a continual decrease, whereas for other settings, the loss begins to increase.

minimum validation loss decreases with dataset size. For the largest datasets ($500k$ and $1M$), the validation loss continues to decrease throughout the entire training budget without showing overfitting. The consistently decreasing pretraining loss, combined with the converged downstream $F_1$-scores comparable to real-world approaches, demonstrates that knowledge-driven simulators provide sufficiently rich pretraining data for effective medical signal analysis.

These findings highlight a critical advantage of simulator-based pretraining: the ability to extend training stably without encountering the stability issues inherent to limited real-world datasets.

### K.4 SCALING ANALYSIS OF SYNTHETIC DATA SIZE

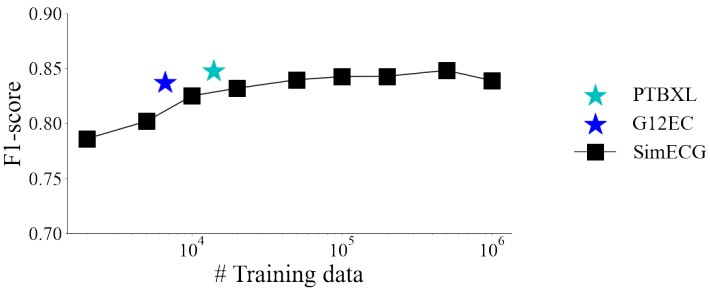

Figure 8: Performance scaling of SimECG pretraining ($2k$-$1M$ samples, black circles) versus real-world pretraining (PTB-XL and G12EC, stars). $F_1$-scores on PTB-XL downstream tasks plotted against training data size (logarithmic scale).

Table 28: Classification results on PTB-XL with varying SimECG-N dataset sizes. For reference, real-world pretraining with G12EC ($6,585$ training samples) achieves $F_1 = 0.8371$ and PTB-XL ($13,975$ training samples) achieves $F_1 = 0.8474$.

| # Data | 2k | 5k | 10k | 20k | 50k | 100k | 200k | 500k | 1M |
|---|---|---|---|---|---|---|---|---|---|
| $\bar{F}_1$ | 0.7860 | 0.8019 | 0.8249 | 0.8319 | 0.8396 | 0.8424 | 0.8427 | 0.8481 | 0.8389 |

Figure 8 illustrates the scaling behavior of SimECG pretraining compared to real-world baselines, while Table 28 provides detailed numerical results across different synthetic dataset sizes. SimECG demonstrates monotonic performance improvement as dataset size increases from $2k$ to $500k$ samples. With only $50k$ synthetic samples ($F_1 = 0.8396$), SimECG achieves competitive performance compared to G12EC pretraining ($F_1 = 0.8371$, $6,585$ samples) and approaches PTB-XL performance ($F_1 = 0.8474$, $13,975$ samples). Peak performance occurs at $500k$ samples ($F_1 = 0.8481$), while the slight decrease at $1M$ samples ($F_1 = 0.8389$) likely reflects model capacity constraints rather than synthetic data quality limitations.

The consistent improvement up to $500k$ samples demonstrates that synthetic data scaling effectively enhances representation learning without encountering an early plateau due to missing real-world nuances. According to neural scaling laws Kaplan et al. (2020), optimal model size should scale with dataset size, suggesting that fully leveraging datasets beyond 1M samples may require larger model architectures—an interesting direction for future research.

### K.5 PRETRAINING VALIDATION LOSS VS DOWNSTREAM TASK PERFORMANCE

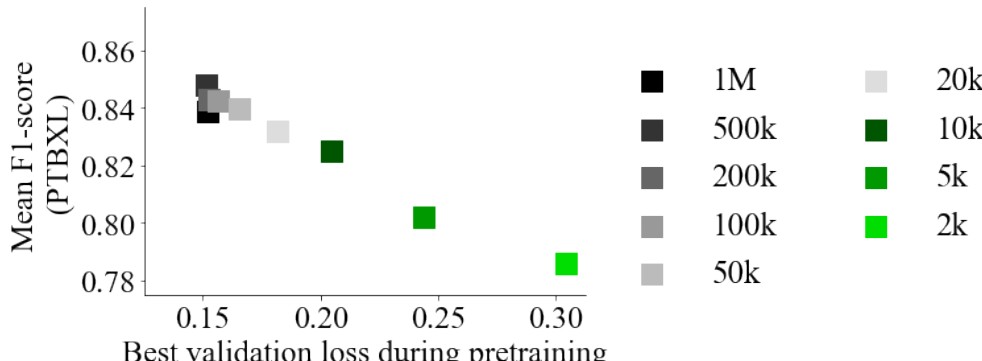

Figure 9: Relationship between pretraining validation loss and downstream $F_1$-score across SimECG-N dataset sizes ($2k$-$1M$ samples).

Figure 9 examines the relationship between pretraining validation loss and downstream performance across different SimECG dataset sizes. A strong negative correlation ($r = -0.988$) emerges between validation loss and $F_1$-score across all configurations. Larger synthetic datasets ($1M, 500k, 200k$ samples) achieve lower validation losses (0.15-0.20) with higher $F_1$-scores (0.84-0.85), while smaller datasets ($2k, 5k, 10k$ samples) exhibit higher validation losses (0.20-0.30) with correspondingly lower $F_1$-scores (0.78-0.82). Critically, validation loss continues to decrease as synthetic dataset size increases to $1M$ samples, without signs of saturation. This sustained improvement, combined with the strong predictive relationship to downstream performance ($r = -0.988$), indicates that representation quality within our studied range is still primarily limited by compute and dataset size rather than by fundamental simulator limitations.

A natural concern with synthetic data is whether performance plateaus due to missing real-world nuances. However, the continued improvement in validation loss and the absence of a pronounced degradation in downstream performance up to $1M$ samples (with only a small drop relative to $500k$, likely due to model-capacity limits) indicate that such limitations have not yet been reached at the scales we explored. Moreover, synthetic pretraining already achieves performance comparable to real-world pretraining ($F_1 \approx 0.84$-$0.85$) under equivalent computational budgets, demonstrating that knowledge-driven simulators capture sufficient signal characteristics for effective representation learning in ECG analysis.

### K.6 DISCUSSION

Our comprehensive analysis under equivalent computational budgets reveals several important insights about simulator-based pretraining for medical signal analysis.

**Computational Equivalence:** By tracking performance across processed data points ($10^6$-$10^9$) rather than training epochs, we ensure fair comparison across different dataset sizes. The convergence of all conditions to similar performance levels ($F_1 \approx 0.84$-$0.85$) demonstrates that synthetic and real-world pretraining achieve comparable effectiveness under equivalent computational resources.

**Over Fitting:** The U-shaped loss curves observed in real-world pretraining (PTB-XL, G12EC) indicate a fundamental limitation when scaling training duration on limited datasets. In contrast,

for sufficiently large SimECG datasets (e.g., $500k$ and $1M$), the loss continues to decrease throughout training, demonstrating the stability advantage of unlimited synthetic data generation, enabling longer training without performance degradation.

**Convergence Behavior:** The similar final performance across all conditions ($10^8$-$10^9$ steps, $F_1 \approx$ 0.84-0.85) suggests that the performance ceiling is determined by downstream task complexity and model capacity, not by inherent limitations of synthetic data. This finding addresses concerns about potential plateaus due to missing real-world nuances—while specific real-world variations may be absent in synthetic data, the core signal characteristics captured by knowledge-driven simulators prove sufficient for effective transfer learning.

**Model Capacity and Data Size:** The performance trajectory across synthetic dataset sizes ($20k$-$1M$) indicates that current model architecture is well-suited for datasets up to $500k$ samples. The slight performance decrease at $1M$ samples suggests that fully exploiting larger synthetic datasets may require proportionally larger model architectures, consistent with neural scaling laws. Future work could explore this relationship to further optimize the synthetic data scaling strategy.

## L  CLASSIFICATION RESULT WITH FROZEN PRETRAINED MODEL

Table 29: $F_1$-scores for classification with or without frozen Transformer parameters.

| Pretrainining dataset | SimECG-N | | PTB-XL | |
|---|---|---|---|---|
| Frozen Parameter | False | True | False | True |
| ABQRS | 0.8033 $\pm$0.0085 | 0.7328 $\pm$0.0037 | 0.8063 $\pm$0.0063 | 0.7841 $\pm$0.0067 |
| AF | 0.9753 $\pm$0.0023 | 0.8804 $\pm$0.0070 | 0.9808 $\pm$0.0030 | 0.9659 $\pm$0.0026 |
| ASMI | 0.7939 $\pm$0.0070 | 0.6529 $\pm$0.0071 | 0.8083 $\pm$0.0089 | 0.5273 $\pm$0.0380 |
| CRBBB | 0.9184 $\pm$0.0066 | 0.5424 $\pm$0.0135 | 0.9189 $\pm$0.0180 | 0.6259 $\pm$0.0566 |
| IAVB | 0.8847 $\pm$0.0050 | 0.4362 $\pm$0.0418 | 0.8875 $\pm$0.0067 | 0.7108 $\pm$0.0186 |
| IMI | 0.8588 $\pm$0.0119 | 0.6596 $\pm$0.0127 | 0.8368 $\pm$0.0069 | 0.7641 $\pm$0.0243 |
| IRBBB | 0.5822 $\pm$0.0200 | 0.4177 $\pm$0.0113 | 0.5758 $\pm$0.0333 | 0.4803 $\pm$0.0203 |
| ISC | 0.8947 $\pm$0.0157 | 0.7023 $\pm$0.0158 | 0.8989 $\pm$0.0120 | 0.7206 $\pm$0.0321 |
| LAFB | 0.9495 $\pm$0.0051 | 0.8806 $\pm$0.0058 | 0.9477 $\pm$0.0055 | 0.9272 $\pm$0.0055 |
| LVH | 0.8312 $\pm$0.0050 | 0.7001 $\pm$0.0113 | 0.8316 $\pm$0.0123 | 0.5805 $\pm$0.0436 |
| PAC | 0.7621 $\pm$0.0433 | 0.4127 $\pm$0.0050 | 0.8070 $\pm$0.0114 | 0.5438 $\pm$0.0342 |
| PVC | 0.9454 $\pm$0.0056 | 0.8054 $\pm$0.0217 | 0.9438 $\pm$0.0057 | 0.8757 $\pm$0.0117 |
| STD | 0.7881 $\pm$0.0156 | 0.4712 $\pm$0.0253 | 0.7911 $\pm$0.0129 | 0.5575 $\pm$0.0574 |

### L.1  PROTOCOL

To quantify how much of the downstream performance comes from linearly separable features versus task-specific adaptation, we compared two transfer regimes using the MAE-Transformer encoder selected in Appendix D.2: (i) *fine-tuning* (all encoder parameters updated) and (ii) *frozen* (the pretrained encoder kept fixed and only a terminal linear classifier trained; i.e., linear probing).

We evaluated both regimes for two pretraining sources: (a) real-world **PTB-XL_All** and (b) **SimECG-N** (1M traces generated by the simulator in Nonaka & Seita (2024)). Downstream tasks followed the main PTB-XL protocol: 13 binary abnormal-vs-normal classifiers on lead II (ABQRS, AF, ASMI, CRBBB, IAVB, IMI, IRBBB, ISC, LAFB, LVH, PAC, PVC, STD). The train/validation/test split, preprocessing (per-sample standardization), augmentations (random shifting and masking), class-imbalance handling (positive-class weighting), optimization (Adam), early stopping, and evaluation metric ($F_1$) were kept identical to the abnormal-ECG experiments. Results are summarized in Table 29.

### L.2 RESULTS AND DISCUSSION

Freezing the pretrained encoder deteriorated performance across *all* 13 PTB-XL tasks, irrespective of the pretraining source (Table 29). The drop was consistently larger with the encoder of Transformer (SimECG-N): in 11/13 tasks the frozen setting hurt more for the synthesized-pretrained model than for the Transformer (PTB-XL). Aggregated over tasks, the mean $F_1$ decreased by **32.8%** for Transformer (SimECG-N) encoder versus **18.9%** for Transformer (PTB-XL) encoder.

The magnitude of the decline varied by diagnosis. Conduction and interval abnormalities (e.g., IAVB, CRBBB) and morphology/segment changes (e.g., STD, PAC) showed especially pronounced drops when the encoder was frozen (e.g., IAVB: $0.8847 \rightarrow 0.4362$ with synthesized pretraining; CRBBB: $0.9184 \rightarrow 0.5424$). In contrast, categories with stronger, more stereotyped morphology (e.g., LAFB, PVC) were relatively less sensitive to freezing, particularly after real-world pretraining (e.g., LAFB: $0.9477 \rightarrow 0.9272$).

These findings imply that MAE alone yields features that are not universally linearly separable for all cardiac conditions on real data; some task-specific adaptation is still required. The larger frozen-vs-fine-tuned gap after synthesized pretraining likely reflects a residual domain shift between simulator outputs and real ECG distributions: without updating encoder weights, the linear head cannot fully bridge differences in rhythm statistics, noise characteristics, or lead-specific morphology.

## M PRETRAINING EXPERIMENTS WITH ALTERNATIVE ARCHITECTURES

To examine the effectiveness of SimECG pretraining across different architectures, we conducted additional experiments applying MAE-style pretraining to the top three non-Transformer baseline models: GRU, LSTM, and ResNet-18. This allows us to verify whether SimECG-based pretraining is beneficial not only for Transformers but also for other neural network architectures.

### M.1 EXPERIMENTAL SETUP

For GRU and LSTM, we adapted the MAE pretraining approach by masking random subsequences of the input time series (masking ratio: 75%) and training the models to reconstruct the masked portions. To preserve temporal dependencies crucial for recurrent architectures, visible tokens were fed to the encoder in chronological order, and the encoded representations were then reordered to match the original random masking pattern before decoding. For ResNet-18, we applied a similar chunk-based masking strategy where visible chunks are processed in temporal order by 1D convolutional layers. All models were pretrained on both PTB-XL and SimECG datasets for the same number of epochs as the Transformer baseline, followed by supervised fine-tuning on each classification task.

### M.2 RESULTS AND ANALYSIS

Tables 30, 31, and 32 present the $F_1$-scores for each of the 13 PTB-XL classification tasks under three conditions: (1) training from scratch (no pretraining), (2) pretraining on PTB-XL, and (3) pretraining on SimECG.

Both GRU and LSTM demonstrate substantial performance improvements with MAE-style pretraining. GRU achieves +5.35% improvement with PTB-XL pretraining and +5.74% with SimECG pretraining compared to training from scratch (macro-averaged across 13 tasks). LSTM shows even larger gains of +7.22% with PTB-XL and +6.86% with SimECG pretraining. To assess statistical significance, we conducted one-sided Wilcoxon signed-rank tests across the 13 tasks with Bonfer-

Table 30: $F_1$-scores for GRU classification with and without pretraining.

| Pretraining | - | PTBXL | SimECG-N |
|---|---|---|---|
| ABQRS | $0.7958 \pm 0.0046$ | $0.7934 \pm 0.0116$ | $0.7973 \pm 0.0080$ |
| AF | $0.9492 \pm 0.0136$ | $0.9643 \pm 0.0047$ | $0.9620 \pm 0.0041$ |
| ASMI | $0.7512 \pm 0.0244$ | $0.7946 \pm 0.0108$ | $0.7891 \pm 0.0154$ |
| CRBBB | $0.8019 \pm 0.0733$ | $0.9013 \pm 0.0109$ | $0.9130 \pm 0.0163$ |
| IAVB | $0.7977 \pm 0.0348$ | $0.8692 \pm 0.0129$ | $0.8681 \pm 0.0072$ |
| IMI | $0.7540 \pm 0.0129$ | $0.8389 \pm 0.0126$ | $0.8333 \pm 0.0094$ |
| IRBBB | $0.5471 \pm 0.0194$ | $0.5246 \pm 0.0167$ | $0.5554 \pm 0.0274$ |
| ISC | $0.8395 \pm 0.0359$ | $0.8872 \pm 0.0137$ | $0.8826 \pm 0.0103$ |
| LAFB | $0.9264 \pm 0.0100$ | $0.9427 \pm 0.0029$ | $0.9444 \pm 0.0033$ |
| LVH | $0.7762 \pm 0.0301$ | $0.8213 \pm 0.0106$ | $0.8070 \pm 0.0134$ |
| PAC | $0.6547 \pm 0.0564$ | $0.7119 \pm 0.0264$ | $0.7287 \pm 0.0218$ |
| PVC | $0.9212 \pm 0.0129$ | $0.9178 \pm 0.0018$ | $0.9294 \pm 0.0061$ |
| STD | $0.6957 \pm 0.0167$ | $0.7886 \pm 0.0167$ | $0.7858 \pm 0.0068$ |
| Mean | 0.7854 | 0.8274 | 0.8305 |

Table 31: $F_1$-scores for LSTM classification with and without pretraining.

| Pretraining | - | PTBXL | SimECG-N |
|---|---|---|---|
| ABQRS | $0.7931 \pm 0.0096$ | $0.7990 \pm 0.0085$ | $0.7895 \pm 0.0095$ |
| AF | $0.9571 \pm 0.0084$ | $0.9590 \pm 0.0067$ | $0.9632 \pm 0.0076$ |
| ASMI | $0.7305 \pm 0.0197$ | $0.7959 \pm 0.0072$ | $0.7893 \pm 0.0178$ |
| CRBBB | $0.7783 \pm 0.0493$ | $0.9203 \pm 0.0174$ | $0.9131 \pm 0.0193$ |
| IAVB | $0.7827 \pm 0.0411$ | $0.8735 \pm 0.0079$ | $0.8713 \pm 0.0066$ |
| IMI | $0.7262 \pm 0.0322$ | $0.8371 \pm 0.0133$ | $0.8316 \pm 0.0202$ |
| IRBBB | $0.5374 \pm 0.0161$ | $0.5335 \pm 0.0344$ | $0.5336 \pm 0.0348$ |
| ISC | $0.8635 \pm 0.0133$ | $0.8934 \pm 0.0042$ | $0.9022 \pm 0.0094$ |
| LAFB | $0.9245 \pm 0.0066$ | $0.9430 \pm 0.0022$ | $0.9503 \pm 0.0049$ |
| LVH | $0.7994 \pm 0.0162$ | $0.8148 \pm 0.0092$ | $0.8035 \pm 0.0024$ |
| PAC | $0.5792 \pm 0.0972$ | $0.6920 \pm 0.0192$ | $0.6893 \pm 0.0459$ |
| PVC | $0.9101 \pm 0.0146$ | $0.9295 \pm 0.0106$ | $0.9163 \pm 0.0115$ |
| STD | $0.6618 \pm 0.0285$ | $0.7784 \pm 0.0127$ | $0.7790 \pm 0.0237$ |
| Mean | 0.7726 | 0.8284 | 0.8256 |

roni correction for multiple comparisons (adjusted significance level: $\alpha = 0.025$). Both GRU and LSTM demonstrated statistically significant improvements over training from scratch for both pretraining datasets ($p < 0.025$ for all four comparisons: GRU (PTB-XL), GRU (SimECG-N), LSTM (PTB-XL), and LSTM (SimECG-N)).

In contrast, ResNet-18 exhibits performance degradation with MAE pretraining (0.6835 with pretraining vs. 0.7617 from scratch), suggesting that the patch-based masking strategy is less suitable for CNN architectures in the ECG domain. This may be attributed to the loss of local spatial structure that CNNs rely on for feature extraction. These results confirm that SimECG-based pretraining benefits not only Transformers but also sequential architectures (GRU, LSTM), demonstrating the broader applicability of simulator-based synthetic ECG data for self-supervised pretraining. Notably, even with pretraining, neither GRU nor LSTM surpasses the performance of Transformer (SimECG) (0.8318), reinforcing our choice of Transformer as the primary architecture for this study.

Table 32: $F_1$-scores for ResNet-18 classification with and without pretraining.

| Pretraining | - | PTBXL | SimECG-N |
|---|---|---|---|
| ABQRS | $0.7923 \pm 0.0101$ | $0.7098 \pm 0.0074$ | $0.7069 \pm 0.0153$ |
| AF | $0.9430 \pm 0.0193$ | $0.8833 \pm 0.0153$ | $0.8804 \pm 0.0100$ |
| ASMI | $0.7544 \pm 0.0251$ | $0.6686 \pm 0.0268$ | $0.6688 \pm 0.0238$ |
| CRBBB | $0.8067 \pm 0.0922$ | $0.8198 \pm 0.0245$ | $0.8133 \pm 0.0142$ |
| IAVB | $0.7910 \pm 0.0155$ | $0.6434 \pm 0.0408$ | $0.6577 \pm 0.0199$ |
| IMI | $0.7701 \pm 0.0284$ | $0.7135 \pm 0.0180$ | $0.6999 \pm 0.0325$ |
| IRBBB | $0.5669 \pm 0.0240$ | $0.4070 \pm 0.0278$ | $0.4048 \pm 0.0487$ |
| ISC | $0.8545 \pm 0.0324$ | $0.7693 \pm 0.0400$ | $0.7761 \pm 0.0081$ |
| LAFB | $0.9224 \pm 0.0120$ | $0.9247 \pm 0.0045$ | $0.9297 \pm 0.0057$ |
| LVH | $0.7845 \pm 0.0411$ | $0.7202 \pm 0.0129$ | $0.6982 \pm 0.0284$ |
| PAC | $0.3606 \pm 0.0213$ | $0.1496 \pm 0.0862$ | $0.1926 \pm 0.0828$ |
| PVC | $0.8838 \pm 0.0209$ | $0.7633 \pm 0.0154$ | $0.7711 \pm 0.0219$ |
| STD | $0.6719 \pm 0.0229$ | $0.7133 \pm 0.0152$ | $0.6589 \pm 0.0176$ |
| Mean | 0.7726 | 0.8284 | 0.8256 |

## N  PHONOCARDIOGRAM CLASSIFICATION RESULT

### N.1  MOTIVATION AND CONTEXT

While PTB-XL provides a relatively large-scale public ECG dataset for validation purposes, such extensive datasets are exceptional in medical signal domains. Most clinical modalities face severe data scarcity due to privacy regulations, high annotation costs, and limited data-sharing infrastructure. To demonstrate the practical value of our simulator-based pretraining approach in more realistic data-limited scenarios, we conducted additional experiments on phonocardiogram (PCG) data, where public datasets are substantially more constrained compared to ECG.

### N.2  EXPERIMENTAL SETUP

#### N.2.1  DATASET

We used the PhysioNet/Computing in Cardiology Challenge 2022 dataset (Reyna et al., 2023) for evaluation, which contains heart sound recordings with labels for three valve abnormalities: aortic regurgitation (AR), aortic stenosis (AS), and mitral regurgitation (MR). Unlike ECG datasets such as PTB-XL (21,799 samples) and MIMIC-IV-ECG ($\sim$800,000 samples), publicly available PCG datasets are significantly smaller, making this domain more representative of typical medical signal analysis challenges.

#### N.2.2  PRETRAINING DATA

We evaluated three pretraining conditions:

- **Real-world PCG**: PhysioNet2022 training split and BMD-HS dataset (Ali et al., 2024)
- **Simulator-based synthetic PCG (SimPCG)**:
- **No pretraining**: Supervised training from scratch

For SimPCG, we generated $100,000$ synthetic PCG samples matching the PhysioNet2022 specifications (sampling rate and duration). The same MAE pretraining protocol used for ECG experiments was applied to PCG data.

#### N.2.3  MODEL ARCHITECTURE AND TRAINING PROTOCOL

We employed the same Transformer architecture and five supervised baselines (GRU, LSTM, Luna, MEGA, ResNet-18) used in our ECG experiments.

### N.3 RESULTS

Table 33: AUROC for classification on phonocardiogram (PCG) data. The Transformer (SimPCG) achieves the best average performance, demonstrating the practical value of simulator-based pretraining in data-limited medical signal domains. Text colors indicate relative performance: Red denotes the best performance, while blue indicates the second-best performance for each task.

| Model | Pretraining Data | AR | AS | MR | Average |
|---|---|---|---|---|---|
| GRU | - | 0.5471 | 0.5508 | 0.4110 | 0.5030 |
| LSTM | - | 0.4424 | 0.5942 | 0.5989 | 0.5452 |
| Luna | - | 0.4327 | 0.3705 | 0.4255 | 0.4096 |
| MEGA | - | 0.5467 | 0.5495 | 0.5616 | 0.5526 |
| ResNet18 | - | 0.4756 | 0.3373 | 0.4042 | 0.4057 |
| Transformer | - | 0.5486 | 0.4449 | 0.5407 | 0.5114 |
| Transformer | PhysioNet2022 | 0.6151 | 0.5851 | 0.5741 | 0.5914 |
| Transformer | BMD-HS | 0.6273 | 0.5856 | 0.5774 | 0.5968 |
| Transformer | SimPCG | 0.6295 | 0.6356 | 0.5786 | 0.6146 |

Table 33 presents the classification results on PCG data. Transformer (SimPCG) achieves the best average AUROC (0.6146), substantially outperforming:

- The strongest supervised baseline (MEGA: 0.5526; $+11.2\%$ relative improvement)

- Transformers pretrained on limited real PCG data (BMD-HS: 0.5968; $+3.0\%$ relative improvement)

- Transformer without pretraining (0.5114; $+20.2\%$ relative improvement)

Notably, the Transformer (SimPCG) achieves the best performance on two out of three tasks (AR and AS) and remains competitive on the third (MR). The consistent improvements across all tasks demonstrate that simulator-based pretraining provides robust initialization even when real-world pretraining data is limited.

# O    CLASSIFICATION RESULT WHEN TRAINED WITH OTHER LEADS

We examined the abnormal ECG classification performance when trained with lead other than lead II. For a preprocessing and an augmentation, in line with previous experiments, we applied a scaling as preprocessing and random shifting and random masking as an augmentation. The hyperparameters were reused from corresponding experiment with lead II ECG. An optimizer, batch size, number of epochs, evaluation frequency and early stopping setting were kept the same as in the previous experiment on abnormal ECG classification.

The detailed results for lead I is in Table 34, lead III is in Table 35, lead aVF is in Table 36, lead aVL is in Table 37, lead aVR is in Table 38, lead V1 is in Table 39, lead V2 is in Table 40, lead V3 is in Table 41, lead V4 is in Table 42, lead V5 is in Table 43, and lead V6 is in Table 44.[11]  In addition to the 11 leads, we show the results with lead II, which are shown in Table 21. For all leads, the Transformer (SimECG) showed the best average ranking over five baseline models and Transformer (Scratch).

Table 34: $F_1$-scores for classification on the PTB-XL dataset using lead I (P.T. = pretraining). Text colors indicate relative performance: Red denotes the best performance, while blue indicates the second-best performance for each task.

| | GRU | LSTM | Luna | MEGA | ResNet18 | Transformer | |
| | - | - | - | - | - | - | |
| P.T. | | | | | | - | SimECG |
| ABQRS | 0.7419 ±0.0093 | 0.7379 ±0.0013 | 0.7193 ±0.0080 | 0.7361 ±0.0055 | 0.7468 ±0.0068 | 0.7180 ±0.0053 | 0.7344 ±0.0145 |
| AF | 0.9715 ±0.0074 | 0.9671 ±0.0102 | 0.9335 ±0.0127 | 0.9497 ±0.0058 | 0.9502 ±0.0309 | 0.9392 ±0.0105 | 0.9753 ±0.0056 |
| ASMI | 0.8031 ±0.0275 | 0.7883 ±0.0153 | 0.5276 ±0.3231 | 0.7642 ±0.0224 | 0.8103 ±0.0218 | 0.7833 ±0.0094 | 0.8281 ±0.0093 |
| CRBBB | 0.9140 ±0.0362 | 0.9368 ±0.0334 | 0.9196 ±0.0167 | 0.9410 ±0.0184 | 0.8987 ±0.0623 | 0.9134 ±0.0285 | 0.9620 ±0.0102 |
| IAVB | 0.7664 ±0.0287 | 0.7667 ±0.0130 | 0.7287 ±0.0195 | 0.7028 ±0.0427 | 0.7296 ±0.0340 | 0.7780 ±0.0406 | 0.8673 ±0.0102 |
| IMI | 0.6185 ±0.0394 | 0.5807 ±0.0446 | 0.4976 ±0.1453 | 0.5878 ±0.0561 | 0.6239 ±0.0190 | 0.6179 ±0.0139 | 0.6894 ±0.0238 |
| IRBBB | 0.4492 ±0.0212 | 0.4533 ±0.0058 | 0.4212 ±0.0321 | 0.4352 ±0.0181 | 0.4713 ±0.0555 | 0.4270 ±0.0116 | 0.5067 ±0.0091 |
| ISC | 0.8692 ±0.0179 | 0.8664 ±0.0217 | 0.8465 ±0.0312 | 0.8598 ±0.0142 | 0.8686 ±0.0256 | 0.8512 ±0.0184 | 0.9061 ±0.0095 |
| LAFB | 0.6080 ±0.0234 | 0.6297 ±0.0104 | 0.6061 ±0.0134 | 0.6186 ±0.0159 | 0.6995 ±0.0500 | 0.6048 ±0.0162 | 0.7036 ±0.0084 |
| LVH | 0.8037 ±0.0215 | 0.8040 ±0.0253 | 0.7879 ±0.0164 | 0.7813 ±0.0294 | 0.7648 ±0.0497 | 0.7445 ±0.0154 | 0.8482 ±0.0090 |
| PAC | 0.7042 ±0.0625 | 0.5973 ±0.0919 | 0.2655 ±0.0342 | 0.6424 ±0.0537 | 0.3502 ±0.0747 | 0.2794 ±0.0222 | 0.7608 ±0.0266 |
| PVC | 0.9167 ±0.0133 | 0.9194 ±0.0141 | 0.7650 ±0.0302 | 0.8795 ±0.0250 | 0.9264 ±0.0122 | 0.8268 ±0.0408 | 0.9542 ±0.0067 |
| STD | 0.7130 ±0.0089 | 0.6958 ±0.0356 | 0.5996 ±0.1201 | 0.6962 ±0.0109 | 0.6737 ±0.0586 | 0.6960 ±0.0301 | 0.8335 ±0.0131 |
| Mean | 0.7600 | 0.7495 | 0.6629 | 0.7380 | 0.7318 | 0.7061 | 0.8130 |
| Avg. rank | 3.15 | 3.54 | 6.31 | 4.54 | 3.62 | 5.54 | 1.31 |

Regarding classification result with lead I ECG, as shown in Table 34, the Transformer (SimECG) exhibits the best performance on 12 out of 13 tasks. The average rank of the Transformer (SimECG) across the 13 tasks 1.31.

---

[11]For lead V5, the results for ASMI, ISC, STD is ommitted, since there were errors in original data.

Table 35: $F_1$-scores for classification on the PTB-XL dataset using lead III (P.T. = pretraining). Text colors indicate relative performance: Red denotes the best performance, while blue indicates the second-best performance for each task.

| | GRU | LSTM | Luna | MEGA | ResNet18 | Transformer | |
|---|---|---|---|---|---|---|---|
| P.T. | - | - | - | - | - | - | SimECG |
| ABQRS | 0.7462 ±0.0072 | 0.7352 ±0.0057 | 0.7175 ±0.0038 | 0.7240 ±0.0077 | 0.7530 ±0.0080 | 0.7284 ±0.0025 | 0.7591 ±0.0038 |
| AF | 0.9489 ±0.0081 | 0.9113 ±0.0257 | 0.7213 ±0.0134 | 0.8424 ±0.0379 | 0.9042 ±0.0144 | 0.7518 ±0.0182 | 0.9567 ±0.0068 |
| ASMI | 0.6326 ±0.0196 | 0.6377 ±0.0160 | 0.6089 ±0.0107 | 0.5452 ±0.0247 | 0.6684 ±0.0342 | 0.6164 ±0.0176 | 0.7070 ±0.0055 |
| CRBBB | 0.6532 ±0.0157 | 0.7170 ±0.0511 | 0.6192 ±0.0504 | 0.6549 ±0.0792 | 0.8294 ±0.0502 | 0.5993 ±0.0491 | 0.8868 ±0.0191 |
| IAVB | 0.7074 ±0.0497 | 0.7034 ±0.0216 | 0.5393 ±0.0402 | 0.5686 ±0.0359 | 0.7156 ±0.0580 | 0.6062 ±0.0324 | 0.8484 ±0.0116 |
| IMI | 0.7322 ±0.0131 | 0.7167 ±0.0265 | 0.6841 ±0.0229 | 0.6884 ±0.0622 | 0.7656 ±0.0272 | 0.7494 ±0.0278 | 0.8585 ±0.0141 |
| IRBBB | 0.3717 ±0.0139 | 0.3194 ±0.0208 | 0.3081 ±0.0157 | 0.3336 ±0.0218 | 0.4195 ±0.0474 | 0.3312 ±0.0133 | 0.4079 ±0.0280 |
| ISC | 0.5527 ±0.0120 | 0.5814 ±0.0416 | 0.5267 ±0.0153 | 0.4876 ±0.0239 | 0.5772 ±0.0534 | 0.5252 ±0.0307 | 0.6903 ±0.0204 |
| LAFB | 0.7208 ±0.0145 | 0.7261 ±0.0044 | 0.7338 ±0.0077 | 0.7150 ±0.0061 | 0.7592 ±0.0169 | 0.7399 ±0.0127 | 0.8096 ±0.0056 |
| LVH | 0.5304 ±0.0281 | 0.5397 ±0.0189 | 0.4907 ±0.0230 | 0.4512 ±0.0233 | 0.5405 ±0.0252 | 0.5188 ±0.0310 | 0.6136 ±0.0214 |
| PAC | 0.5628 ±0.0379 | 0.3348 ±0.1328 | 0.1635 ±0.0398 | 0.2177 ±0.0216 | 0.2105 ±0.0151 | 0.1819 ±0.0383 | 0.6519 ±0.0405 |
| PVC | 0.8846 ±0.0208 | 0.8833 ±0.0156 | 0.7665 ±0.0251 | 0.8020 ±0.0430 | 0.8397 ±0.0448 | 0.7925 ±0.0227 | 0.9174 ±0.0151 |
| STD | 0.4289 ±0.0063 | 0.4328 ±0.0168 | 0.3747 ±0.0118 | 0.4001 ±0.0139 | 0.4147 ±0.0508 | 0.4220 ±0.0279 | 0.5572 ±0.0252 |
| Mean | 0.6517 | 0.6338 | 0.5580 | 0.5716 | 0.6460 | 0.5818 | 0.7434 |
| Avg. rank | 3.46 | 3.54 | 6.38 | 5.69 | 2.77 | 5.08 | 1.08 |

Regarding classification result with lead III ECG, as shown in Table 35, the Transformer (SimECG) exhibits the best performance on 12 out of 13 tasks and the second-best performance on the remaining task. The average rank of the Transformer (SimECG) across the 13 tasks was 1.08.

Regarding classification result with lead aVF ECG, as shown in Table 36, the Transformer (SimECG) exhibits the best performance on 12 out of 13 tasks. The average rank of the Transformer (SimECG) across the 13 tasks was 1.23.

Regarding classification result with lead aVL ECG, as shown in Table 37, the Transformer (SimECG) exhibits the best performance on 12 out of 13 tasks and the second-best performance on remaining 3 tasks. The average rank of the Transformer (SimECG) across the 13 tasks was 1.23.

Regarding classification result with lead aVR ECG, as shown in Table 38, the Transformer (SimECG) exhibits the best performance on 12 out of 13 tasks. The average rank of the Transformer (SimECG) across the 13 tasks was 1.15.

Regarding classification result with lead V1 ECG, as shown in Table 39, the Transformer (SimECG) exhibits the best performance on 12 out of 13 tasks. The average rank of the Transformer (SimECG) across the 13 tasks was 1.23.

Regarding classification result with lead V2 ECG, as shown in Table 40, the Transformer (SimECG) exhibits the best performance on all 13 tasks. The average rank of the Transformer (SimECG) across the 13 tasks was 1.00.

Table 36: $F_1$-scores for classification on the PTB-XL dataset using lead aVF (P.T. = pretraining). Text colors indicate relative performance: Red denotes the best performance, while blue indicates the second-best performance for each task.

| | GRU | LSTM | Luna | MEGA | ResNet18 | Transformer | |
| --- | --- | --- | --- | --- | --- | --- | --- |
| P.T. | - | - | - | - | - | - | SimECG |
| ABQRS | 0.7698 ±0.0055 | 0.7674 ±0.0047 | 0.7512 ±0.0055 | 0.7544 ±0.0086 | 0.7788 ±0.0036 | 0.7629 ±0.0090 | 0.7788 ±0.0072 |
| AF | 0.9574 ±0.0122 | 0.9311 ±0.0245 | 0.8735 ±0.0122 | 0.9100 ±0.0132 | 0.9369 ±0.0172 | 0.8695 ±0.0082 | 0.9656 ±0.0040 |
| ASMI | 0.6610 ±0.0157 | 0.6681 ±0.0123 | 0.6217 ±0.0211 | 0.6192 ±0.0161 | 0.7216 ±0.0167 | 0.6422 ±0.0081 | 0.7481 ±0.0099 |
| CRBBB | 0.5710 ±0.0411 | 0.6102 ±0.0633 | 0.5549 ±0.0297 | 0.5829 ±0.0907 | 0.8191 ±0.0636 | 0.5623 ±0.0308 | 0.8801 ±0.0159 |
| IAVB | 0.6976 ±0.0230 | 0.7227 ±0.0644 | 0.6393 ±0.0487 | 0.6388 ±0.0126 | 0.7515 ±0.0183 | 0.6520 ±0.0284 | 0.8662 ±0.0068 |
| IMI | 0.8004 ±0.0104 | 0.8131 ±0.0238 | 0.7544 ±0.0211 | 0.7879 ±0.0211 | 0.8419 ±0.0413 | 0.7986 ±0.0238 | 0.8952 ±0.0118 |
| IRBBB | 0.4489 ±0.0136 | 0.4268 ±0.0118 | 0.4343 ±0.0146 | 0.4317 ±0.0186 | 0.4564 ±0.0603 | 0.4467 ±0.0115 | 0.4374 ±0.0134 |
| ISC | 0.6616 ±0.0149 | 0.6807 ±0.0129 | 0.6296 ±0.0338 | 0.6204 ±0.0280 | 0.6450 ±0.0289 | 0.6409 ±0.0166 | 0.7478 ±0.0180 |
| LAFB | 0.8550 ±0.0097 | 0.8569 ±0.0118 | 0.8434 ±0.0104 | 0.8605 ±0.0023 | 0.8732 ±0.0084 | 0.8597 ±0.0067 | 0.9019 ±0.0063 |
| LVH | 0.6408 ±0.0146 | 0.6510 ±0.0219 | 0.6171 ±0.0238 | 0.5924 ±0.0144 | 0.6542 ±0.0289 | 0.6068 ±0.0088 | 0.7378 ±0.0108 |
| PAC | 0.5674 ±0.0480 | 0.4988 ±0.1307 | 0.2107 ±0.0470 | 0.2746 ±0.1281 | 0.2518 ±0.0177 | 0.2408 ±0.0183 | 0.6867 ±0.0168 |
| PVC | 0.8859 ±0.0181 | 0.8818 ±0.0169 | 0.7737 ±0.0248 | 0.8165 ±0.0380 | 0.8427 ±0.0535 | 0.8029 ±0.0320 | 0.9327 ±0.0084 |
| STD | 0.5342 ±0.0226 | 0.4988 ±0.0131 | 0.5095 ±0.0354 | 0.5298 ±0.0219 | 0.4994 ±0.0493 | 0.5266 ±0.0219 | 0.6583 ±0.0086 |
| Mean | 0.6962 | 0.6929 | 0.6318 | 0.6476 | 0.6979 | 0.6471 | 0.7874 |
| Avg. rank | 3.23 | 3.77 | 6.15 | 5.31 | 2.77 | 5.08 | 1.23 |

Regarding classification result with lead V3 ECG, as shown in Table 41, the Transformer (SimECG) exhibits the best performance on 12 out of 13 tasks and the second-best performance on the remaining task. The average rank of the Transformer (SimECG) across the 13 tasks was 1.08.

Regarding classification result with lead V4 ECG, as shown in Table 42, the Transformer (SimECG) exhibits the best performance on 10 out of 13 tasks. The average rank of the Transformer (SimECG) across the 13 tasks was 1.23.

Regarding classification result with lead V5 ECG, as shown in Table 43, the Transformer (SimECG) exhibits the best performance on 9 out of 10 tasks. The average rank of the Transformer (SimECG) across the 10 tasks was 1.20.

Regarding classification result with lead V6 ECG, as shown in Table 44, the Transformer (SimECG) exhibits the best performance on 12 out of 13 tasks. The average rank of the Transformer (SimECG) across the 13 tasks was 1.15.

Table 37: $F_1$-scores for classification on the PTB-XL dataset using lead aVL (P.T. = pretraining). Text colors indicate relative performance: Red denotes the best performance, while blue indicates the second-best performance for each task.

| | GRU | LSTM | Luna | MEGA | ResNet18 | Transformer | |
| --- | --- | --- | --- | --- | --- | --- | --- |
| P.T. | - | - | - | - | - | - | SimECG |
| ABQRS | 0.7240 ±0.0077 | 0.7239 ±0.0055 | 0.6934 ±0.0099 | 0.7151 ±0.0132 | 0.7448 ±0.0091 | 0.7225 ±0.0044 | 0.7401 ±0.0050 |
| AF | 0.9166 ±0.0280 | 0.9227 ±0.0099 | 0.7349 ±0.0236 | 0.8531 ±0.0327 | 0.8954 ±0.0235 | 0.7193 ±0.0250 | 0.9561 ±0.0028 |
| ASMI | 0.6980 ±0.0395 | 0.7004 ±0.0114 | 0.5960 ±0.1491 | 0.6484 ±0.0131 | 0.7690 ±0.0187 | 0.6869 ±0.0185 | 0.7681 ±0.0181 |
| CRBBB | 0.8606 ±0.0187 | 0.8235 ±0.0546 | 0.7747 ±0.0186 | 0.8402 ±0.0109 | 0.8620 ±0.0492 | 0.8331 ±0.0105 | 0.9404 ±0.0071 |
| IAVB | 0.6303 ±0.0358 | 0.5904 ±0.0495 | 0.4647 ±0.0209 | 0.4956 ±0.0398 | 0.7354 ±0.0622 | 0.5106 ±0.0317 | 0.8259 ±0.0179 |
| IMI | 0.6273 ±0.0220 | 0.6421 ±0.0233 | 0.5930 ±0.0121 | 0.5865 ±0.0347 | 0.6788 ±0.0259 | 0.6387 ±0.0168 | 0.7534 ±0.0092 |
| IRBBB | 0.3682 ±0.0063 | 0.3304 ±0.0186 | 0.3021 ±0.0206 | 0.3257 ±0.0147 | 0.4432 ±0.0131 | 0.3106 ±0.0042 | 0.4234 ±0.0207 |
| ISC | 0.6559 ±0.0235 | 0.6969 ±0.0222 | 0.6409 ±0.0121 | 0.6502 ±0.0102 | 0.7503 ±0.0395 | 0.6255 ±0.0262 | 0.8108 ±0.0105 |
| LAFB | 0.6465 ±0.0215 | 0.6695 ±0.0109 | 0.6626 ±0.0128 | 0.6555 ±0.0107 | 0.6907 ±0.0133 | 0.6620 ±0.0136 | 0.7405 ±0.0115 |
| LVH | 0.6310 ±0.0372 | 0.6242 ±0.0216 | 0.5986 ±0.0153 | 0.5902 ±0.0209 | 0.6345 ±0.0443 | 0.6268 ±0.0249 | 0.7398 ±0.0062 |
| PAC | 0.6169 ±0.0432 | 0.4302 ±0.1580 | 0.0680 ±0.0353 | 0.1710 ±0.0441 | 0.1908 ±0.0335 | 0.1932 ±0.0089 | 0.6595 ±0.0364 |
| PVC | 0.8731 ±0.0172 | 0.8829 ±0.0261 | 0.7577 ±0.0461 | 0.7381 ±0.1418 | 0.8727 ±0.0253 | 0.7934 ±0.0226 | 0.9329 ±0.0044 |
| STD | 0.5241 ±0.0096 | 0.5165 ±0.0242 | 0.4636 ±0.0838 | 0.5140 ±0.0099 | 0.5275 ±0.0347 | 0.5055 ±0.0077 | 0.6617 ±0.0254 |
| Mean | 0.6748 | 0.6580 | 0.5654 | 0.5987 | 0.6765 | 0.6022 | 0.7656 |
| Avg. rank | 3.54 | 3.54 | 6.38 | 5.77 | 2.31 | 5.23 | 1.23 |

## P    12-LEAD ECG CLASSIFICATION RESULTS

In this appendix, we investigate whether models pretrained on single-lead ECG data can be effectively transferred to standard 12-lead ECG classification. Specifically, we reuse the lead-II–pretrained models and evaluate their performance when fine-tuned on 12-lead ECGs for the same 13 diagnostic labels.

### P.1    EXPERIMENTAL SETUP

We keep training hyperparameters identical to the single-lead setting, including the optimizer, learning rate schedule, batch size, and number of training epochs. For the end-to-end baselines (ResNet-18, GRU, LSTM, LUNA, MEGA, and Transformer), we simply extend the input from one channel to 12 channels and leave the rest of the architecture unchanged.

For the pretrained MAE-based Transformer (PTB-XL and SimECG-N), we adapt the single-lead encoder to 12-lead input using a shared-encoder wrapper. Given a 12-lead ECG $x \in \mathbb{R}^{B \times 12 \times T}$, the single-lead encoder is applied independently to each lead to obtain 12 per-lead embeddings. These 12 embeddings are then stacked and passed through a small transformer encoder that models inter-lead dependencies and produces an aggregated representation, which is finally fed to the same classification head as in the single-lead experiments.

Table 38: $F_1$-scores for classification on the PTB-XL dataset using lead aVR (P.T. = pretraining). Text colors indicate relative performance: Red denotes the best performance, while blue indicates the second-best performance for each task.

| | GRU | LSTM | Luna | MEGA | ResNet18 | Transformer | |
|---|---|---|---|---|---|---|---|
| P.T. | - | - | - | - | - | - | SimECG-N |
| ABQRS | 0.7853 ±0.0045 | 0.7771 ±0.0064 | 0.7525 ±0.0090 | 0.7722 ±0.0064 | 0.7875 ±0.0082 | 0.7667 ±0.0058 | 0.7850 ±0.0071 |
| AF | 0.9667 ±0.0102 | 0.9588 ±0.0221 | 0.9473 ±0.0091 | 0.9516 ±0.0180 | 0.9562 ±0.0145 | 0.8053 ±0.3046 | 0.9806 ±0.0042 |
| ASMI | 0.7997 ±0.0150 | 0.7883 ±0.0072 | 0.6798 ±0.1913 | 0.7635 ±0.0108 | 0.7749 ±0.0464 | 0.7835 ±0.0102 | 0.8238 ±0.0199 |
| CRBBB | 0.9278 ±0.0153 | 0.9212 ±0.0136 | 0.9035 ±0.0301 | 0.9171 ±0.0227 | 0.9266 ±0.0220 | 0.9188 ±0.0284 | 0.9653 ±0.0115 |
| IAVB | 0.8233 ±0.0233 | 0.8112 ±0.0132 | 0.7524 ±0.0433 | 0.7935 ±0.0271 | 0.8362 ±0.0218 | 0.8130 ±0.0103 | 0.8792 ±0.0049 |
| IMI | 0.6867 ±0.0130 | 0.6964 ±0.0178 | 0.6702 ±0.0311 | 0.6974 ±0.0422 | 0.6596 ±0.0166 | 0.6829 ±0.0195 | 0.7831 ±0.0053 |
| IRBBB | 0.5695 ±0.0218 | 0.5503 ±0.0287 | 0.5350 ±0.0119 | 0.5424 ±0.0132 | 0.5534 ±0.0369 | 0.5296 ±0.0206 | 0.5698 ±0.0250 |
| ISC | 0.9001 ±0.0176 | 0.8994 ±0.0196 | 0.8967 ±0.0060 | 0.8787 ±0.0207 | 0.9205 ±0.0138 | 0.8836 ±0.0104 | 0.9335 ±0.0028 |
| LAFB | 0.7773 ±0.0048 | 0.7833 ±0.0109 | 0.7772 ±0.0144 | 0.7767 ±0.0138 | 0.8304 ±0.0126 | 0.7881 ±0.0134 | 0.8315 ±0.0083 |
| LVH | 0.8614 ±0.0111 | 0.8387 ±0.0168 | 0.7852 ±0.0397 | 0.8135 ±0.0148 | 0.8264 ±0.0435 | 0.8026 ±0.0083 | 0.8657 ±0.0094 |
| PAC | 0.6982 ±0.0635 | 0.7177 ±0.0298 | 0.2815 ±0.0438 | 0.5939 ±0.0795 | 0.3865 ±0.0475 | 0.3306 ±0.0402 | 0.7736 ±0.0384 |
| PVC | 0.9358 ±0.0104 | 0.9228 ±0.0155 | 0.7786 ±0.0274 | 0.9040 ±0.0188 | 0.9234 ±0.0323 | 0.8556 ±0.0148 | 0.9490 ±0.0067 |
| STD | 0.7349 ±0.0190 | 0.7194 ±0.0357 | 0.7175 ±0.0233 | 0.7104 ±0.0273 | 0.7862 ±0.0283 | 0.7318 ±0.0156 | 0.8343 ±0.0106 |
| Mean | 0.8051 | 0.7988 | 0.7290 | 0.7781 | 0.7821 | 0.7455 | 0.8442 |
| Avg. rank | 2.69 | 3.69 | 6.46 | 5.38 | 3.31 | 5.31 | 1.15 |

## P.2 PER-LABEL PERFORMANCE

Table 45 presents a comparison of pretraining methods on 12-lead ECG classification tasks. Transformer (PTB-XL) and PTB-XL (SimECG-N) were compared across 13 diagnostic tasks, showing nearly equivalent performance with mean $F_1$ values of 0.9110 and 0.9098, respectively. For individual tasks, Transformer (SimECG-N) slightly outperformed Transformer (PTB-XL) on tasks such as IRBBB and IMI, while Transformer (PTB-XL) showed marginal advantages on tasks including AF, ASMI, and IAVB. Overall, the performance difference between the two methods was very small across all tasks.

Table 46 presents per-label 12-lead ECG classification performance across six different model architectures (GRU, LSTM, Luna, MEGA, ResNet18, and Transformer), with the Transformer model pre-trained using SimECG. The SimECG pre-trained Transformer achieved a mean $F_1$ of 0.9098 with a relative improvement of 1.5628% in $F_1$, substantially outperforming all baseline architectures trained from scratch. Among the scratch-trained models, GRU showed the best performance with a mean $F_1$ of 0.8958, followed by LSTM (0.8779) and Transformer (0.8752). For individual diagnostic tasks, the SimECG pre-trained model demonstrated particularly strong performance on AF (0.9833), CRBBB (0.9842), ASMI (0.9532), and ISC (0.9605), marking the best or second-best results across most labels. Notably, on challenging tasks such as PAC and PVC, SimECG pretraining provided substantial improvements over scratch-trained models, with PAC performance reaching 0.8021 compared to the best scratch-trained baseline of 0.6845.

Table 39: $F_1$-scores for classification on the PTB-XL dataset using lead V1 (P.T. = pretraining). Text colors indicate relative performance: Red denotes the best performance, while blue indicates the second-best performance for each task.

| | GRU | LSTM | Luna | MEGA | ResNet18 | Transformer | |
|---|---|---|---|---|---|---|---|
| P.T. | - | - | - | - | - | - | SimECG-N |
| ABQRS | 0.7459 ±0.0069 | 0.7394 ±0.0081 | 0.7077 ±0.0058 | 0.7367 ±0.0107 | 0.7474 ±0.0074 | 0.7227 ±0.0061 | 0.7389 ±0.0061 |
| AF | 0.9605 ±0.0077 | 0.9500 ±0.0073 | 0.8606 ±0.0210 | 0.8963 ±0.0186 | 0.9227 ±0.0140 | 0.8399 ±0.0219 | 0.9635 ±0.0063 |
| ASMI | 0.7846 ±0.0219 | 0.7668 ±0.0177 | 0.5471 ±0.2047 | 0.7458 ±0.0147 | 0.7886 ±0.0256 | 0.7662 ±0.0071 | 0.8190 ±0.0080 |
| CRBBB | 0.9624 ±0.0187 | 0.9663 ±0.0092 | 0.9628 ±0.0102 | 0.9716 ±0.0137 | 0.9371 ±0.0712 | 0.9621 ±0.0162 | 0.9842 ±0.0023 |
| IAVB | 0.7657 ±0.0167 | 0.7433 ±0.0562 | 0.6286 ±0.0473 | 0.6600 ±0.0255 | 0.7506 ±0.0254 | 0.6437 ±0.0287 | 0.8826 ±0.0124 |
| IMI | 0.5630 ±0.0274 | 0.5702 ±0.0245 | 0.1688 ±0.1062 | 0.3994 ±0.0248 | 0.5984 ±0.0496 | 0.5318 ±0.0134 | 0.6575 ±0.0215 |
| IRBBB | 0.7688 ±0.0152 | 0.7436 ±0.0169 | 0.7283 ±0.0097 | 0.7343 ±0.0189 | 0.7669 ±0.0336 | 0.7430 ±0.0152 | 0.8022 ±0.0116 |
| ISC | 0.6627 ±0.0071 | 0.6703 ±0.0371 | 0.5333 ±0.0196 | 0.6074 ±0.0156 | 0.6184 ±0.0501 | 0.6098 ±0.0145 | 0.7500 ±0.0196 |
| LAFB | 0.5610 ±0.0193 | 0.5866 ±0.0204 | 0.5485 ±0.0207 | 0.5776 ±0.0213 | 0.5917 ±0.0233 | 0.5766 ±0.0041 | 0.6413 ±0.0115 |
| LVH | 0.6392 ±0.0189 | 0.6650 ±0.0227 | 0.5325 ±0.1267 | 0.6147 ±0.0106 | 0.6865 ±0.0385 | 0.5867 ±0.0330 | 0.7338 ±0.0175 |
| PAC | 0.5873 ±0.0697 | 0.6122 ±0.0438 | 0.1311 ±0.0162 | 0.2477 ±0.1704 | 0.2188 ±0.0296 | 0.2001 ±0.0124 | 0.7443 ±0.0434 |
| PVC | 0.8990 ±0.0111 | 0.8841 ±0.0185 | 0.7367 ±0.0201 | 0.8427 ±0.0255 | 0.8718 ±0.0375 | 0.7591 ±0.0112 | 0.9444 ±0.0072 |
| STD | 0.4274 ±0.0229 | 0.4810 ±0.0193 | 0.3763 ±0.0207 | 0.4454 ±0.0237 | 0.5011 ±0.0490 | 0.4399 ±0.0185 | 0.6160 ±0.0411 |
| Mean | 0.7175 | 0.7214 | 0.5740 | 0.6523 | 0.6923 | 0.6447 | 0.7906 |
| Avg. rank | 3.38 | 3.08 | 6.69 | 4.85 | 3.15 | 5.62 | 1.23 |

## P.3 DISCUSSION

These 12-lead results support the argument in the main text: both Real-world and SimECG pretraining provide substantial gains over end-to-end training, and the relative ranking between the methods is largely preserved when moving from single-lead (lead II) to the clinically standard 12-lead setting. In particular, the gains from pretraining are most pronounced for difficult diagnostic labels such as AF, PAC, and STD, indicating that the learned representations transfer well even when the input dimensionality and inter-lead dependencies become more complex.

Table 40: $F_1$-scores for classification on the PTB-XL dataset using lead V2 (P.T. = pretraining). Text colors indicate relative performance: Red denotes the best performance, while blue indicates the second-best performance for each task.

| | GRU | LSTM | Luna | MEGA | ResNet18 | Transformer | |
|---|---|---|---|---|---|---|---|
| P.T. | - | - | - | - | - | - | SimECG-N |
| ABQRS | 0.7424 ±0.0106 | 0.7395 ±0.0075 | 0.7027 ±0.0062 | 0.7322 ±0.0106 | 0.7410 ±0.0094 | 0.7307 ±0.0065 | 0.7470 ±0.0151 |
| AF | 0.9472 ±0.0131 | 0.9286 ±0.0267 | 0.8170 ±0.0080 | 0.9189 ±0.0139 | 0.9406 ±0.0183 | 0.8088 ±0.0234 | 0.9752 ±0.0039 |
| ASMI | 0.8948 ±0.0110 | 0.8858 ±0.0250 | 0.8215 ±0.0643 | 0.8583 ±0.0196 | 0.8906 ±0.0112 | 0.8927 ±0.0205 | 0.9324 ±0.0064 |
| CRBBB | 0.9014 ±0.0375 | 0.8770 ±0.0489 | 0.8263 ±0.0335 | 0.8289 ±0.0397 | 0.9625 ±0.0058 | 0.8745 ±0.0217 | 0.9662 ±0.0061 |
| IAVB | 0.7516 ±0.0130 | 0.7413 ±0.0287 | 0.6137 ±0.0432 | 0.6312 ±0.0203 | 0.7585 ±0.0357 | 0.6323 ±0.0383 | 0.8674 ±0.0214 |
| IMI | 0.5449 ±0.0224 | 0.5286 ±0.0116 | 0.3930 ±0.1321 | 0.3906 ±0.0254 | 0.5547 ±0.0536 | 0.5010 ±0.0237 | 0.6422 ±0.0235 |
| IRBBB | 0.5969 ±0.0148 | 0.5522 ±0.0345 | 0.5182 ±0.0333 | 0.5425 ±0.0201 | 0.6329 ±0.0742 | 0.5608 ±0.0128 | 0.6886 ±0.0067 |
| ISC | 0.5925 ±0.0226 | 0.6313 ±0.0302 | 0.5131 ±0.0272 | 0.5653 ±0.0253 | 0.7145 ±0.0168 | 0.6053 ±0.0273 | 0.7283 ±0.0300 |
| LAFB | 0.5932 ±0.0143 | 0.5987 ±0.0113 | 0.5312 ±0.0290 | 0.5845 ±0.0142 | 0.6216 ±0.0402 | 0.5830 ±0.0098 | 0.6646 ±0.0078 |
| LVH | 0.5273 ±0.0381 | 0.5829 ±0.0222 | 0.5394 ±0.0258 | 0.5228 ±0.0252 | 0.5983 ±0.0493 | 0.5372 ±0.0267 | 0.6430 ±0.0393 |
| PAC | 0.6431 ±0.0564 | 0.5451 ±0.1943 | 0.2168 ±0.0379 | 0.6219 ±0.0762 | 0.3946 ±0.1426 | 0.2575 ±0.0172 | 0.7702 ±0.0322 |
| PVC | 0.9038 ±0.0086 | 0.8466 ±0.0591 | 0.7292 ±0.0222 | 0.8746 ±0.0096 | 0.9095 ±0.0212 | 0.7778 ±0.0608 | 0.9509 ±0.0034 |
| STD | 0.4847 ±0.0159 | 0.4767 ±0.0325 | 0.3542 ±0.0450 | 0.4764 ±0.0075 | 0.4484 ±0.0486 | 0.4898 ±0.0232 | 0.6261 ±0.0155 |
| Mean | 0.7018 | 0.6873 | 0.5828 | 0.6575 | 0.7052 | 0.6347 | 0.7848 |
| Avg. rank | 3.15 | 4.00 | 6.62 | 5.46 | 2.85 | 4.92 | 1.00 |

Table 41: $F_1$-scores for classification on the PTB-XL dataset using lead V3 (P.T. = pretraining). Text colors indicate relative performance: Red denotes the best performance, while blue indicates the second-best performance for each task.

| | GRU | LSTM | Luna | MEGA | ResNet18 | Transformer | |
|---|---|---|---|---|---|---|---|
| P.T. | - | - | - | - | - | - | SimECG-N |
| ABQRS | 0.7278 | 0.7257 | 0.6908 | 0.7202 | 0.7227 | 0.7091 | 0.7272 |
| | ±0.0104 | ±0.0112 | ±0.0046 | ±0.0056 | ±0.0132 | ±0.0107 | ±0.0187 |
| AF | 0.9580 | 0.9498 | 0.8707 | 0.9174 | 0.9524 | 0.8852 | 0.9765 |
| | ±0.0132 | ±0.0133 | ±0.0244 | ±0.0170 | ±0.0096 | ±0.0301 | ±0.0037 |
| ASMI | 0.8800 | 0.8680 | 0.8566 | 0.8491 | 0.8789 | 0.8821 | 0.9108 |
| | ±0.0245 | ±0.0215 | ±0.0074 | ±0.0125 | ±0.0290 | ±0.0025 | ±0.0070 |
| CRBBB | 0.8187 | 0.8479 | 0.8124 | 0.8273 | 0.8916 | 0.8153 | 0.9469 |
| | ±0.0554 | ±0.0546 | ±0.0463 | ±0.0485 | ±0.0863 | ±0.0998 | ±0.0070 |
| IAVB | 0.7767 | 0.7523 | 0.6787 | 0.6975 | 0.8123 | 0.7212 | 0.8651 |
| | ±0.0296 | ±0.0349 | ±0.0353 | ±0.0301 | ±0.0278 | ±0.0188 | ±0.0095 |
| IMI | 0.5319 | 0.5461 | 0.5002 | 0.4863 | 0.5525 | 0.5010 | 0.6427 |
| | ±0.0180 | ±0.0153 | ±0.0267 | ±0.0475 | ±0.0449 | ±0.0147 | ±0.0259 |
| IRBBB | 0.4634 | 0.4193 | 0.3961 | 0.4167 | 0.5145 | 0.4277 | 0.5278 |
| | ±0.0093 | ±0.0081 | ±0.0453 | ±0.0179 | ±0.0558 | ±0.0160 | ±0.0299 |
| ISC | 0.6657 | 0.6936 | 0.6412 | 0.6300 | 0.7200 | 0.6468 | 0.7668 |
| | ±0.0201 | ±0.0170 | ±0.0295 | ±0.0278 | ±0.0265 | ±0.0149 | ±0.0166 |
| LAFB | 0.5871 | 0.5989 | 0.6119 | 0.6110 | 0.5903 | 0.6137 | 0.6648 |
| | ±0.0198 | ±0.0135 | ±0.0114 | ±0.0144 | ±0.0582 | ±0.0054 | ±0.0200 |
| LVH | 0.6059 | 0.6318 | 0.6058 | 0.5639 | 0.6526 | 0.6168 | 0.7141 |
| | ±0.0223 | ±0.0297 | ±0.0266 | ±0.0288 | ±0.0378 | ±0.0179 | ±0.0090 |
| PAC | 0.6469 | 0.6442 | 0.2632 | 0.5486 | 0.3805 | 0.2885 | 0.7556 |
| | ±0.0392 | ±0.0532 | ±0.0531 | ±0.1206 | ±0.1247 | ±0.0458 | ±0.0312 |
| PVC | 0.9060 | 0.9033 | 0.7904 | 0.8942 | 0.9276 | 0.8055 | 0.9464 |
| | ±0.0170 | ±0.0172 | ±0.0318 | ±0.0155 | ±0.0127 | ±0.0545 | ±0.0056 |
| STD | 0.5597 | 0.5856 | 0.5585 | 0.5431 | 0.5508 | 0.5636 | 0.7054 |
| | ±0.0179 | ±0.0235 | ±0.0279 | ±0.0113 | ±0.0319 | ±0.0185 | ±0.0157 |
| Mean | 0.7021 | 0.7051 | 0.6367 | 0.6696 | 0.7036 | 0.6520 | 0.7808 |
| Avg. rank | 3.54 | 3.62 | 6.23 | 5.69 | 3.23 | 4.62 | 1.08 |

Table 42: $F_1$-scores for classification on the PTB-XL dataset using lead V4 (P.T. = pretraining). Text colors indicate relative performance: Red denotes the best performance, while blue indicates the second-best performance for each task.

| | GRU | LSTM | Luna | MEGA | ResNet18 | Transformer | |
| --- | --- | --- | --- | --- | --- | --- | --- |
| P.T. | - | - | - | - | - | - | SimECG-N |
| ABQRS | 0.7437 ±0.0070 | 0.7489 ±0.0059 | 0.7032 ±0.0090 | 0.7288 ±0.0057 | 0.7487 ±0.0062 | 0.7351 ±0.0063 | 0.7360 ±0.0127 |
| AF | 0.9426 ±0.0240 | 0.9537 ±0.0164 | 0.8969 ±0.0130 | 0.9447 ±0.0068 | 0.9520 ±0.0132 | 0.9010 ±0.0143 | 0.9812 ±0.0045 |
| ASMI | 0.8179 ±0.0115 | 0.8129 ±0.0190 | 0.8016 ±0.0077 | 0.7819 ±0.0137 | 0.8330 ±0.0450 | 0.8105 ±0.0086 | 0.8549 ±0.0059 |
| CRBBB | 0.8746 ±0.0263 | 0.8699 ±0.0350 | 0.7839 ±0.0343 | 0.8147 ±0.0488 | 0.9211 ±0.0291 | 0.7792 ±0.0689 | 0.9478 ±0.0089 |
| IAVB | 0.7913 ±0.0266 | 0.7824 ±0.0360 | 0.7328 ±0.0234 | 0.6879 ±0.0432 | 0.8003 ±0.0325 | 0.7461 ±0.0150 | 0.8716 ±0.0050 |
| IMI | 0.6189 ±0.0268 | 0.5922 ±0.0207 | 0.5864 ±0.0067 | 0.5841 ±0.0346 | 0.6327 ±0.0398 | 0.6140 ±0.0263 | 0.6907 ±0.0203 |
| IRBBB | 0.4719 ±0.0076 | 0.4685 ±0.0149 | 0.4010 ±0.0298 | 0.4410 ±0.0107 | 0.4165 ±0.0938 | 0.4314 ±0.0230 | 0.5538 ±0.0290 |
| ISC | 0.7910 ±0.0235 | 0.7849 ±0.0375 | 0.7629 ±0.0238 | 0.7677 ±0.0062 | 0.8102 ±0.0139 | 0.7351 ±0.0912 | 0.8570 ±0.0093 |
| LAFB | 0.6992 ±0.0143 | 0.7080 ±0.0172 | 0.7037 ±0.0156 | 0.7130 ±0.0204 | 0.7051 ±0.0466 | 0.7202 ±0.0136 | 0.7647 ±0.0152 |
| LVH | 0.7401 ±0.0540 | 0.7609 ±0.0202 | 0.6943 ±0.0340 | 0.6909 ±0.0199 | 0.7472 ±0.0637 | 0.7008 ±0.0446 | 0.7969 ±0.0081 |
| PAC | 0.6722 ±0.0265 | 0.6202 ±0.0529 | 0.2683 ±0.0163 | 0.6181 ±0.0444 | 0.4000 ±0.0569 | 0.2767 ±0.0109 | 0.7877 ±0.0234 |
| PVC | 0.9221 ±0.0126 | 0.9025 ±0.0133 | 0.7919 ±0.0079 | 0.8492 ±0.0206 | 0.9237 ±0.0051 | 0.8464 ±0.0352 | 0.9439 ±0.0048 |
| STD | 0.6401 ±0.0188 | 0.6428 ±0.0398 | 0.6158 ±0.0307 | 0.6133 ±0.0217 | 0.6598 ±0.0465 | 0.6376 ±0.0274 | 0.7742 ±0.0087 |
| Mean | 0.7481 | 0.7421 | 0.6725 | 0.7104 | 0.7346 | 0.6872 | 0.8123 |
| Avg. rank | 3.46 | 3.31 | 6.38 | 5.46 | 2.92 | 5.23 | 1.23 |

Table 43: $F_1$-scores for classification on the PTB-XL dataset using lead V5 (P.T. = pretraining). Text colors indicate relative performance: Red denotes the best performance, while blue indicates the second-best performance for each task.

| | GRU | LSTM | Luna | MEGA | ResNet18 | Transformer | |
|---|---|---|---|---|---|---|---|
| P.T. | - | - | - | - | - | - | SimECG-N |
| ABQRS | 0.7818 | 0.7783 | 0.7412 | 0.7506 | 0.7714 | 0.7505 | 0.7741 |
| | ±0.0039 | ±0.0096 | ±0.0081 | ±0.0094 | ±0.0110 | ±0.0049 | ±0.0033 |
| AF | 0.9625 | 0.9518 | 0.8836 | 0.9455 | 0.9334 | 0.9025 | 0.9824 |
| | ±0.0139 | ±0.0108 | ±0.0253 | ±0.0110 | ±0.0141 | ±0.0145 | ±0.0027 |
| CRBBB | 0.8342 | 0.8535 | 0.7782 | 0.8166 | 0.9121 | 0.8244 | 0.9625 |
| | ±0.0984 | ±0.0660 | ±0.0450 | ±0.0556 | ±0.0154 | ±0.0496 | ±0.0061 |
| IAVB | 0.7566 | 0.7721 | 0.7065 | 0.7380 | 0.8153 | 0.7855 | 0.8589 |
| | ±0.0547 | ±0.0192 | ±0.0237 | ±0.0237 | ±0.0239 | ±0.0167 | ±0.0134 |
| IMI | 0.6801 | 0.6793 | 0.5596 | 0.6365 | 0.6757 | 0.6409 | 0.7493 |
| | ±0.0313 | ±0.0390 | ±0.1708 | ±0.0284 | ±0.0343 | ±0.0163 | ±0.0084 |
| IRBBB | 0.4824 | 0.4927 | 0.4563 | 0.4686 | 0.4805 | 0.4681 | 0.5421 |
| | ±0.0234 | ±0.0150 | ±0.0105 | ±0.0210 | ±0.0647 | ±0.0090 | ±0.0236 |
| LAFB | 0.7536 | 0.7751 | 0.7669 | 0.7735 | 0.7702 | 0.7763 | 0.8177 |
| | ±0.0261 | ±0.0125 | ±0.0128 | ±0.0182 | ±0.0238 | ±0.0149 | ±0.0076 |
| LVH | 0.8123 | 0.8394 | 0.7695 | 0.8020 | 0.7817 | 0.8006 | 0.8799 |
| | ±0.0146 | ±0.0167 | ±0.0238 | ±0.0257 | ±0.0304 | ±0.0250 | ±0.0028 |
| PAC | 0.6237 | 0.6523 | 0.2784 | 0.5836 | 0.3765 | 0.2882 | 0.7750 |
| | ±0.0323 | ±0.0143 | ±0.0134 | ±0.0264 | ±0.0309 | ±0.0211 | ±0.0445 |
| PVC | 0.9148 | 0.9093 | 0.7865 | 0.8999 | 0.9229 | 0.8486 | 0.9472 |
| | ±0.0115 | ±0.0086 | ±0.0230 | ±0.0204 | ±0.0137 | ±0.0118 | ±0.0045 |
| Mean | 0.7602 | 0.7704 | 0.6727 | 0.7415 | 0.7440 | 0.7086 | 0.8289 |
| Avg. rank | 3.30 | 2.80 | 6.90 | 4.90 | 3.90 | 5.00 | 1.20 |

Table 44: $F_1$-scores for classification on the PTB-XL dataset using lead V6 (P.T. = pretraining). Text colors indicate relative performance: Red denotes the best performance, while blue indicates the second-best performance for each task.

| | GRU | LSTM | Luna | MEGA | ResNet18 | Transformer | |
|---|---|---|---|---|---|---|---|
| P.T. | - | - | - | - | - | - | SimECG-N |
| ABQRS | 0.7803 ±0.0042 | 0.7747 ±0.0045 | 0.7512 ±0.0095 | 0.7707 ±0.0056 | 0.7679 ±0.0133 | 0.7640 ±0.0068 | 0.7722 ±0.0158 |
| AF | 0.9602 ±0.0059 | 0.9607 ±0.0058 | 0.9269 ±0.0074 | 0.9344 ±0.0108 | 0.9582 ±0.0026 | 0.9186 ±0.0108 | 0.9824 ±0.0030 |
| ASMI | 0.7917 ±0.0177 | 0.7955 ±0.0093 | 0.5976 ±0.2178 | 0.7568 ±0.0159 | 0.7866 ±0.0290 | 0.7845 ±0.0095 | 0.8191 ±0.0086 |
| CRBBB | 0.8833 ±0.0569 | 0.9033 ±0.0371 | 0.8305 ±0.0488 | 0.8511 ±0.0498 | 0.8905 ±0.0256 | 0.8672 ±0.0310 | 0.9620 ±0.0106 |
| IAVB | 0.7668 ±0.0468 | 0.7401 ±0.0147 | 0.7602 ±0.0275 | 0.7172 ±0.0409 | 0.7868 ±0.0299 | 0.7813 ±0.0166 | 0.8676 ±0.0206 |
| IMI | 0.6891 ±0.0223 | 0.6707 ±0.0231 | 0.5618 ±0.1392 | 0.5397 ±0.2023 | 0.6993 ±0.0246 | 0.6519 ±0.0249 | 0.7484 ±0.0181 |
| IRBBB | 0.5325 ±0.0376 | 0.4744 ±0.0141 | 0.4667 ±0.0125 | 0.4785 ±0.0157 | 0.5086 ±0.0518 | 0.4975 ±0.0189 | 0.5607 ±0.0111 |
| ISC | 0.9182 ±0.0099 | 0.9272 ±0.0163 | 0.9106 ±0.0094 | 0.9058 ±0.0112 | 0.9328 ±0.0174 | 0.9145 ±0.0116 | 0.9495 ±0.0069 |
| LAFB | 0.7874 ±0.0210 | 0.7893 ±0.0117 | 0.7888 ±0.0137 | 0.7897 ±0.0227 | 0.7704 ±0.0292 | 0.7888 ±0.0119 | 0.8357 ±0.0105 |
| LVH | 0.8254 ±0.0299 | 0.8369 ±0.0093 | 0.8161 ±0.0140 | 0.8304 ±0.0261 | 0.7925 ±0.0459 | 0.8152 ±0.0226 | 0.8936 ±0.0077 |
| PAC | 0.6984 ±0.0238 | 0.6470 ±0.0606 | 0.2525 ±0.0275 | 0.5655 ±0.0193 | 0.3310 ±0.0626 | 0.2761 ±0.0193 | 0.7708 ±0.0518 |
| PVC | 0.9035 ±0.0160 | 0.9180 ±0.0160 | 0.7778 ±0.0254 | 0.8954 ±0.0161 | 0.8965 ±0.0207 | 0.8464 ±0.0326 | 0.9410 ±0.0054 |
| STD | 0.7196 ±0.0315 | 0.7189 ±0.0251 | 0.6806 ±0.0188 | 0.6768 ±0.0076 | 0.7306 ±0.0203 | 0.7364 ±0.0202 | 0.8410 ±0.0044 |
| Mean | 0.7890 | 0.7813 | 0.7016 | 0.7471 | 0.7578 | 0.7417 | 0.8418 |
| Avg. rank | 3.23 | 3.23 | 6.15 | 5.23 | 3.85 | 4.92 | 1.15 |

Table 45: Comparison of pretraining methods on 12-lead ECG classification measured with $F_1$ (mean ± standard deviation over runs).

| | Real-world | SimECG-N |
|---|---|---|
| ABQRS | 0.8644 ± 0.0065 | 0.8644 ± 0.0030 |
| AF | 0.9871 ± 0.0026 | 0.9833 ± 0.0028 |
| ASMI | 0.9581 ± 0.0054 | 0.9532 ± 0.0030 |
| CRBBB | 0.9880 ± 0.0094 | 0.9842 ± 0.0064 |
| IAVB | 0.8962 ± 0.0043 | 0.8833 ± 0.0105 |
| IMI | 0.9078 ± 0.0140 | 0.9171 ± 0.0089 |
| IRBBB | 0.7905 ± 0.0154 | 0.8152 ± 0.0105 |
| ISC | 0.9618 ± 0.0065 | 0.9605 ± 0.0035 |
| LAFB | 0.9394 ± 0.0055 | 0.9391 ± 0.0076 |
| LVH | 0.8990 ± 0.0043 | 0.8938 ± 0.0060 |
| PAC | 0.8064 ± 0.0199 | 0.8021 ± 0.0243 |
| PVC | 0.9607 ± 0.0038 | 0.9592 ± 0.0053 |
| STD | 0.8832 ± 0.0071 | 0.8725 ± 0.0188 |
| Mean | 0.9110 | 0.9098 |

Table 46: Per-label 12-lead ECG classification performance measured with $F_1$ (mean $\pm$ standard deviation over runs). We report the same evaluation metric as in Table 21 in the main text for 13 diagnostic labels (AF, ASMI, ABQRS, CRBBB, IMI, IRBBB, ISC, LAFB, LVH, PAC, PVC, STD, IAVB). Text colors indicate relative performance: Red denotes the best, while blue indicates the second-best for each task. P.T. = Pretraining. Rel. score = Relative score.

| | GRU | LSTM | Luna | MEGA | ResNet18 | Transformer | |
| P.T. | - | - | - | - | - | - | SimECG-N |
| --- | --- | --- | --- | --- | --- | --- | --- |
| ABQRS | 0.8660 ±0.0070 | 0.8696 ±0.0031 | 0.8503 ±0.0056 | 0.8680 ±0.0036 | 0.8549 ±0.0195 | 0.8645 ±0.0061 | 0.8644 ±0.0030 |
| AF | 0.9660 ±0.0070 | 0.9667 ±0.0062 | 0.9607 ±0.0056 | 0.9658 ±0.0044 | 0.9595 ±0.0040 | 0.9666 ±0.0052 | 0.9833 ±0.0028 |
| ASMI | 0.9461 ±0.0082 | 0.9380 ±0.0044 | 0.9290 ±0.0072 | 0.9371 ±0.0015 | 0.9375 ±0.0072 | 0.9434 ±0.0034 | 0.9532 ±0.0030 |
| CRBBB | 0.9870 ±0.0067 | 0.9870 ±0.0102 | 0.9853 ±0.0060 | 0.9963 ±0.0055 | 0.9917 ±0.0035 | 0.9925 ±0.0083 | 0.9842 ±0.0064 |
| IAVB | 0.8760 ±0.0061 | 0.8771 ±0.0051 | 0.8421 ±0.0130 | 0.8694 ±0.0140 | 0.8505 ±0.0096 | 0.8639 ±0.0071 | 0.8833 ±0.0105 |
| IMI | 0.9168 ±0.0144 | 0.9029 ±0.0145 | 0.8850 ±0.0065 | 0.9080 ±0.0155 | 0.9052 ±0.0083 | 0.9086 ±0.0086 | 0.9171 ±0.0089 |
| IRBBB | 0.8227 ±0.0074 | 0.8155 ±0.0109 | 0.7708 ±0.0053 | 0.8100 ±0.0015 | 0.8049 ±0.0149 | 0.8095 ±0.0109 | 0.8152 ±0.0105 |
| ISC | 0.9507 ±0.0030 | 0.9464 ±0.0073 | 0.9435 ±0.0037 | 0.9472 ±0.0043 | 0.9315 ±0.0055 | 0.9481 ±0.0057 | 0.9605 ±0.0035 |
| LAFB | 0.9429 ±0.0041 | 0.9429 ±0.0049 | 0.9336 ±0.0048 | 0.9367 ±0.0023 | 0.9515 ±0.0053 | 0.9429 ±0.0034 | 0.9391 ±0.0076 |
| LVH | 0.8846 ±0.0053 | 0.8825 ±0.0026 | 0.8700 ±0.0071 | 0.8868 ±0.0093 | 0.8606 ±0.0085 | 0.8799 ±0.0048 | 0.8938 ±0.0060 |
| PAC | 0.6845 ±0.0070 | 0.5147 ±0.0174 | 0.4978 ±0.0160 | 0.4580 ±0.0213 | 0.3829 ±0.0717 | 0.5054 ±0.0361 | 0.8021 ±0.0243 |
| PVC | 0.9450 ±0.0065 | 0.9283 ±0.0084 | 0.9234 ±0.0098 | 0.9014 ±0.0124 | 0.8856 ±0.0142 | 0.9257 ±0.0129 | 0.9592 ±0.0053 |
| STD | 0.8514 ±0.0102 | 0.8414 ±0.0130 | 0.8238 ±0.0108 | 0.8288 ±0.0096 | 0.7864 ±0.0183 | 0.8270 ±0.0073 | 0.8725 ±0.0188 |
| Mean | 0.8958 | 0.8779 | 0.8627 | 0.8703 | 0.8541 | 0.8752 | 0.9098 |
| Rel. score | - | −1.9998 | −3.6947 | −2.8542 | −4.6643 | −2.3004 | 1.5628 |