# OpenReview forum: "Simulator‑Based Synthetic ECGs for Self-Supervised Pretraining"
_ICLR.cc/2026/Conference — Submitted to ICLR 2026_

### Official Review · Reviewer_eCu6 · 2025-10-15

**Soundness:** 3
**Presentation:** 3
**Contribution:** 2
**Rating:** 4
**Confidence:** 4

**Summary:**

In their submission, the authors investigate the prospects of using synthetic single-lead ECG samples for pretraining ECG classification models through self-supervised pretraining (masked auto-encoding, but also other approaches are investigated). They consider different sources of synthetic models from knowledge-based simulator ECGs to ECG generated via generative models. They probe the pretrained models through finetuning on different binary classification tasks (normal vs. x) and find indications for robust improvements through self-supervised pretraining in comparison to training from scratch. The authors provide insights into lead-dependence and fairness aspects by considering different patient subgroups.

**Strengths:**

* The authors cover a comprehensive set of experimental conditions, ranging from two different categories of synthetic samples with two approaches each, over different self-supervised learning algorithms to three downstream tasks covering several conditions each along with a diverse set of supervised baselines.
* The authors identify settings (MAE with pretraining on SimECG) that show robust performance across several datasets
* The authors present ablation studies to underline the robustness of their findings (varying ECG leads, fairness evaluation)

**Weaknesses:**

* The motivation of using ECG as a test-case for other domains does not seem convincing since the insights from this work are highly ECG-specific. And in the ECG domain there is no shortage of publicly available unlabeled data, as the authors acknowledge themselves, while not even mentioning HEEDB as the currently largest ECG dataset with more than 10M samples as well as CODE-15. Nevertheless, an approach as proposed by the authors could be interesting for other domains where also unlabeled data is very scarce such as invasive data e.g. intracranial EEGs etc.
* The related work misses essentially all works works on ECG foundation models, such as [1][2] and many more. While this just misrepresents the research landscape, the omission of essentially all modern diffusion-based generative models for ECG data, see e.g. [3],[4] among many others,  is a more severe issue as it also impacts the choice of generative models for the later experiments. This raises doubts about the quality of the generated samples and questions the conclusion about the superiority of knowledge-based approaches.
* The authors consider a very special setup (single-lead input) and investigate artificial normal vs. diseases settings. The problem with such binary setups is that if they are not carefully balanced according to other covariates that it will often be very simple for the model to distinguish cases based on characteristics that have nothing to do with the actual disease label under consideration.
* The authors provide only relative comparisons to own baselines. From my point of view it would raise the impact of their contribution considerably if they would show results on established benchmarks, be it single-lead or 12-lead, where benchmarking results are known.

[1] Li, J., Aguirre, A. D., Junior, V. M., Jin, J., Liu, C., Zhong, L., ... & Hong, S. (2025). An Electrocardiogram Foundation Model Built on over 10 Million Recordings. NEJM AI, 2(7), AIoa2401033.
[2] McKeen, K., Masood, S., Toma, A., Rubin, B., & Wang, B. (2024). Ecg-fm: An open electrocardiogram foundation model. arXiv preprint arXiv:2408.05178.
[3] Lai, Y., Liu, B., Guan, X., Zhao, Q., Li, H., & Hong, S. (2025). ECGTwin: Personalized ECG Generation Using Controllable Diffusion Model. arXiv preprint arXiv:2508.02720.
[4] Alcaraz, J. M. L., & Strodthoff, N. (2023). Diffusion-based conditional ECG generation with structured state space models. Computers in biology and medicine, 163, 107115.

**Questions:**

* Are there any reasons that prevent the authors from considering multi-lead (potentially knowledge-based ECGs are not available for this setting?) and more realistic i.e. non-binary evaluation modes?
* Could the authors comment on their choice of the baseline architecture for the self-supervised pretraining as a transformer model? At least in their supervised experiments it performs poorly in comparison to other model architectures. This gets overcompensated through pretraining, but a different architecture might profit even more from self-supervised pretraining?
* In 5.2, the authors claim to use a per sample standardization. I cannot image that this is really the optimal setting since certain diseases rely on absolute scales e.g. R-amplitudes in LVH as a very simple example. This information gets largely lost through per sample standardization. Did the authors experiment with other setups?

---

> ### Author Response · Authors · 2025-11-21
> **Reply to Reviewer eCu6**
>
> We appreciate the reviewer's insightful comments, which have been highly beneficial to our work. As we are conducting further experiments, we will revise our responses as new results are obtained. At this time, we respond to the questions that we are able to address with the current data.
>
> # Weaknesses
>
> ## Application to other domains
>
> > The motivation of using ECG as a test-case for other domains does not seem convincing since the insights from this work are highly ECG-specific. And in the ECG domain there is no shortage of publicly available unlabeled data, as the authors acknowledge themselves, while not even mentioning HEEDB as the currently largest ECG dataset with more than 10M samples as well as CODE-15. Nevertheless, an approach as proposed by the authors could be interesting for other domains where also unlabeled data is very scarce such as invasive data e.g. intracranial EEGs etc.
>
> We conducted additional experiments using PCG data to examine the applicability to other domains. We pre-trained Transformer models using MAE on two publicly available real-world datasets (PhysioNet2022 and BMD-HS) and synthetic heart sounds, then compared them with baseline models.
> The results (also presented in response to Reviewer ThrN) showed patterns similar to those observed in ECG experiments: pre-trained models outperformed training from scratch, and models pre-trained on synthetic data achieved comparable performance to those pre-trained on real-world data with self-supervised learning.
>
> Table eCu6-T1: Classification result on phonocardiogram (AUROC) | Identical to Table GHNt-T5.
> *Note: We report AUROC for PCG tasks following standard practice in heart sound classification literature, while F1-score is used for ECG tasks as established in PTB-XL and related benchmarks.*
> ||Pretraining Data| AR | AS | MR | Average |
> |---|---|---|---|---|---|
> | GRU | - |0.5471 ± 0.0652 | 0.5508 ± 0.1291|0.4110 ± 0.2647 |	0.5030 |
> |LSTM |-|0.4424 ± 0.2292 | 0.5942 ± 0.1069	| **0.5989 ± 0.0352** |0.5452 |
> | LUNA |-| 0.4327 ± 0.2167|0.3705 ± 0.1712|0.4255 ± 0.1940|0.4096|
> | MEGA |-|0.5467 ± 0.0556|0.5495 ± 0.0984|0.5616 ± 0.0282|0.5526|
> |ResNet18| -|0.4756 ± 0.0756 |0.3373 ± 0.0812|0.4042 ± 0.0827|0.4057|
> |Transformer|-|0.5486 ± 0.0356|0.4449 ± 0.2417|0.5407 ± 0.0367|0.5114|
> |Transformer|PhysioNet2022|0.6151 ± 0.0666|0.5851 ± 0.0644|0.5741 ± 0.0393|0.5914|
> |Transformer|BMD-HS|0.6273 ± 0.0176|0.5856 ± 0.0787|0.5774 ± 0.0500|0.5968|
> |Transformer|SimPCG|**0.6295 ± 0.0538**|**0.6356 ± 0.0909**|0.5786 ± 0.0590|**0.6146**|
>
> ## Missing related works
>
> > The related work misses essentially all works works on ECG foundation models, such as [1][2] and many more. While this just misrepresents the research landscape, the omission of essentially all modern diffusion-based generative models for ECG data, see e.g. [3],[4] among many others, is a more severe issue as it also impacts the choice of generative models for the later experiments. This raises doubts about the quality of the generated samples and questions the conclusion about the superiority of knowledge-based approaches.
>
> Thank you for your suggestion. We have added all the papers listed here to the related work section.

---

> > ### Author Response · Authors · 2025-11-27
> > **Reply to Reviewer eCu6 (2)**
> >
> > ## Comparison with established benchmarks
> >
> > > The authors provide only relative comparisons to own baselines. From my point of view it would raise the impact of their contribution considerably if they would show results on established benchmarks, be it single-lead or 12-lead, where benchmarking results are known.
> >
> > Thank you for this valuable feedback. We agree that comparison with established benchmarks would significantly strengthen our contribution.
> >
> > **Evaluation on PTB-XL Benchmark Tasks:**
> >
> > Following your suggestion, we conducted experiments on the established PTB-XL benchmark tasks defined in Strodthoff et al. [eCu6-c1].
> > While the original benchmark uses all 12 leads, we report results using only lead II—a clinically important lead commonly used in ambulatory monitoring and wearable devices.
> > We use the same evaluation protocol (fold structure, macro AUROC metric) to ensure comparability.
> >
> > **Key Findings:**
> >
> > Our results (Table eCu6-T2) demonstrate:
> >
> > 1. **Validation of our approach on established tasks:**
> > Our SimECG-pretrained Transformer achieves competitive performance (macro AUROC: 0.8280 on the "all" task), approaching the performance of real-data pretraining on PTB-XL (0.8377) and consistently outperforming training from scratch across all diagnostic categories.
> >
> > 2. **Consistent improvement across diverse architectures:**
> > The benefit of pretraining holds across various baseline architectures (GRU, LSTM, MEGA, LUNA, Transformer), with pretrained models showing improvements particularly in the "form" and "diagnostic" tasks.
> >
> > 3. **Performance characteristics of single-lead analysis:**
> > As expected, single-lead performance  (macro AUROC: 0.8280-0.8377) is lower than 12-lead results reported in [eCu6-c1] (e.g., 0.925 for xresnet1d101 [eCu6-c1]).
> > This reflects the inherent information loss when using 1/12 of the available signals. Lead-specific analysis (Table 4 in original manuscript) reveals that certain conditions (e.g., IRBBB) show lower performance with lead II across all methods, as these conditions are better characterized by other leads (e.g., V1 for right ventricular activity).
> >
> > 3. **Task-specific observations:**
> > Our approach shows particularly strong performance on rhythm-related tasks (macro AUROC: 0.9568 with SimECG), approaching the performance of models trained with real data pretraining (0.9612).
> >
> >
> > Table eCu6-T2: Performance on PTBXL benchmark tasks.
> > ||Pre-training data| Lead | all | diag. | sub-diag. | super-diag. | form | rhythm |
> > |---|---|---|---|---|---|---|---|---|
> > |GRU| - | II |0.8167 | 0.7939 | 0.8073 | 0.8392 | 0.6331 | 0.9063 |
> > |LSTM| - | II | 0.8234 | 0.7940 | 0.8132 | 0.8474 | 0.7678 | 0.9512 |
> > |MEGA| - | II | 0.8000 | 0.8053 | 0.7948 | 0.8367 | 0.7239 | 0.8787 |
> > |LUNA| - | II | 0.8249 | 0.8201 | 0.8241 | 0.8512 | 0.7776 | 0.9496 |
> > |Transformer| - | II | 0.8062 | 0.8095 | 0.8381 | 0.8408 | 0.7128 | 0.8838 |
> > |Transformer| PTBXL | II | 0.8377 | 0.8457 | 0.8428 | 0.8533 | 0.7979 | 0.9612 |
> > |Transformer| SimECG | II | 0.8280 | 0.8262 | 0.8426 | 0.8535 | 0.7838 | 0.9568 |
> > |---|---|---|---|---|---|---|---|---|
> > |xresnet1d101 [from eCu6-c1]| - | all | 0.925 | 0.939 | 0.933 | 0.934 | 0.907 | 0.965 |
> >
> >
> > [eCu6-c1] Strodthoff, Nils, et al. "Deep learning for ECG analysis: Benchmarks and insights from PTB-XL." IEEE journal of biomedical and health informatics 25.5 (2020): 1519-1528.

---

> > > ### Comment · Reviewer_eCu6 · 2025-11-27
> > > **Comment on authors' response**
> > >
> > > I appreciate the authors' response and their willingness to provide additional experiments to improve the paper. My view on the authors' response is the following:
> > > * The additional experiments on PCG do not fix the central point of criticism: Also PCG is not an example where unlabeled data is scarce. This does not really support the narrative of the paper.
> > > * The authors did not post an updated paper, therefore I cannot comment on the updated references, just would like to stress that the mentioned works were just meant as pointers to the vast amount of literature- the point is not about citing precisely these, but to make it a balanced literature review. More importantly, the submission fails to convey in how far the approach is superior other self-supervised pretraining approaches/foundation models.
> > > * The PTB-XL benchmark is a step in the right direction. However, I doubt that the presented supervised models are competitive. It would be insightful to see CNN-based baselines. In several cases the pretrained model fails to beat the strongest supervised model. From my perspective, this fails to convey the added value of the proposed approach.
> > > * The author did not attempt to answer the questions concerning multi-channel input and the choice of the transformer architecture.
> > >
> > > Unless the final revised paper goes significantly beyond the material described in the responses, I do not feel inclined to modify my score.

---

> ### Author Response · Authors · 2025-12-01
> **Reply to Reviewer eCu6-2**
>
> We sincerely thank the reviewer for the careful consideration of our responses and for providing this additional feedback.
> We apologize that the updated PDF was not available at the time of your review - we have now posted the revised manuscript incorporating the additional experiments and references discussed in our responses.
>
> ## On the strength of supervised baselines on PTB-XL
>
> > The PTB-XL benchmark is a step in the right direction. However, I doubt that the presented supervised models are competitive. It would be insightful to see CNN-based baselines. In several cases the pretrained model fails to beat the strongest supervised model. From my perspective, this fails to convey the added value of the proposed approach.
>
> We thank the reviewer for pointing out that our supervised baselines on PTB-XL might not be competitive and for suggesting CNN-based baselines.
> To address this, we added three additional supervised CNN baselines implemented with the fastai library: XResNet1D‑101, XceptionTime, and XResNet1D‑152.
> These architectures are representative strong 1D CNN models for ECG and time‑series classification and are closely related to the CNN families used in prior PTB‑XL benchmark studies (XResNet1D-101 shows the best average performance in [eCu6-c1]).
> All models were trained within our unified 500 Hz pipeline using the same input representation, data augmentations, and optimization settings as for SimECG, to ensure a fair comparison.
> The best supervised CNN baseline, XResNet1D‑101, indeed improves over our previously reported supervised models. However, the SimECG‑pretrained model still outperforms this strong baseline on 4 out of 6 tasks (see new Table eCu6-T3), which demonstrates that pretraining brings a consistent benefit over competitive purely supervised CNNs.
>
> Table eCu6-T3: Performance on PTBXL benchmark tasks (extended from Table eCu6-T2).
> | |Pre-training data|Lead|all|diag.|sub-diag.|super-diag.|form|rhythm|
> |---|---|---|---|---|---|---|---|---|
> |GRU|-|II|0.8167|0.7939|0.8073|0.8392|0.6331|0.9063|
> |LSTM|-|II|0.8234|0.7940|0.8132|0.8474|0.7678|0.9512|
> |MEGA|-|II|0.8000|0.8053|0.7948|0.8367|0.7239|0.8787|
> |LUNA|-|II|0.8249|0.8201|0.8241|0.8512|0.7776|0.9496|
> |ResNet18|-|II|0.8069|0.8248|0.8327|0.8481|0.7454|0.9232|
> |Transformer|-|II|0.8062|0.8095|0.8381|0.8408|0.7128|0.8838|
> |---|---|---|---|---|---|---|---|---|
> |XceptionTime|-|II|0.8034|0.8048|0.8029|0.8526|0.6877|0.9044|
> |XResNet1D101|-|II|0.8294|0.8067|0.8487|0.8495|0.7577|0.9291|
> |XResNet1D152|-|II|0.7901|0.8125|**0.8513**|0.8531|0.7437|0.9296|
> |---|---|---|---|---|---|---|---|---|
> |Transformer|PTBXL|II|**0.8377**|**0.8457**|0.8428|0.8533|**0.7979**|**0.9612**|
> |Transformer|SimECG|II|0.8280|0.8262|0.8426|**0.8535**|0.7838|0.9568|
> |---|---|---|---|---|---|---|---|---|
> |ref. XResNet1D101 [from eCu6-c1]|-|all|0.925|0.939|0.933|0.934|0.907|0.965|
>
>
> **Additional analysis on label efficiency (PTB‑XL)**
>
> We further investigated how the different pretraining strategies behave under limited labeled data on PTB‑XL.
> Specifically, we trained all models on subsampled versions of the training set containing 50%, 25%, 10%, 5%, 2.5%, and 1% of the original data.
> For each fraction, we considered both random and label‑stratified subsampling.
> Across all data fractions, we observed a consistent ranking of pretraining strategies: real‑data pretraining performed best, followed by SimECG pretraining, and finally training from scratch.
> This tendency is robust for both random and stratified subsampling.
> Compared to our binary classification experiments, the performance gaps between pretraining strategies on PTB‑XL are smaller, but pretraining still brings clear gains especially at low data regimes, and SimECG remains competitive with real‑data pretraining.
>
> These results are included in Appendix I.

---

> > ### Author Response · Authors · 2025-12-01
> > **Reply to Reviewer eCu6-2 (2)**
> >
> > # the transformer architecture.
> >
> > > Could the authors comment on their choice of the baseline architecture for the self-supervised pretraining as a transformer model? At least in their supervised experiments it performs poorly in comparison to other model architectures. This gets overcompensated through pretraining, but a different architecture might profit even more from self-supervised pretraining?
> >
> > We appreciate the reviewer's insightful question regarding our choice of Transformer architecture for self-supervised pretraining.
> > We acknowledge that Transformers showed relatively modest performance in our supervised baseline experiments, and we agree this warrants further investigation.
> >
> > ### Choice of Transformer for MAE pretraining:
> >
> > Our choice of Transformer was motivated by two key factors.
> >
> > First, Transformers have become widely adopted in ECG analysis due to their ability to capture long-range temporal dependencies through self-attention mechanisms, which is particularly well-suited for physiological time-series signals.
> > Second, several recent ECG foundation models employ Transformer-based architectures for self-supervised pretraining.
> > Notably, ST-MEM [eCu6-c2] uses a ViT-1D transformer trained as a masked autoencoder to learn spatio-temporal features from 12-lead ECG data, while ECG-FM [eCu6-c2]  adopts a transformer-based architecture pretrained on 2.5 million ECG samples using hybrid contrastive and generative self-supervised learning.
> >
> > These developments demonstrate the suitability of Transformers for large-scale ECG representation learning.
> > Additionally, the MAE pretraining framework [eCu6-c4] was originally designed and validated with Vision Transformers, and we followed this established approach for consistency with the broader self-supervised learning literature.
> >
> > ### Additional experiments with alternative architectures:
> > Following your comment and similar feedback from Reviewer GHNt, we conducted additional experiments applying MAE-style pretraining to the top three non-Transformer baseline models: GRU, LSTM, and ResNet-18.
> > The detailed results are presented in Appendix L.
> >
> > **Key findings:**
> >
> > - **GRU and LSTM** demonstrated substantial performance improvements with MAE-style pretraining, achieving statistically significant gains over training from scratch (p < 0.025, Wilcoxon signed-rank test with Bonferroni correction). GRU improved by +5.74% with SimECG pretraining (0.7854 → 0.8305), while LSTM showed +6.86% improvement (0.7726 → 0.8256).
> >
> > - **ResNet-18** exhibited performance degradation with MAE pretraining (0.7617 → 0.6835), suggesting that patch-based masking strategies are less suitable for CNN architectures in the ECG domain. This degradation likely stems from the loss of local spatial structure that CNNs rely on for feature extraction.
> >
> > - Notably, even with pretraining, neither GRU nor LSTM surpassed the pretrained Transformer's performance (0.8318 with SimECG), reinforcing our choice of Transformer as the primary architecture for this study.
> >
> > These results confirm that SimECG-based pretraining benefits not only Transformers but also sequential architectures, demonstrating the broader applicability of simulator-based synthetic data for self-supervised pretraining.
> >
> > ---
> > [eCu6-c2] Na, Yeongyeon, et al. "Guiding masked representation learning to capture spatio-temporal relationship of electrocardiogram." arXiv preprint arXiv:2402.09450 (2024).
> >
> > [eCu6-c3] McKeen, Kaden, et al. "Ecg-fm: An open electrocardiogram foundation model." JAMIA open 8.5 (2025): ooaf122.
> >
> > [eCu6-c4] He, Kaiming, et al. "Masked autoencoders are scalable vision learners." Proceedings of the IEEE/CVF conference on computer vision and pattern recognition. 2022.

---

> > > ### Author Response · Authors · 2025-12-01
> > > **Reply to Reviewer eCu6-2 (3)**
> > >
> > > # Application beyond ECG and "data scarce" domain
> > >
> > > > The additional experiments on PCG do not fix the central point of criticism: Also PCG is not an example where unlabeled data is scarce. This does not really support the narrative of the paper.
> > >
> > > We appreciate the reviewer’s perspective and agree that there are modalities with even more limited access to unlabeled data, such as intracranial EEG.
> > > Our use of the term “data-scarce,” however, is intended in a relative sense with respect to the scales typically required by modern self-supervised learning methods, rather than as a binary categorization into “scarce” and “not scarce.”
> > >
> > > For PCG, widely used public datasets such as BMD-HS ($\approx$1,000 recordings) and PhysioNet2022 ($\approx$3,000 recordings) together yield on the order of 4,000 to 5000 recordings.
> > > This is roughly three orders of magnitude fewer than the ECG datasets cited by the reviewer (e.g., HEEDB with more than 10 million samples).
> > > Contemporary self-supervised approaches, particularly Transformer-based architectures such as MAE, are typically pre-trained on hundreds of thousands to millions of samples to be effective. From this viewpoint, having access to only a few thousand PCG recordings constitutes a practically data-scarce regime for these methods.
> > >
> > > Our PCG experiments are therefore designed to highlight precisely the setting in which synthetic data is most relevant: domains where some unlabeled data is available, but not at the scale typically assumed for large-scale self-supervised pre-training.
> > > We do not claim that PCG is as data-limited as invasive recordings; instead, we regard it as a realistic intermediate regime between data-rich modalities such as ECG and extremely low-resource invasive measurements.
> > >
> > > Consistent with this interpretation, the PCG experiments qualitatively replicate our main ECG findings: MAE pre-training improves over training from scratch, and pre-training on synthetic data performs comparably to self-supervised pre-training on real data. In the revised manuscript, we now explicitly frame PCG as such an intermediate data regime and clarify this more nuanced positioning of our claims.

---

> > > > ### Author Response · Authors · 2025-12-01
> > > > **Reply to Reviewer eCu6-2 (4)**
> > > >
> > > > # Literature and position of our work
> > > >
> > > > > The authors did not post an updated paper, therefore I cannot comment on the updated references, just would like to stress that the mentioned works were just meant as pointers to the vast amount of literature- the point is not about citing precisely these, but to make it a balanced literature review. More importantly, the submission fails to convey in how far the approach is superior other self-supervised pretraining approaches/foundation models.
> > > >
> > > > We thank the reviewer for the helpful comments. In the revised version (attached), we have substantially revised the Related Work section to provide a more balanced literature review and to better position our contribution relative to existing self-supervised and foundation-model approaches for ECG.
> > > >
> > > > Concretely:
> > > >
> > > > - We reorganized the related work along two main axes:
> > > > **(i) type of pretraining signal** (supervised vs. self-supervised/foundation-style pretraining), and
> > > > **(ii) source of data** (real clinical ECG corpora vs. synthetic data from knowledge-driven simulators vs. data-driven generative models).
> > > > This makes clear how our setting differs from both classic supervised setups and from large-scale foundation models.
> > > >
> > > > - We added and discussed the ECG-specific self-supervised methods and ECG foundation models pointed out by the reviewer, together with additional recent work. The revised Related Works section now explicitly covers contrastive/reconstruction-based ECG SSL (e.g., CLOCS, ST-MEM) as well as ECG foundation models and multimodal time-series foundation models trained on millions of real ECGs and accompanying reports.
> > > >
> > > > - Most importantly, **we clarified the relationship between our approach and these works**. Our central proposal is to use knowledge-driven ECG simulators as the pretraining data source. This is orthogonal to the choice of architecture or self-supervised objective used in prior ECG SSL and foundation models. We now explicitly state that we are not competing with ECG foundation models trained on massive real datasets; instead, we target an intermediate data regime (thousands–tens of thousands of unlabeled recordings) where such large-scale pretraining is not feasible due to privacy and governance constraints.
> > > >
> > > > - To address the question “in how far the approach is superior,” we revised the introduction and discussion to explain more clearly what advantage we claim: in this intermediate regime, pretraining on simulator-generated ECG can replace real-data self-supervised pretraining to a large extent, while avoiding any exposure to identifiable patient records, and still outperforming purely supervised baselines across a wide range of tasks. We also emphasize that, because our contribution is about the data source, it is in principle complementary and combinable with existing ECG self-supervised and foundation-model architectures.
> > > >
> > > > We hope that these changes make both the literature review and the position of our contribution much clearer.

---

> > > > > ### Author Response · Authors · 2025-12-02
> > > > > **Reply to Reviewer eCu6-2 (5)**
> > > > >
> > > > > # Leads and Non-binary classification
> > > > >
> > > > > > Are there any reasons that prevent the authors from considering multi-lead (potentially knowledge-based ECGs are not available for this setting?) and more realistic i.e. non-binary evaluation modes?
> > > > >
> > > > > We appreciate the reviewer’s suggestion to consider multi-lead inputs and more realistic, non-binary evaluation settings.
> > > > >
> > > > > **Multi-lead vs. single-lead.**
> > > > >
> > > > > We focus our pretraining setup on single-lead ECGs (lead II) for two main reasons.
> > > > > First, while 12‑lead ECGs are the standard for diagnostic recordings (e.g., health check-ups), in many practical monitoring scenarios only a single lead is available: bedside monitors in ICUs typically display a single rhythm lead (often lead II or a morphologically similar lead), and most wearable or patch devices also operate in a single-lead configuration.
> > > > > Second, both ECG simulators used in this work [eCu6-c5, c6] are designed and calibrated to generate lead‑II–like signals rather than full 12‑lead recordings, so single-lead pretraining is the natural and technically feasible setting for a fair comparison between real and synthetic data.
> > > > > We now make this rationale explicit in the revised manuscript (Appendix B.5).
> > > > >
> > > > > **Transfer to other leads and 12‑lead ECGs.**
> > > > >
> > > > > Importantly, our conclusions are not tied to a specific lead.
> > > > > In Appendix O, we take the same lead‑II–pretrained encoder and **fine‑tune it on each of the 12 standard leads separately**.
> > > > > Across all leads, the SimECG‑pretrained Transformer achieves the best average rank compared with all supervised baselines and a Transformer trained from scratch, indicating that the learned representation transfers well to non–lead‑II inputs and is robust to electrode placement.
> > > > > Furthermore, in Appendix P we reuse the lead‑II–pretrained encoder **for 12‑lead classification** by wrapping it in a shared‑encoder architecture that processes all 12 leads jointly.
> > > > > In this 12‑lead setting, both real‑data and SimECG pretraining still provide substantial improvements over end‑to‑end 12‑lead training, and the relative ordering between pretraining sources remains almost unchanged.
> > > > > These results suggest that, although the simulators themselves are single‑lead, pretraining on synthetic lead‑II waveforms yields representations that remain effective when extended to multi‑lead inputs.
> > > > >
> > > > > **Non‑binary / multi‑label evaluation.**
> > > > >
> > > > > We agree that non‑binary output spaces are clinically more realistic than one‑vs‑normal tasks.
> > > > > Our main abnormal‑ECG experiments use one‑vs‑normal binary classifiers to harmonize label definitions across PTB‑XL, G12EC, and CPSC.
> > > > > To address the reviewer’s concern, we additionally evaluate our approach on the **standard multi‑label PTB‑XL benchmark** (all, diagnostic, sub‑diagnostic, super‑diagnostic, form, rhythm) in Appendix J (results are also reported in previous post as a Table eCu6-T3).
> > > > > On these multi‑label tasks, both PTB‑XL‑ and SimECG‑pretrained Transformers outperform strong supervised CNN and Transformer baselines, and the performance gap between real‑ and synthetic‑pretrained models is comparable to (or smaller than) that observed in our binary experiments.
> > > > > This shows that the benefits of simulator‑based pretraining extend to more realistic non‑binary evaluation modes, not only to one‑vs‑normal classification.
> > > > >
> > > > > Overall, the revised manuscript now clarifies why we adopt a single‑lead pretraining setup and demonstrates that our conclusions hold in both multi‑lead and multi‑label (non‑binary) evaluation settings.
> > > > >
> > > > > [eCu6-c5]: McSharry, Patrick E., et al. "A dynamical model for generating synthetic electrocardiogram signals." IEEE transactions on biomedical engineering 50.3 (2003): 289-294.;
> > > > >
> > > > > [eCu6-c6]: Nonaka, Naoki, and Jun Seita. "Boosting ECG Classification Performance by Pre-training with Synthesized Data." ECML DEARING 2024;

---

> ### Author Response · Authors · 2025-12-03
> **Summary of replies to Reviewer eCu6**
>
> We sincerely thank you for your detailed, critical, and very helpful comments. They led to substantial revisions and new experiments.
>
> In particular, we:
>
> - Greatly expanded and restructured the related‑work section to provide a more balanced overview of ECG self‑supervised methods, ECG foundation models, and modern diffusion‑based generative models, and to more clearly position our contribution as about the data source for pretraining rather than a new architecture or SSL objective ([Reply to Reviewer eCu6-2 (4)](https://openreview.net/forum?id=gUXKz2PwQb&noteId=nmuLzRgeuU) / Related work-Page 2).
>
> - Added PCG experiments and clarified that we target an intermediate regime where some unlabeled data exist but not at the scale typically assumed for large‑scale SSL, refining our “data‑scarce” narrative ([Reply to Reviewer eCu6-2 (3)](https://openreview.net/forum?id=gUXKz2PwQb&noteId=e1PFYkN2qV)).
>
> - Evaluated on the established PTB‑XL benchmark tasks (multi‑label, not only one‑vs‑normal) and added strong CNN baselines (e.g., XResNet1D‑101/XceptionTime) within our unified pipeline; we show that SimECG‑pretrained models still improve over these competitive supervised baselines on most tasks ([Reply to Reviewer eCu6-2](https://openreview.net/forum?id=gUXKz2PwQb&noteId=PSACpqx8ye) / Appendix J-Page 37).
>
> - Performed MAE‑style pretraining for GRU/LSTM/ResNet‑18, showing that SimECG pretraining helps non‑Transformer architectures as well, while the Transformer remains the strongest backbone under MAE ([Reply to Reviewer eCu6-2 (2)](https://openreview.net/forum?id=gUXKz2PwQb&noteId=wP5ZJnmWjn) / Appendix M-Page 45).
>
> - Clarified our focus on single‑lead pretraining (motivated by both simulator design and real‑world single‑lead use cases) and added experiments on all 12 individual leads and on 12‑lead PTB‑XL classification using a shared‑encoder wrapper, showing that our conclusions extend beyond the lead‑II, binary setting ([Reply to Reviewer eCu6-2 (5)](https://openreview.net/forum?id=gUXKz2PwQb&noteId=C4SEvG1h2K) / Appendix P-Page 52).
>
> We truly appreciate the time you invested in reading our rebuttal, in following up with additional comments, and in pushing us to strengthen and better position the work.

---

### Official Review · Reviewer_ThrN · 2025-10-25

**Soundness:** 3
**Presentation:** 3
**Contribution:** 3
**Rating:** 6
**Confidence:** 5

**Summary:**

This paper investigates whether simulator-generated ECG signals can serve as effective pretraining data for self-supervised learning and subsequently benefit downstream ECG classification tasks.

**Strengths:**

I find this exploratory study valuable. Although it does not introduce new model architectures, it provides meaningful empirical insights and implications that may inspire future research directions.

**Weaknesses:**

1. However, given that the contribution lies primarily in empirical exploration rather than methodological novelty, I believe this work may be more suitable for a journal submission rather than a conference. Conference page limits may restrict the authors from fully presenting the depth of their analyses and findings.
2. According to Table 1, when downstream training data are limited, the model pretrained on simulator-generated signals exhibits inferior performance; yet, as the amount of downstream data increases, its performance converges toward that of models pretrained on real ECG signals. This trend appears intuitive: with increased training data for fine-tuning, downstream performance depends more on the classification data itself than on the pretraining source. A similar phenomenon is evident in Figure 2, where the scratch model—without any pretraining—achieves comparable performance when sufficient labeled data are available. These observations may weaken the practical impact of the claim regarding the advantages of simulator-based self-supervised pretraining.
3. In fact, a more significant contribution of this study could lie in its comparison with deep generative models such as GAN- and VAE-based ECG synthesis. Unfortunately, this part is relegated to the appendix and is not sufficiently highlighted. Moreover, the appendix presents mixed outcomes—simulator-based models outperform generative models in some cases and underperform in others—without synthesizing clear or generalizable conclusions. The authors are encouraged to extract higher-level insights rather than simply listing experimental results.
4. Simulator-generated signals may lack patient-specific variability. By contrast, deep generative models trained on large-scale real ECG datasets can produce signals with richer pathological diversity and patient-specific morphology. Therefore, similar to the analyses in Figures 1 and 2, the authors should include comparative experiments between simulator-based pretraining and various deep generative models under varying downstream data sizes. Additionally, comparisons with diffusion-based ECG generation methods—such as Biomedically Informed ECG Synthesis: Customizing Cardiac Cycle Phases with Diffusion Models (Y. Lin et al., IEEE BIBM 2024)—should be incorporated to improve completeness and relevance.

**Questions:**

none

---

> ### Author Response · Authors · 2025-11-21
> **Reply to Reviewer ThrN**
>
> We are grateful to the reviewer for their constructive and valuable feedback. We are currently performing additional experiments and will incorporate the findings into our responses as they become available. For now, we provide answers to the points that can be addressed at present.
>
> ## 2.
>
> > According to Table 1, when downstream training data are limited, the model pretrained on simulator-generated signals exhibits inferior performance; yet, as the amount of downstream data increases, its performance converges toward that of models pretrained on real ECG signals. This trend appears intuitive: with increased training data for fine-tuning, downstream performance depends more on the classification data itself than on the pretraining source. A similar phenomenon is evident in Figure 2, where the scratch model—without any pretraining—achieves comparable performance when sufficient labeled data are available. These observations may weaken the practical impact of the claim regarding the advantages of simulator-based self-supervised pretraining.
>
> We appreciate the reviewer's insightful observation. We acknowledge that this trend could potentially diminish the contribution of our approach under conditions where sufficient real-world data are available.
> However, the ECG domain represents an exceptionally well-resourced case where large-scale public datasets are available due to extensive research efforts. In contrast, most medical signal domains face significant data scarcity. For instance, publicly available PCG datasets remain limited: PhysioNet 2016 contains 3,126 records [ThrN-c1] and PhysioNet 2022 contains 5,272 recordings from 1,568 patients [ThrN-c2].
> To validate our approach under realistic data-limited conditions, we conducted additional experiments on PCG data using BMD-HS dataset [ThrN-c3]. Our preliminary results show that even when using the entire available dataset, the simulator-pretrained model outperforms the scratch model for AR and AS classifications. These findings confirm that simulator-based self-supervised pretraining provides meaningful advantages in practical scenarios where large-scale real data are not readily available.
>
> Table ThrN-T1: Classification result on phonocardiogram (AUROC) | Identical to Table GHNt-T5.
> *Note: We report AUROC for PCG tasks following standard practice in heart sound classification literature, while F1-score is used for ECG tasks as established in PTB-XL and related benchmarks.*
> ||Pretraining Data| AR | AS | MR | Average |
> |---|---|---|---|---|---|
> | GRU | - |0.5471 ± 0.0652 | 0.5508 ± 0.1291|0.4110 ± 0.2647 |	0.5030 |
> |LSTM |-|0.4424 ± 0.2292 | 0.5942 ± 0.1069	| **0.5989 ± 0.0352** |0.5452 |
> | LUNA |-| 0.4327 ± 0.2167|0.3705 ± 0.1712|0.4255 ± 0.1940|0.4096|
> | MEGA |-|0.5467 ± 0.0556|0.5495 ± 0.0984|0.5616 ± 0.0282|0.5526|
> |ResNet18| -|0.4756 ± 0.0756 |0.3373 ± 0.0812|0.4042 ± 0.0827|0.4057|
> |Transformer|-|0.5486 ± 0.0356|0.4449 ± 0.2417|0.5407 ± 0.0367|0.5114|
> |Transformer|PhysioNet2022|0.6151 ± 0.0666|0.5851 ± 0.0644|0.5741 ± 0.0393|0.5914|
> |Transformer|BMD-HS|0.6273 ± 0.0176|0.5856 ± 0.0787|0.5774 ± 0.0500|0.5968|
> |Transformer|SimPCG|**0.6295 ± 0.0538**|**0.6356 ± 0.0909**|0.5786 ± 0.0590|**0.6146**|
>
> [ThrN-c1] Classification of normal/abnormal heart sound recordings: The PhysioNet/Computing in Cardiology Challenge 2016
>
> [ThrN-c2] Heart Murmur Detection from Phonocardiogram Recordings: The George B. Moody PhysioNet Challenge 2022
>
> [ThrN-c3] BUET Multi-disease Heart Sound Dataset: A Comprehensive Auscultation Dataset for Developing Computer-Aided Diagnostic Systems
>
> ## 4.
>
> > Simulator-generated signals may lack patient-specific variability. By contrast, deep generative models trained on large-scale real ECG datasets can produce signals with richer pathological diversity and patient-specific morphology. Therefore, similar to the analyses in Figures 1 and 2, the authors should include comparative experiments between simulator-based pretraining and various deep generative models under varying downstream data sizes. Additionally, comparisons with diffusion-based ECG generation methods—such as Biomedically Informed ECG Synthesis: Customizing Cardiac Cycle Phases with Diffusion Models (Y. Lin et al., IEEE BIBM 2024)—should be incorporated to improve completeness and relevance.
>
> Thank you for your suggestion.
> We conducted additional experiments with models pretrained using VAE/GAN-generated data under varying downstream data sizes. The results are shown in updated Figure 1.
> For all classification tasks, Real-world pretraining consistently achieved the highest performance. SimECG(both McSharry and Nonaka) pretraining showed the performance trend following Real-world pretrained model, while VAE and GAN-based pretraining remained in lower positions, consistent with the full-data results reported in Table 1.

---

> > ### Author Response · Authors · 2025-12-01
> > **Reply to Reviewer ThrN (2)**
> >
> > ## 3.
> >
> > > In fact, a more significant contribution of this study could lie in its comparison with deep generative models such as GAN- and VAE-based ECG synthesis. Unfortunately, this part is relegated to the appendix and is not sufficiently highlighted. Moreover, the appendix presents mixed outcomes—simulator-based models outperform generative models in some cases and underperform in others—without synthesizing clear or generalizable conclusions. The authors are encouraged to extract higher-level insights rather than simply listing experimental results.
> >
> > We appreciate the reviewer's suggestion to extract higher-level insights from our comparison with deep generative models.
> > While we consider our primary contribution to be demonstrating that knowledge-driven synthetic data can match real-world pretraining performance, we agree that the comparison with VAE/DCGAN deserves clearer synthesis.
> > In the revision, we have reorganized and expanded this analysis, now summarized in Appendix F.
> >
> > First, we **clarify the overall ranking of pretraining corpora**.
> > For our primary objective (MAE), simulator-based pretraining (especially SimECG (Nonaka)) consistently outperforms VAE/DCGAN pretraining and is competitive with real PTB-XL across PTB-XL, G12EC, and CPSC: SimECG (Nonaka) nearly matches real-data pretraining on PTB-XL/G12EC and slightly surpasses it on CPSC, while VAE is clearly weakest and DCGAN lies in between.
> > We also added a non-inferiority analysis (Appendix F.1.1, Tables 13–14) showing that, under a 2.5% relative margin, SimECG (Nonaka) is statistically non-inferior to real-data pretraining, whereas VAE and DCGAN are not.
> >
> > Second, we discuss **how these trends depend on the self-supervised objective**.
> > For SimCLR, DCGAN pretraining often yields the highest mean F1, with SimECG close on PTB-XL and CPSC but less competitive on G12EC; for DINO, SimECG (Nonaka) is slightly below real-data pretraining on PTB-XL/CPSC but surpasses it on G12EC.
> > This "mixed" behavior is now explicitly framed as an interaction between (i) the data-generation mechanism and (ii) the self-supervised learning loss.
> >
> > To provide the requested higher-level interpretation, Appendix F.4 now contrasts the properties of SimECG (knowledge-driven, high morphological control with idealized noise) with those of a basic VAE (smoothed, averaged morphologies) and a DCGAN (sharper morphology and realistic PTB-XL-like noise but possible mode bias), and relates them to the inductive biases of MAE, SimCLR, and DINO.
> > We explicitly highlight that MAE benefits most from the diverse but clean morphology provided by simulators, SimCLR leverages realistic nuisance variation captured well by DCGAN, and DINO sits between these extremes.
> >
> > Finally, we will update PDF to briefly summarize these conclusions in the main text (Section 5/Discussion) and now point explicitly to Appendix F, so that the role of generative-model pretraining and our takeaways—namely, that knowledge-driven simulators are a robust, privacy-preserving alternative that typically outperform straightforward VAE/DCGAN pretraining—are more clearly foregrounded.

---

> > > ### Author Response · Authors · 2025-12-02
> > > **Updated reply to Reviewer ThrN weakness-4**
> > >
> > > # 4. Comparison against deep generative models.
> > >
> > > > Simulator-generated signals may lack patient-specific variability. By contrast, deep generative models trained on large-scale real ECG datasets can produce signals with richer pathological diversity and patient-specific morphology. Therefore, similar to the analyses in Figures 1 and 2, the authors should include comparative experiments between simulator-based pretraining and various deep generative models under varying downstream data sizes. Additionally, comparisons with diffusion-based ECG generation methods—such as Biomedically Informed ECG Synthesis: Customizing Cardiac Cycle Phases with Diffusion Models (Y. Lin et al., IEEE BIBM 2024)—should be incorporated to improve completeness and relevance.
> > >
> > > [This post is a updated version of our reply to weakness point 4.]
> > >
> > > Thank you again for this suggestion.
> > > We have conducted experiments comparing simulator-based pretraining with deep generative models (VAE and DCGAN) under varying downstream data sizes.
> > >
> > > **Results.**
> > > As shown in Figure 2 (page 8) and Table 1 (page 6), we additionally evaluated pretraining on VAE/DCGAN-generated ECG alongside SimECG and real-world data across 13 PTB-XL tasks under both full-data and reduced training sets (n ∈ {1, 2, 5, …, 1000} samples per class):
> > >
> > > - Real-world pretraining achieved the highest performance (mean F1 = 0.8488, rank 1.54).
> > >
> > > - SimECG (Nonaka) closely matched real-world pretraining (F1 = 0.8451, rank 1.85), followed by SimECG (McSharry).
> > >
> > > - VAE (F1 = 0.7763, rank 4.88) and DCGAN (F1 = 0.8111, rank 3.65) underperformed across tasks and label budgets; in Figure 2, their curves consistently trail those of real-data and SimECG pretraining.
> > >
> > > These results indicate that when trained on PTB-XL, data‑driven generative models do not close the gap to simulator-based pretraining, supporting our thesis that physiological simulators can substitute for learned generative distributions while avoiding direct access to patient records.
> > >
> > > **Regarding diffusion-based methods.**
> > > We agree that recent diffusion models for ECG synthesis (e.g., [ThrN-c4, c5, c6, c7, c8]) represent an important advance. However, like VAEs and GANs, these models must be trained on large collections of real ECGs and therefore inherit the same privacy and governance constraints as real-data pretraining. Moreover, prior work has shown that high-capacity generative models can memorize and leak training examples ([ThrN-c9, c10]), which is particularly concerning for medical data.
> > > Our VAE/DCGAN experiments already suggest that PTB-XL–trained data-driven generators do not outperform simulator-based pretraining in our setting.
> > >
> > > For this reason, we view diffusion-based ECG generators as complementary to our work rather than as direct substitutes for knowledge-driven simulators.
> > > In the revised manuscript, we have expanded the Related Work section to more explicitly position diffusion-based ECG synthesis as a complementary, data-driven approach and to clarify its privacy and data-access assumptions relative to our simulator-based pretraining.
> > >
> > >
> > > [ThrN-c4]: Lin, Yuxin, et al. "Biomedically Informed ECG Synthesis: Customizing Cardiac Cycle Phases with Diffusion Model." 2024 IEEE International Conference on Bioinformatics and Biomedicine (BIBM). IEEE, 2024.
> > > ;  Bedin et al., 2024; Lai et al., 2025)
> > >
> > > [ThrN-c5]: Alcaraz, Juan Miguel Lopez, and Nils Strodthoff. "Diffusion-based conditional ECG generation with structured state space models." Computers in biology and medicine 163 (2023): 107115.
> > >
> > > [ThrN-c6]: Neifar, Nour, et al. "Diffecg: A versatile probabilistic diffusion model for ecg signals synthesis." 2024 IEEE/ACIS 22nd International Conference on Software Engineering Research, Management and Applications (SERA). IEEE, 2024.
> > >
> > > [ThrN-c7]: Bedin, Lisa, et al. "Leveraging an ECG beat diffusion model for morphological reconstruction from indirect signals." Advances in Neural Information Processing Systems 37 (2024): 84409-84446.
> > >
> > > [ThrN-c8]: Lai, Yongfan, et al. "Ecgtwin: Personalized ecg generation using controllable diffusion model." arXiv preprint arXiv:2508.02720 (2025).
> > >
> > > [ThrN-c9]: Carlini, Nicolas, et al. "Extracting training data from diffusion models." 32nd USENIX security symposium (USENIX Security 23). 2023.
> > >
> > > [ThrN-c10]: Hayes, Jamie, et al. "Logan: Membership inference attacks against generative models." arXiv preprint arXiv:1705.07663 (2017).

---

> > > > ### Author Response · Authors · 2025-12-02
> > > > **Reply to Reviewer ThrN**
> > > >
> > > > # 1. Recommendation to a journal
> > > >
> > > > > However, given that the contribution lies primarily in empirical exploration rather than methodological novelty, I believe this work may be more suitable for a journal submission rather than a conference. Conference page limits may restrict the authors from fully presenting the depth of their analyses and findings.
> > > >
> > > > We appreciate the reviewer’s perspective and agree that our main contribution is an empirical exploration of simulator‑synthesized ECG data for pretraining.
> > > > However, we believe this type of systematic empirical study is well aligned with ICLR’s scope: our work provides new, actionable insights on when and how simulator‑generated ECG is beneficial, which we expect to be valuable for both researchers and practitioners working on clinical time‑series models.
> > > >
> > > > Regarding the concern about conference page limits, we have structured the paper so that the key messages and practical takeaways are fully contained in the main text, while additional analyses, ablations, and extended discussions are provided in the appendix to preserve depth without sacrificing readability.
> > > > In this sense, we view a conference venue as appropriate for rapidly disseminating these empirical findings to the community.

---

> ### Author Response · Authors · 2025-12-03
> **Summary of replies to Reviewer ThrN**
>
> Thank you very much for your thoughtful review and for your positive yet critical assessment of the paper.
>
> Following your suggestions, we
>
> (i) added PCG experiments to better illustrate the value of simulator‑based pretraining in more realistically data‑limited regimes ([Reply to Reviewer ThrN](https://openreview.net/forum?id=gUXKz2PwQb&noteId=J8Mrihp0pb) / Appendix N-Page 47);
>
> (ii) expanded and reorganized the comparison with VAE/GAN‑based synthetic data, including under varying downstream data sizes, and summarized the key take‑home messages rather than only listing results ([Reply to Reviewer ThrN (2)](https://openreview.net/forum?id=gUXKz2PwQb&noteId=eePpCtE8FW) / Appendix G.4-Page 29);
>
> (iii) broadened the related‑work discussion to include diffusion‑based ECG generation, clarifying that such data‑driven generators are complementary to (but have different privacy assumptions from) our knowledge‑driven simulators ([Updated reply to Reviewer ThrN weakness-4](https://openreview.net/forum?id=gUXKz2PwQb&noteId=8t4kP4Vzz7) / Related work-Page 2).
>
> We also re‑framed the main message to emphasize not just that “synthetic pretraining helps,” but that simulator‑based pretraining can, in intermediate data regimes, effectively stand in for real‑data self‑supervised pretraining while retaining much of its benefits in terms of label efficiency and robustness.
>
> We greatly appreciate your encouraging overall recommendation.

---

### Official Review · Reviewer_GHNt · 2025-10-26

**Soundness:** 3
**Presentation:** 1
**Contribution:** 1
**Rating:** 2
**Confidence:** 4

**Summary:**

The authors present an empirical comparison that shows that pre-training on synthetic ECG data generated through simulators can benefit the downstream performance of transformer-based models compared with approaches without pre-training or pre-training only on (smaller) real-world datasets.

**Strengths:**

- The paper is easy to follow and delivers one main message
- overall the approach is to some degree convincing

**Weaknesses:**

- While the idea overall shows convincing results, the main message is limited in that pre-training on synthetic data helps. A more in-depth analysis on the implications, clinical need and improvement, and similar aspects would be interested. In its current form the insight of the manuscript are limited.
- Your results indicate that pre-training on PTB-XL already gives the best performance. PTB-XL is a rather small-scale dataset. Thus pre-training on a synthetic dataset 50x the size of PTB-XL seems excessive if in many settings medium to large real-world datasets are available.
- A computational comparison on the pre-training budget on real-data and synthetic data is not presented. This it is unclear how the gains of your approach would be under the same computational conditions.
- many information double and scattered. The manuscript reads at times repetitive. Example: Caption table 1 and text right below describes essentially the same.
- in the implications you mention that pre-training size can be scaled more easily in the simulator-based approach. However, you do not show that more pre-training on synthetic data could improve performance. I would expect some kind of plataeu because of missing real-world nuisances.

**Questions:**

- Some references to ECG-based pretraining methods are missing, e.g. [1] and others.
- please double check writing. E.g. missing blankspace in line 056, 067, and others
- line 139. How exactly do you parameterize the simulator. You say "with parameters randomly sampled from predefined distributions", which parameters do you choose or how do you choose the distributions?
- section 3: It is unclear to me whether you focus your whole development on lead II only or why you are extracting lead II only here. Can you clarify?
- section 5: for each of the 4 axes, how do you define the metrics, e.g. performance, data efficiency and robustness?
- Table 1:
  - Unclear abbreviations: e.g. DGM as VAE or as GAN
  - Can you provide std of the metrics over multiple runs? Meaning are the results significant
- Table 2:
  - Can you provide std of the metrics over multiple runs? Meaning are the results significant
  - the comparison is somewhat flawed since you compare non-pretrained models with a pretrained model. If you only compare the non-pretrained models, then the GRU actually performs best. A rigorous comparison would compare on pretraining in the same way.
- Figure 1:
  - font size too small
  - what is the x-axis? (it is number of pretraining samples but please make this explicity)
- line 311: "degrades more gracefully " what does that mean?
- why do you focus on single lead classification instead of 12-lead which often the standard in clinical settings?


[1] Kiyasseh, Dani, Tingting Zhu, and David A. Clifton. "Clocs: Contrastive learning of cardiac signals across space, time, and patients. "International Conference on Machine Learning.

---

> ### Author Response · Authors · 2025-11-21
> **Reply to Reviewer GHNt**
>
> We thank the reviewer for their thoughtful and insightful comments, which we found extremely valuable. We are currently conducting additional experiments and will update our responses accordingly as the results become available. In the meantime, we address the points that we can respond to at this stage.
>
> # Weaknesses
>
> **Pre-training computational budget:**
>
> > A computational comparison on the pre-training budget on real-data and synthetic data is not presented. This it is unclear how the gains of your approach would be under the same computational conditions.
>
> We thank the reviewer for this important point and apologize for the lack of clarity in our presentation.
> All pretraining conditions used identical computational budgets (Appendix C.2, Section 5.3): 1M training samples, ~1B sample exposures, 6 days on single NVIDIA A5000 GPU, identical hyperparameters. We will clarify this explicitly in the revised Section 5.3.
>
>
> # Questions
>
> ### references
>
> > Some references to ECG-based pretraining methods are missing, e.g. [1] and others.
>
> Thank you for your suggestion. We updated the related work section.
>
> ### Typos
>
> > please double check writing. E.g. missing blankspace in line 056, 067, and others
>
> Thank you for the notification. We have fixed the manuscript.
>
> ### Table1
>
> Thank you for your valuable feedback.
>
> > Unclear abbreviations: e.g. DGM as VAE or as GAN
>
> Thank you for pointing this out. As you noted, the DGM abbreviation was unclear, so we have revised it to specify VAE/DCGAN.
>
> > Can you provide std of the metrics over multiple runs? Meaning are the results significant
>
> ** Standard deviations:**
> We apologize for the confusion. We report the mean and standard deviation over 5 runs in Appendix E (Table 10,11,12).
>
> **Statistical significance:**
>
> Thank you also for your question about statistical significance.
> We conducted a non-inferiority test using the Real-world data condition as the reference.
> For each of the 26 tasks, we computed the mean F1-score (over five runs) for the Real-world condition and for each synthetic-data condition.
> We then defined a relative non-inferiority margin $\delta$ (e.g., $\delta$ = 0.025 corresponds to allowing up to a 2.5% decrease in F1) and, for each task i, formed the difference
>
> $d_i = F1_{test,i} - (1 - \delta) \cdot F1_{real-world,i}$
>
> A one-sided Wilcoxon signed-rank test was applied to $d_i$.
>
> The null hypothesis was $H_0$: median$(d_i) \leq 0$ (the test condition is inferior to the Real-world condition by more than the margin), and the alternative hypothesis was $H_1$: median$(d_i) > 0$ (the test condition is non-inferior to the Real-world condition). This formulation is equivalent to testing whether the F1 score under the test condition is at least (1 − $\delta$) times the F1-score under the Real-world condition.
>
> The main analysis used a non-inferiority margin of $\delta = 2.5\%$, with sensitivity analyses conducted at 1%, 5%, and 10%. Because multiple data conditions were tested against the same Real-world reference, we applied a Bonferroni correction [GHNt-c1]; we report both uncorrected and corrected p-values.
>
> Table GHNt-T1 shows the results across all 26 tasks.
> The SimECG (Nonaka) condition satisfied the non-inferiority criterion (p-value: 0.025) relative to the Real-world condition when $\delta$ was set at 2.5%.
> For all other data conditions, the null hypothesis could not be rejected.
> Table GHNt-T2 presents the sensitivity analysis results.
>
>
> Table GHNt-T1. Non-inferrior test for result shown in Table 1 (PTBXL+G12EC+CPSC).
> || p-value | p-value (corrected) |
> |---|---|---|
> | SimECG (McSharry) | 0.1767 | 0.7066 |
> | SimECG (Nonaka)  | 0.0002 | 0.0008 |
> | VAE | 1.0000 | 1.0000 |
> | DCGAN | 0.9387 | 1.0000 |
>
> Table GHNt-T2. Sensitivity test of $\delta$ for Non-inferrior test of results shown in Table 1 (corrected p-value).
> |$\delta$| 0.01 | 0.025 | 0.05 | 0.10 |
> |---|---|---|---|---|
> | SimECG (McSharry) | 1.0000 | 0.7066 | 0.0015 | 0.0000 |
> | SimECG (Nonaka)  | 1.0000 | 0.0008 | 0.0000 | 0.0000 |
> | VAE | 1.0000 | 1.0000 | 1.0000 | 0.4345 |
> | DCGAN | 1.0000 | 1.0000 | 1.0000 | 0.0380 |
>
>
> ### Figure1
>
> > font size too small
> > what is the x-axis? (it is number of pretraining samples but please make this explicity)
>
> Thank you for pointing this out.
> We have increased the font size and added an explicit x-axis label to clarify that it represents the number of pretraining samples.
>
> [GHNt-c1]: Dunn, Olive Jean. "Multiple comparisons among means." Journal of the American statistical association 56.293 (1961): 52-64.

---

> > ### Author Response · Authors · 2025-11-27
> > **Reply to Reviewer GHNt (2)**
> >
> > # Questions
> >
> > ### Table2
> >
> > > Can you provide std of the metrics over multiple runs? Meaning are the results significant
> >
> > **Standard deviations:**
> > Thank you for your comments. Regarding the standard deviation, we apologize for the unclear presentation. We have reported the mean and standard deviation over 5 trials in Appendix F (Tables 19, 20, 21).
> >
> > **Statistical significance:**
> > Thank you also for your question about statistical significance.
> > To assess the superiority of the Transformer (SimECG) condition over the other models, we performed paired superiority tests on the per-task F1 scores.
> > For each ECG classification task, we computed the mean F1 score for the Transformer (SimECG) condition and for each comparison model, and then formed the per-task differences
> >
> > $d_i = F1_{Transformer\(SimECG\), i} - F1_{comparison, i}$
> >
> > We applied a one-sided Wilcoxon signed-rank test to these paired differences across the 26 tasks, as this test is appropriate for paired data and does not require the assumption of normality.
> > The null hypothesis was $H_0: median(d_i) \leq 0$ (no superiority of the Transformer (SimECG) condition over the comparison model), and the alternative hypothesis was $H_1: median(d_i) > 0$ (the Transformer (SimECG) condition is superior to the comparison model).
> >
> > Multiple comparison correction across models was performed using the Bonferroni method [GHNt-c1], and we present both the uncorrected and corrected p-values.
> > The results across all 26 tasks are shown in Table GHNt-T3.
> >
> > Table GHNt-T3. Statistical significance test for result shown in Table 2 (PTBXL+G12EC+CPSC).
> > |Baseline model| p-value | p-value (corrected) |
> > |---|---|---|
> > | GRU | 0.0000 | 0.0000 |
> > | LSTM  |  0.0000 | 0.0000 |
> > | LUNA |  0.0000 | 0.0000 |
> > | MEGA |  0.0000 | 0.0000 |
> > | ResNet18 | 0.0000 | 0.0000 |
> > | Transformer (scratch) | 0.0000 | 0.0000 |
> >
> >
> >
> > > the comparison is somewhat flawed since you compare non-pretrained models with a pretrained model. If you only compare the non-pretrained models, then the GRU actually performs best. A rigorous comparison would compare on pretraining in the same way.
> >
> > Thank you for your valuable suggestion.
> > We additionally performed MAE-style pretraining for the top three baseline models (GRU/LSTM/ResNet) and evaluated them on the 13 PTBXL tasks.
> > The respective results are shown in Table GHNt-T4.
> > For reference, we also include the Transformer results (identical to those in Tables 10 and 19).
> > We will report detailed results with standard deviation of 5 runs in Appendix.
> >
> > Table GHNt-T4. Summary of PTBXL classification result with MAE pretraining (PTBXL, SimECG(Nonaka)) and non-pretrained model of GRU, LSTM, ResNet18, and Transformer.
> > | | Scratch | PTBXL pretrained | SimECG(Nonaka) pretrained |
> > | --- | --- | --- | --- |
> > | GRU | 0.7854 | 0.8274 | 0.8305 |
> > | LSTM | 0.7726| 0.8284 | 0.8256 |
> > | ResNet18 | 0.7617 | 0.6835 | 0.6814 |
> > | Transformer | 0.7378 | 0.8488 | 0.8451 |
> >
> > The results demonstrate that when using MAE-style pretraining, both GRU and LSTM show substantial improvements (GRU: +5.35% with PTBXL pretraining, +5.74% with SimECG; LSTM: +7.22% with PTBXL pretraining, +6.86% with SimECG).
> > To assess statistical significance, we conducted one-sided Wilcoxon signed-rank tests across the 13 PTBXL tasks with Bonferroni correction for multiple comparisons.
> > Both GRU and LSTM demonstrated statistically significant performance gains over training from scratch for both pretraining conditions (p < 0.025 after Bonferroni correction for all four comparisons: GRU-PTBXL, GRU-SimECG, LSTM-PTBXL, and LSTM-SimECG).
> > In contrast, ResNet18 shows performance degradation with MAE pretraining (0.6835 vs 0.7617 from scratch).
> > These results confirm that the Transformer architecture remains the most effective backbone for ECG classification when combined with masked autoencoder pretraining.

---

> > > ### Author Response · Authors · 2025-11-27
> > > **Reply to Reviewer GHNt (3)**
> > >
> > > # Weaknesses
> > >
> > > > Your results indicate that pre-training on PTB-XL already gives the best performance. PTB-XL is a rather small-scale dataset. Thus pre-training on a synthetic dataset 50x the size of PTB-XL seems excessive if in many settings medium to large real-world datasets are available.
> > >
> > > Thank you for this important observation.
> > > We agree that PTB-XL is relatively small-scale, and our results show strong performance with this dataset.
> > > However, we emphasize that ECG represents an exceptional case in medical signal domains where large-scale public datasets are available through extensive research efforts over decades.
> > > We selected ECG specifically for validation purposes because such large real-world datasets enable rigorous comparison and evaluation of our approach.
> > >
> > > We discuss the performance of PTB-XL pre-training and the rationale for synthetic data scale in our response to [Reply to Reviewer GHNt (4)], which addresses related concerns about sample efficiency.
> > >
> > > **Application beyond ECG:**
> > >
> > > In most medical signal domains, such large-scale public datasets are not available.
> > > To demonstrate the practical value of our approach in more realistic data-limited scenarios, we conducted additional experiments on phonocardiogram (PCG) data, where public datasets are substantially more limited (detailed in Appendix).
> > > We evaluate PCG classification using AUROC, which is widely used in the medical signal processing literature to assess model performance on imbalanced datasets, particularly relevant for PCG data where certain cardiac conditions are significantly underrepresented.
> > >
> > > The results shown in Table GHNt-T5 show that synthetic pre-training provides improvements in PCG classification tasks, demonstrating the practical utility of our approach when large real-world datasets are unavailable.
> > > Notably, our synthetic PCG approach (SimPCG) achieves the best average F1-score of $0.6146$, outperforming both models without pre-training (best: $0.5526$) and those pre-trained on limited real PCG data (best: $0.5968$).
> > >
> > > Table GHNt-T5: Classification result on phonocardiogram (AUROC).
> > > *Note: We report AUROC for PCG tasks following standard practice in heart sound classification literature, while F1-score is used for ECG tasks as established in PTB-XL and related benchmarks.*
> > > ||Pretraining Data| AR | AS | MR | Average |
> > > |---|---|---|---|---|---|
> > > | GRU | - |0.5471 ± 0.0652 | 0.5508 ± 0.1291|0.4110 ± 0.2647 |	0.5030 |
> > > |LSTM |-|0.4424 ± 0.2292 | 0.5942 ± 0.1069	| **0.5989 ± 0.0352** |0.5452 |
> > > | LUNA |-| 0.4327 ± 0.2167|0.3705 ± 0.1712|0.4255 ± 0.1940|0.4096|
> > > | MEGA |-|0.5467 ± 0.0556|0.5495 ± 0.0984|0.5616 ± 0.0282|0.5526|
> > > |ResNet18| -|0.4756 ± 0.0756 |0.3373 ± 0.0812|0.4042 ± 0.0827|0.4057|
> > > |Transformer|-|0.5486 ± 0.0356|0.4449 ± 0.2417|0.5407 ± 0.0367|0.5114|
> > > |Transformer|PhysioNet2022|0.6151 ± 0.0666|0.5851 ± 0.0644|0.5741 ± 0.0393|0.5914|
> > > |Transformer|BMD-HS|0.6273 ± 0.0176|0.5856 ± 0.0787|0.5774 ± 0.0500|0.5968|
> > > |Transformer|SimPCG|**0.6295 ± 0.0538**|**0.6356 ± 0.0909**|0.5786 ± 0.0590|**0.6146**|
> > >
> > > **Scale impact of synthetic data:**
> > >
> > > Regarding the scale of synthetic data, we conducted ablation studies varying the pre-training dataset size from $1M$ to $2k$ samples.
> > > The results in Table GHNt-T6 show that performance remains strong even with substantially reduced synthetic data: at $100k$ samples (approximately 15× the size of PTB-XL or G12EC), the mean F1-score is $0.8424$, comparable to the $1M$ sample condition ($0.8389$).
> > > At $50k$ samples—roughly 4-8× the size of real-world datasets—the approach achieves a mean F1 of $0.8396$, which still exceeds the performance of pre-training on G12EC real data ($0.8371$).
> > > Even at $20k$ samples—comparable to PTB-XL's training set size—our approach achieves a mean F1 of $0.8319$, demonstrating effectiveness without requiring excessive amounts of synthetic data.
> > > This indicates that our approach does not require the full $1M$ samples to be effective, directly addressing your concern about the 50× scale.
> > >
> > > Table GHNt-T6: Classification result with reduced number of (synthetic) pretraining data.
> > > ||2k | 5k|10k| 20k | 50k | 100k | 200k | 500k | 1M |
> > > |---|---|---|---|---|---|---|---|---|---|
> > > |Mean F1score|0.7860 | 0.8019| 0.8249|0.8319|0.8396|0.8424|0.8427|0.8481|0.8389|
> > > |Relative improvements |-|2.02%|2.87%|0.84%|0.93%|0.34%|0.04%| 0.64% | -1.08% |
> > >
> > > c.f.
> > > - G12EC: Mean F1 = 0.8371 / Training data = 6,585
> > > - PTBXL: Mean F1 = 0.8474 / Training data = 13,975

---

> ### Author Response · Authors · 2025-11-27
> **Reply to Reviewer GHNt (4)**
>
> # Weaknesses
>
> > in the implications you mention that pre-training size can be scaled more easily in the simulator-based approach. However, you do not show that more pre-training on synthetic data could improve performance. I would expect some kind of plataeu because of missing real-world nuisances.
>
> Thank you for this insightful comment. In summary, our new experiments show that increasing the amount of synthetic pre‑training data from $2k$ to $1M$ samples continues to improve downstream performance, although the gains saturate beyond about $200k$ samples under a fixed compute budget.
>
> To directly address your concern, we conducted additional experiments to study the scaling behavior of synthetic pre‑training (details in the Appendix; new Figure ~~4~~ 7a–c, Figure ~~3~~ 6, and Table GHNt-T6 in [Reply to Reviewer GHNt (3)]).
>
> **Experimental Setup:**
>
> We systematically varied the synthetic pre‑training dataset size from $2k$ to $1M$ samples ($2k, 5k, 10k, 20k, 50k, 100k, 200k, 500k, 1M$) while keeping the pre‑training compute comparable across conditions.
> For each configuration, we monitored the pre‑training validation loss and the downstream F1‑score after fine‑tuning.
>
> **Key Findings:**
>
> **(1) Scaling behavior of validation loss (Figure ~~4~~7a–c).**
>
> The validation loss curves (Figure ~~4~~7a, b, c in Appendix) reveal:
>
> - For real‑world pre‑training (G12EC, PTB‑XL), we observe U‑shaped curves: the validation loss first decreases and then increases with prolonged training, consistent with overfitting.
>
> - For synthetic pre‑training (SimECG), small datasets ($2k–50k$) show similar U‑shaped curves. As we increase the dataset size ($100k–1M$), overfitting is reduced and the minimum validation loss decreases with dataset size, following scaling‑law‑like behavior. For $200k–1M$ samples, the curves have similar shapes and their minima are close, indicating diminishing returns at the largest scales we explored.
>
> **(2) Performance scaling under equal computational budget (Figure ~~3~~6):**
>
> When we compare pre‑training methods under the same amount of processed data (i.e., training compute):
>
> - **Early phase (~$10^6$ to ~$10^7$ processed data)**: Real-world pre-training achieves faster initial gains, reaching $F1\approx0.82-0.84$
>
> - **Convergence phase (~$10^8$ processed data)**: Synthetic pre-training with larger datasets ($200k-1M$) converges to comparable performance ($F1\approx0.84-0.85$), matching real-world pre-training levels
>
> - **Plateau phase (>$10^8$ processed data)**: All pre-training approaches plateau at $F1\approx0.83-0.85$
>
> **(3) Regarding the expected plateau of synthetic pre‑training:**
>
> Consistent with your intuition, we do observe diminishing returns as the synthetic dataset is scaled:
>
> - The marginal improvement in downstream F1 becomes small between $50k$ and $100k$ synthetic samples.
>
> - Performance stabilizes at approximately $200k$ samples, with F1 scores in the range 0.8319–0.8481 across different synthetic dataset sizes (Table GHNt-T6 from previous post)
>
> - Across all configurations, there is a strong negative correlation between the minimum pre‑training validation loss and downstream F1 ($r = –0.98, p < 10^6$), indicating that lower pre‑training loss is strongly associated with better downstream performance.
>
> Because the minimum validation loss continues to improve as we increase the SimECG dataset size, this correlation suggests that, within the range we studied, representation quality is still limited primarily by compute and dataset size rather than by an intrinsic ceiling of the simulator.
> We agree that synthetic data will eventually hit a fundamental limit due to missing real‑world nuances, but our results indicate that this limit has not yet been reached at 1M samples and that synthetic pre‑training can already match the performance of real‑world pre‑training.
>
> **(4) Practical implications:**
>
> These findings refine our original statement about scalability.
> Real‑world ECG data are more sample‑efficient, but synthetic data offer two practical advantages in data‑scarce settings: (a) the dataset size can be scaled by allocating more computation, without being constrained by data collection; and (b) synthetic pre‑training avoids catastrophic forgetting and enables stable, long‑term pre‑training.
> In domains such as PCG, where real‑world datasets are severely limited (≤3,163 recordings in PhysioNet 2022), the ability to scale synthetic pre‑training to hundreds of thousands or millions of samples is therefore crucial.
>
> We have added these new results to Appendix (Figure ~~4~~7a–c, Figure ~~3~~ 6, together with Figure ~~5~~ ~~6~~ 9 showing the correlation between pre-training validation loss and performance on downstream tasks).

---

> > ### Comment · Reviewer_GHNt · 2025-11-27
> >
> > I thank the authors for their detailed answers and rebuttal. You do have achieved a lot of progress, however some concerns remain:
> > - You speak about Fig. 3, Fig. 4, and Fig. 5, but these do not exist.
> > - I encourage you to add the analysis which is currently on in the rebutal, to the main manuscript.
> > - The new significance tests are interesting and good. But the presentation is unclear. Please report clearly the hypotheses, margins, and correction procedures in the main text and state the implications of the results.
> > - The rationale for focusing on single-lead ECG classification is not addressed. Given that 12-lead setups are standard in clinical practice, the paper should justify this choice more explicitly or provide complementary 12-lead results.
> > - Some earlier weaknesses (e.g., scattered presentation, repetition, missing parameterization details of the simulator) appear only partially resolved.
> > - You claim that ECGs are a data rich modality and thus compare with other data sources. However, your paper is titled `Simulator‑Based Synthetic ECGs for Self-Supervised Pretraining`, so you do have a focus on ECGs and should justify the limitations in that domain.
> >
> > I increase my score by one point until further limitations are addressed.

---

> ### Author Response · Authors · 2025-11-28
> **Reply to Reviewer GHNt's comment**
>
> Thank you for your continued engagement and for increasing your score.
> We appreciate the time you have taken to provide constructive feedback, which has significantly helped us improve the manuscript.
> (We update figure numbers due to )
>
> ## Regarding Figures 3, 4, and 5:
>
> We apologize for the confusion caused by the timing of our manuscript update. The current version of the manuscript now includes all figures referenced in our rebuttal:
>
> - Figure ~~3~~6: Performance comparison under equal computational budget
> - Figure ~~4~~7: Scaling behavior of validation loss
> - Figure ~~5~~8: Downstream performance scaling under varying pretraining data size
> - Figure ~~6~~9: Relationship between pre-training validation loss and downstream F1-score across SimECG dataset sizes
>
> We will also correct any figure number discrepancies in our (previous) responses.
>
> **Update on SimECG 10k, 5k, and 2k experiments:**
>
> We are currently completing the remaining experiments for SimECG 10k, 5k, and 2k conditions.
> Specifically, we will add during the rebuttal period:
>
> - Downstream task results under 5e8 and 1e9 computational budgets (Figure ~~3~~6)
> - Pretraining validation loss curves up to 1e9 FLOPs (Figure ~~4~~7)
>
> Based on the pretraining validation loss trends already observed, we expect that the best-performing conditions have been identified, and the main conclusions regarding SimECG's effectiveness will remain unchanged. If any unexpected results emerge that require correction, we will clearly report them.
>
> ## Regarding statistical significance tests:
>
> We appreciate your feedback on the presentation clarity.
> We will incorporate the detailed statistical analysis from our rebuttal (including the hypotheses, non-inferiority margins, correction procedures, and their implications) into the main manuscript during the rebuttal period.
>
> ## Regarding the focus on single-lead ECG and other remaining concerns:
>
> We will address your concerns about the rationale for single-lead ECG classification and the remaining weaknesses (presentation clarity, repetition, and simulator parameterization details) in separate follow-up responses shortly.
>
> Thank you again for your valuable feedback and for giving us the opportunity to strengthen our work.

---

> > ### Author Response · Authors · 2025-12-01
> > **Reply to Reviewer GHNt-2**
> >
> > We thank the reviewer for these constructive suggestions.
> > We are revising the manuscript to address the remaining concerns and will continue to update our responses as the revisions progress.
> >
> > > I encourage you to add the analysis which is currently on in the rebutal, to the main manuscript.
> >
> > We appreciate this suggestion.
> > Due to page constraints, we have placed some analyses in the Appendix.
> > We are currently revising the manuscript to incorporate key results from the rebuttal into the main text while maintaining the page limit.
> > Specifically, we plan to move the most critical computational equivalence and scaling analysis results to the Result section (Section 6).
> >
> > > The new significance tests are interesting and good. But the presentation is unclear. Please report clearly the hypotheses, margins, and correction procedures in the main text and state the implications of the results.
> >
> > We have revised Section 5.2 to include a dedicated "Statistical Testing" subsection that clearly describes:
> >
> > - Null and alternative hypotheses
> > - Non-inferiority margins (δ = 2.5%)
> > - Wilcoxon signed-rank test procedure with Bonferroni correction
> >
> > The main text now presents the key results concisely, with detailed test statistics and p-values provided in Appendix E Table 13 (non-inferiority test corresponding to results on Table1) and Appendix F Table 24 (superiority test corresponding to results on Table2).
> > This structure ensures the statistical rigor is transparent while maintaining readability in the main manuscript.
> >
> > > Some earlier weaknesses (e.g., scattered presentation, repetition, missing parameterization details of the simulator) appear only partially resolved.
> >
> > We acknowledge that not all presentational issues have been fully resolved and are continuing to improve the manuscript structure and clarity.
> > Regarding **the missing parameterization details of the simulator**, we have added a detailed description in Appendix C.
> > Our parameterization approach follows the parameters adopted in prior work and samples over ranges that cover physiologically known values.
> > We allow the sampling to include some examples beyond standard physiological ranges to ensure diversity while maintaining physiological plausibility.
> >
> > We are working on further revisions to reduce redundancy and improve the logical flow of the presentation throughout the paper.

---

> > > ### Author Response · Authors · 2025-12-01
> > > **Reply to Reviewer GHNt-2 (2)**
> > >
> > > # Issues related to implications
> > >
> > > > While the idea overall shows convincing results, the main message is limited in that pre-training on synthetic data helps. A more in-depth analysis on the implications, clinical need and improvement, and similar aspects would be interested. In its current form the insight of the manuscript are limited.
> > >
> > > We thank the reviewer for highlighting the need to better articulate the broader implications and clinical relevance of our findings.
> > > We agree that the original version could be read as merely showing that “pre-training on synthetic data helps.”
> > >
> > > Our intent, which we now make more explicit in the revised manuscript, is to study **whether knowledge‑driven ECG simulators can serve as an effective alternative to real patient ECGs as a pre‑training source in an intermediate data regime** where thousands to tens of thousands of unlabeled recordings are available, but foundation‑model–scale corpora are often difficult or impractical to assemble.
> > > To clarify this message, we have expanded the Introduction and Related Work to frame our contribution as orthogonal to ECG foundation models and alternative SSL objectives: we fix the encoder family and self‑supervised algorithm and vary only the pre‑training corpus, isolating the effect of simulator‑based synthetic data itself.
> > >
> > > In addition, we have reorganized the Result and Discussion section around clinically motivated axes:
> > >
> > > **Label efficiency and small‑cohort settings:** we emphasize that simulator‑based pre‑training produces label‑efficient representations that match or exceed strong supervised baselines even when only 1–5% of PTB‑XL labels are available, and in concrete examples such as ISC (50/50 positives/negatives) and PAC (100/100) the SimECG‑pretrained Transformer reaches or surpasses baselines trained on the full dataset. This corresponds to typical situations where expert annotation is limited.
> > >
> > > **Demographic and lead‑wise robustness:** we highlight that both real‑data and simulator‑based pre‑training substantially reduce performance gaps across age and sex, and that the SimECG‑pretrained model achieves near‑uniform performance across all 12 ECG leads, supporting deployment across heterogeneous devices and electrode placements (e.g., single‑lead wearables vs. 12‑lead clinical systems).
> > >
> > > **Privacy and data‑governance constraints:** the revised Discussion now makes explicit that simulator‑based pre‑training can be scaled without accessing patient‑identifiable data, avoiding institutional review, cross‑site data‑sharing agreements, and harmonization efforts, while still delivering strong initializations that reduce labeled‑data requirements and narrow demographic gaps.
> > >
> > > **Beyond ECG‑rich domains:** we added experiments on phonocardiogram (PCG) classification, a modality with substantially smaller public datasets, and show that simulator‑based pre‑training improves performance there as well, illustrating that our approach can also be beneficial in at least one other data‑scarce medical signal modality.
> > >
> > > We believe these additions make clearer that the main insight of the manuscript is not only that synthetic pre‑training improves performance, but that knowledge‑driven simulators can provide a practical, privacy‑preserving alternative to real‑data pre‑training in clinically relevant intermediate regimes, while preserving the key benefits of self‑supervised initialization in terms of label efficiency, robustness, and fairness.

---

> > > > ### Author Response · Authors · 2025-12-01
> > > > **Reply to Reviewer GHNt-2 (3)**
> > > >
> > > > # Points raised in initial "Question" section
> > > >
> > > > > section 5: for each of the 4 axes, how do you define the metrics, e.g. performance, data efficiency and robustness?
> > > >
> > > > We thank the reviewer for this clarification request.
> > > > We have revised Section 5 to explicitly define the metrics for each of the four evaluation axes:
> > > >
> > > > - **(i) Overall classification performance:** Mean F1-score and average rank across 26 abnormal-ECG classification tasks spanning PTB-XL (13 tasks), G12EC (7 tasks), and CPSC2018 (6 tasks).
> > > >
> > > > - **(ii) Data efficiency:** Performance (F1-score) as a function of training set size under subsampling (n ∈ {1, 2, 5, 10, 25, 50, 100, 250, 500, 1000} positives and negatives per class), evaluated on PTB-XL. We also report area under the learning curve as an aggregate metric.
> > > >
> > > > - **(iii) Demographic robustness:** Normalized performance disparity across age groups (four quartiles) and sex, quantified as |F₁,subgroup_max − F₁,subgroup_min| / F₁,overall on PTB-XL test sets.
> > > >
> > > > - **(iv) Lead-wise robustness:** Mean F1, standard deviation, and minimum F1 across all 12 standard ECG leads (I, II, III, aVR, aVL, aVF, V1–V6) to assess invariance to electrode placement.
> > > >
> > > > These definitions now appear explicitly in the revised Section 5 (lines 162–171) and are further detailed in the corresponding appendices (Appendices H–O).
> > > >
> > > > > line 311: "degrades more gracefully " what does that mean?
> > > >
> > > > We thank the reviewer for pointing out this ambiguous expression.
> > > > We have revised the text to provide a quantitative comparison with concrete F1 scores.
> > > >
> > > > The revised sentence now reads:
> > > >
> > > > "Figure 3 (Appendix H) shows that the Transformer (SimECG) maintains higher performance than baselines across most low-data regimes. For example, with n=100 samples per class, it achieves mean F1=0.7777 averaged across 13 PTB-XL tasks, compared to the best supervised baseline's F_1=0.6971 (+11.6% relative improvement)."

---

> > > > > ### Author Response · Authors · 2025-12-02
> > > > > **Reply to Reviewer GHNt-2 (4)**
> > > > >
> > > > > # Focus on ECG
> > > > >
> > > > > > You claim that ECGs are a data rich modality and thus compare with other data sources. However, your paper is titled Simulator‑Based Synthetic ECGs for Self-Supervised Pretraining, so you do have a focus on ECGs and should justify the limitations in that domain.
> > > > >
> > > > > We agree with the reviewer that our study is primarily focused on ECGs, and we now state this scope and its limitations more explicitly in the manuscript.
> > > > > Our pre‑training uses 10‑second 12‑lead ECGs from PTB‑XL (lead II), and all downstream evaluations are performed on short‑duration 12‑lead ECG datasets (PTB‑XL, G12EC, CPSC2018) with lead II as the primary input.
> > > > > The knowledge‑driven simulators we rely on are configured to match these specifications and to model adult cardiac electrophysiology under comparable conditions.
> > > > > As a consequence, our conclusions are most directly applicable to standard clinical 12‑lead ECG analysis, and may not fully transfer to other ECG settings such as long‑term Holter monitoring, intracardiac signals, pediatric ECGs, or scenarios with markedly different noise and artifact characteristics.
> > > > >
> > > > > In the revised Discussion (Limitations), we explicitly state that we only evaluate ECG and PCG in intermediate‑scale settings, that a domain gap remains between simulator outputs and real ECGs (with real‑data pre‑training still superior in some regimes), that we focus on single‑lead models with a single self‑supervised objective, and that the minimum fidelity required of synthetic data remains uncharacterized.
> > > > > Together, these points clarify the domain in which our conclusions should be interpreted and caution against over‑generalizing the results to arbitrary data‑scarce modalities.
> > > > > At the same time, we note that ECG is a natural first testbed for simulator‑based pre‑training, given the existence of several large public 12‑lead ECG datasets and the central role of ECG in clinical practice.
> > > > > To begin exploring transfer beyond ECG, we also report an experiment on phonocardiograms (PCGs); in the revised manuscript we present this as initial evidence that our approach can be useful in more typical data‑limited medical signal domains, rather than a definitive validation across all modalities.

---

> > > > > > ### Author Response · Authors · 2025-12-02
> > > > > > **Reply to Reviewer GHNt-2 (5)**
> > > > > >
> > > > > > # Issues related lead II setting
> > > > > >
> > > > > > ## Comment-1
> > > > > > > It is unclear to me whether you focus your whole development on lead II only or why you are extracting lead II only here. Can you clarify?
> > > > > >
> > > > > > We apologize for the lack of clarity.
> > > > > > In the current work all pretraining is performed on single‑lead lead‑II ECGs, and our main experiments also use lead II at fine‑tuning time.
> > > > > >
> > > > > > We chose lead II for three reasons:
> > > > > >
> > > > > > - (i) it provides the most stable P–QRS–T morphology and is the standard lead for rhythm analysis
> > > > > > - (ii) both knowledge‑driven simulators we use [GHNt-c2, c3] are parameterized to generate a single, lead‑II–like channel
> > > > > > - (iii) using a single lead allows us to keep the input format identical across PTB‑XL, G12EC, and CPSC2018 and across all pretraining data sources.
> > > > > >
> > > > > > To check that this choice does not overly specialize the model to one electrode placement, we additionally fine‑tune and evaluate the same models on each of the 12 standard leads on PTB‑XL (Page9 Table3 and Appendix O).
> > > > > > The simulator‑pretrained Transformer remains the top‑ranked model across all leads (average rank 1.00–1.31) and its mean F1 is consistently higher than strong supervised baselines, indicating that the learned representation is robust to lead choice and electrode placement.
> > > > > >
> > > > > > We revised Section 3 and Appendix B.5 to explicitly state that (i) all pretraining and main experiments are carried out on lead‑II ECGs, (ii) the rationale for this choice, and (iii) that we also report full 12‑lead robustness results in Appendix O.
> > > > > >
> > > > > > ## Comment-2
> > > > > >
> > > > > > > Why do you focus on single lead classification instead of 12‑lead which is often the standard in clinical settings?
> > > > > >
> > > > > > We fully agree that 12‑lead ECG is the standard in many diagnostic workflows.
> > > > > > Our focus on single‑lead classification is deliberate and is motivated by both **clinical use‑cases and technical constraints**:
> > > > > >
> > > > > > 1. **Clinical motivation (bedside/wearables).** In many real‑world settings the available signal is not a full 12‑lead ECG: bedside monitors, Holter monitors, and most wearables typically record one (or a few) leads rather than the full 12‑lead configuration. We therefore regard single‑lead performance as clinically relevant, especially for the “intermediate data regime” that we target (local hospital monitors, ambulatory/wearable devices, etc.).
> > > > > >
> > > > > > 2. **Alignment with knowledge‑driven simulators.** The two ECG simulators we use are single‑lead models calibrated to lead‑II morphology. Extending them to physiologically realistic 12‑lead outputs would require additional modeling of the 3‑D cardiac dipole and chest geometry, which is beyond the scope of this work. Using multi‑lead generators trained on real data (e.g., GANs or diffusion models) would reintroduce the privacy and governance issues that our simulator‑only pretraining is meant to avoid. For this reason we treat single‑lead simulators as our primary pretraining source.
> > > > > >
> > > > > > 3. **Controlling confounders.** Restricting the main experiments to a single lead keeps the architecture and input dimensionality fixed, allowing us to isolate the effect of pretraining data source (simulator vs. real vs. VAE/GAN) without conflating it with the additional modeling choices required for multi‑lead inputs.
> > > > > >
> > > > > > That said, we agree that it is important to verify that our conclusions are not an artifact of the single‑lead setting. We therefore performed two additional analyses:
> > > > > >
> > > > > > - **Lead‑wise robustness:** as noted above, we train and evaluate models separately on each of the 12 standard leads in PTB‑XL (Page9 Table3 and Appendix O). The simulator‑pretrained Transformer achieves the best or second‑best average rank on every lead, showing that the benefits of simulator‑based pretraining are not limited to lead II.
> > > > > >
> > > > > > - **12‑lead classification:** in Appendix P, we adapt the single‑lead encoders to 12‑lead input using a shared‑encoder wrapper and fine‑tune on 12‑lead PTB‑XL for the same 13 diagnostic labels. Both real‑data and SimECG pretraining continue to outperform end‑to‑end supervised baselines, and the relative ranking between real‑data and simulator pretraining is largely preserved, although the performance gap between them becomes slightly smaller in the 12‑lead setting.
> > > > > >
> > > > > > These results suggest that single‑lead pretraining on simulator‑generated ECGs already learns representations that transfer well to both other leads and full 12‑lead ECGs, while respecting the current limitation that our knowledge‑driven simulators are single‑lead.
> > > > > > We clarified this motivation and summarized the 12‑lead results with detailed numbers in Appendices O and P.
> > > > > >
> > > > > > [GHNt-c2]: McSharry, Patrick E., et al. "A dynamical model for generating synthetic electrocardiogram signals." IEEE transactions on biomedical engineering 50.3 (2003): 289-294.;
> > > > > >
> > > > > > [GHNt-c3]: Nonaka, Naoki, and Jun Seita. "Boosting ECG Classification Performance by Pre-training with Synthesized Data." ECML DEARING 2024;

---

> > > > > > > ### Author Response · Authors · 2025-12-03
> > > > > > > **Reply to Reviewer GHNt-2 (6)**
> > > > > > >
> > > > > > > # Parameterization of simulators
> > > > > > >
> > > > > > > > line 139. How exactly do you parameterize the simulator. You say "with parameters randomly sampled from predefined distributions", which parameters do you choose or how do you choose the distributions?
> > > > > > >
> > > > > > > Thank you for this valuable feedback. We have added detailed specifications of the simulator parameterization to Appendix C.
> > > > > > >
> > > > > > > - McSharry model (Appendix C.1):
> > > > > > >
> > > > > > > Heart rate parameters: mean heart rate $h \sim N(80, 10^2)$ bpm, standard deviation $H \sim N(0, 1^2)$ bpm
> > > > > > > Amplitude noise: $a \sim N(1, 1^2)$ mV
> > > > > > > Frequency domain parameters: LF band (f, v) and HF band (F, V) center frequencies and standard deviations sampled from uniform distributions
> > > > > > >
> > > > > > > - Nonaka model (Appendix C.2):
> > > > > > >
> > > > > > > Each component (P wave, QRS complex including Q/R/S waves, and T wave) is defined by amplitude, temporal position, and width parameters using Gaussian functions
> > > > > > > Employs a two-level perturbation scheme capturing both inter-individual and intra-individual variability
> > > > > > >
> > > > > > > The distributions were chosen to cover physiologically plausible ranges.
> > > > > > >
> > > > > > > Complete numerical specifications are provided in Appendix C.1-C.2.

---

> ### Author Response · Authors · 2025-12-03
> **Summary of replies to Reviewer GHNt**
>
> We are very grateful for your detailed and constructive comments throughout the review and rebuttal process.
> They significantly improved the paper.
>
> In the revised version, we:
>
> - (i) clarified that all pretraining conditions use matched computational budgets ("Reply to Reviewer GHNt's comment", "Reply to Reviewer GHNt-2" / Section 6.1 Figure 1 Page 6, Appendix K-Page 41);
>
> - (ii) added formal non‑inferiority and superiority tests (with hypotheses, margins, and Bonferroni correction) and summarized their implications in the main text ("Reply to Reviewer GHNt", "Reply to Reviewer GHNt (2)" / Section 5.3 Page 5, Appendix F-Page 22);
>
> - (iii) expanded the scaling analysis of synthetic pretraining (including new figures on validation‑loss and downstream‑performance scaling) ("Reply to Reviewer GHNt (4)" / Appendix M-Page 45);
>
> - (iv) clarified the parameterization of the simulators ("Reply to Reviewer GHNt-2 (6)" / Appendix C-Page 17);
>
> - (v) better motivated and documented our single‑lead setting, while adding 12‑lead and per‑lead analyses to show that the conclusions are not specific to lead II ("Reply to Reviewer GHNt-2 (5)" / Appendix P-Page 52).
>
> We also reorganized the results/discussion around performance, label efficiency, demographic robustness, and lead‑wise robustness, and added PCG experiments to illustrate transfer to another medical signal modality.
>
> Thank you again for engaging so deeply with the work and for adjusting your score during the rebuttal.

---

### Official Review · Reviewer_r4Jv · 2025-11-02

**Soundness:** 2
**Presentation:** 2
**Contribution:** 1
**Rating:** 0
**Confidence:** 5

**Summary:**

This paper presents a case study on how knowledge driven ECG simulators compare with real data for pretraining. The paper is a simple case-study, and as such, does not provide any contribution in either data generation or pretraining approaches. There is no theoretical and algorithmic contributions to science of signal generation and pretraining. For these reasons, the paper should be rejected.

**Strengths:**

None noted.

**Weaknesses:**

Lack of originality and novelty.

**Questions:**

None

---

> ### Author Response · Authors · 2025-11-21
> **Reply to reviewer r4Jv**
>
> We thank the reviewer for their time, though we respectfully disagree with the assessment.
>
> **On our contribution:** We acknowledge that we do not propose novel algorithms or theory. Our contribution is empirical: providing rigorous evidence on whether knowledge-driven synthetic ECG data can effectively support pretraining when real data is limited. This addresses a practically important question, especially, for medical ML practitioners.
>
> **On the value of empirical work:** The rationale that papers without "theoretical and algorithmic contributions" should be rejected appears inconsistent with ICLR's scope. ICLR has consistently valued rigorous empirical studies that provide actionable insights. The critical question should be whether our experiments are well-designed and our findings useful—not whether we propose new algorithms.
>
> **On medical domain significance:** In medical applications, data acquisition involves substantial costs due to privacy regulations, expert annotation needs, and patient recruitment challenges. Understanding the efficacy of synthetic pretraining data has direct practical value for researchers facing these constraints.
>
> **Request for substantive feedback:** We note the review lacks specific technical critiques of our methodology, analysis, or conclusions. We welcome concrete feedback on experimental design, soundness of our findings, or suggestions for additional experiments that would strengthen the work.

---

### Meta-Review · Area_Chair_zFiA · 2025-12-24

**Summary:**

The paper studies whether simulated ECGs can be used for pretraining a transformer with a masked prediction objective in lieu of real ECGs. These pretrained representations are evaluatd on mainly rhythm issues that can be read off from an ECG. The study is interesting and relatively well executed, but it really seems like it belongs somewhere else. Another way to frame this was via a question in my mind when I was looking at the paper was what insight do I get for some class of general AI/ML problems. The best I could think of was that simulation (under some assumption of quality) is good enough for pretraining objectives developed elsewhere. There was also some weakness to me in rhythm classification as it's easy, but that's minor

**Reviewer Concerns:**

The main reviewer concern about that's outstanding is one of fit. Even the positive reviewer questions whether this type of paper should appear in ICLR or whether it's better suited for a journal.

**Reviewer Scores:**

r4Jv - remains the same (though I discounted this review)

GHNt - the response addresses the key question of what happens if you scale pretraining data. the score would likely go up to a 4

ThrN - positive and remain the same

eCu6 - remain the same. the primary concern of unclear generalizable insight would remain, though the PCG experiments would lessen the feeling about the results only holding up for ECGs

---

### Decision · Program_Chairs · 2026-01-26

Reject